# Wildfires on a changing planet

Olivia Haas [1,2] ✉, Iain Colin Prentice [1,2] & Sandy P. Harrison [1,3]

The distribution of wildfires on Earth will change as climate, land-use, and vegetation change. We use global empirical models of burnt area, fire size and fire intensity to explore future wildfire trajectories under ~1.5 and 3-4 °C warming with middle of the road future socio-economic conditions. Even under ~1.5 °C warming we find a change in wildfire patterns by the end of the 21$^{st}$ century with reduced burning in tropical regions driven by changes in human activity but larger and more intense wildfires in extra-tropical regions driven by changes in climate and $CO_2$. With low climate change mitigation, burnt areas increase greatly across all vegetation types, overwhelming the current global decline. These findings suggest that even with ambitious climate change mitigation, current fire-suppression policies will fail in much of the world and mitigation scenarios that rely on expanding forest areas will be unrealistic unless they are designed with wildfire risks in mind.

There is growing concern that recent increases in wildfire occurrence and intensity in many regions will persist in the future[1–6]. Nevertheless, satellite observations show that burnt area has declined globally over the last two decades, most notably in the tropical savannas of South America and Africa and grasslands across the Asian steppe[7,8]. Most of this decline has been associated with human activities such as cropland expansion, although climate and $CO_2$ changes may also have played a role[8,9]. Weather conditions promoting fire, as measured by fire danger indices, consistently increase in future climate-change projections[1,9–11], contributing to the concern raised by recent megafires, which have been shown to be increasing in frequency in some regions[12]. However, these physical indices do not take account of the potential for changes in vegetation to moderate the effect of future climate changes: warmer and drier conditions increase the incidence of fire weather, for example, but the effect that changes in climate may have on vegetation is complex, and its impact on fire is not straightforward. On the one hand, warmer conditions can reduce fuel loads and fuel continuity, decreasing fire risk[9,12–15]. However, warmer and drier conditions can also have indirect effects on vegetation composition and type, which in turn influences landscape flammability. They also influence individual species, changing plant traits and vegetation structure[16–18]. Analyses that consider only physical indices such as the fire weather index are often used to assess future fire risk, but do not account for other influences on wildfires, such as vegetation and human activity[4]. Empirical analyses have considered vegetation and human impacts on

fire risk and are being used in the Inter-Sectoral Impact Model Inter-comparison Project (ISIMIP) to assess potential future wildfire regimes under climate change[4,6,19] However, these studies tend to focus on extreme fire behaviour as well as changes in high burnt areas[6] with little attention paid to other fire properties such as size or intensity. Process-based global models, which simulate full fire behaviour, still differ considerably in their predictions of historic trends[20] and do not skill-fully capture wildfire seasonality and interannual variability[21], although recent efforts to improve these models have led to greater constraint in the longer-term trends[22]. Nevertheless, the ability of these process-based models to provide insights into future trends remains limited. Given the impact of wildfires on natural resources and human well-being[6], a better understanding of potential future wildfire trajectories is needed to devise appropriate management strategies and mitigate risks.

In this study, we use recently developed empirical generalised linear models (GLMs) of monthly mean burnt area, median fire size and median fire intensity driven by lightning ignitions, climate, vegetation properties, topography, and human factors (see Table 1)[23]. A quasi-binomial model was fitted to burnt area, and a quasi-Poisson model to fire size and intensity. These models have been shown to capture modern spatial patterns in fire properties[23]. In addition, they have shown to have the ability to simulate the impact of large changes in climate during the last glacial in a realistic way, which other global fire models have yet to demonstrate[24]. Another innovative feature of these

[1]Leverhulme Centre for Wildfires, Environment and Society, Imperial College London, London, UK. [2]Department of Life Sciences, Georgina Mace Centre for the Living Planet, Imperial College London, Ascot, UK. [3]Geography and Environmental Science, University of Reading, Reading, UK. ✉e-mail: oliviammhs6@gmail.com

**Table 1 | Summary of modern and future realistic experiments and the associated sensitivity experiments**

| Experiments | |
| --- | --- |
| Realistic modern | 2010–2015 WFDE5 climate[100] |
| | 2010–2015 annual global average $CO_2$ (gml.noaa.gov/ccgg/trends/) |
| | HYDE 3.2 population density and cropland[109] |
| | GRIP4 road density[107] |
| Realistic future | 2090–2100 climate from GFDL-ESM2M, HadGEM2-ES, IPSL-CM5A-LR, and MIROC5 for RCP2.6 and RCP6.0[25] |
| | 2090–2100 average $CO_2$ for RCP2.6 and RCP6.0 |
| | SSP2 socioeconomic pathway, average 2090–2100 |
| *Sensitivity experiments* | |
| Climate change only | 2090–2100 climate from GFDL-ESM2M, HadGEM2-ES, IPSL-CM5A-LR, and MIROC5 for RCP2.6 and RCP6.0[25] |
| | *$CO_2$ and human factors held constant at modern levels* |
| $CO_2$ change only | 2090–2100 average $CO_2$ for RCP2.6 and RCP6.0. |
| | *Climate and human factors held constant at modern levels* |
| CO2 and climate change only | 2090–2100 climate from GFDL-ESM2M, HadGEM2-ES, IPSL-CM5A-LR, MIROC5 for RCP2.6 and RCP6.0[25] |
| | 2090–2100 average $CO_2$ for RCP2.6 and RCP6.0. |
| | *Human factors held constant at modern levels* |
| Human activity change only | SSP2 socioeconomic pathway, average 2090–2100 |
| | *Climate and $CO_2$ factors held constant at modern levels* |

Additional information on the source and availability of the data for each experiment can be found in Table 3 (see 'Methods').

empirical models is that they make use of vegetation inputs which account for the physiological effect of $CO_2$ on vegetation, independently of climate, providing a way to examine the effect of $CO_2$ fertilisation on wildfires separately using a first-principle productivity model, the P Model[24]. We used these empirical models to examine the consequences of two scenarios: one where global warming by the end of the century is kept under 2 °C and another with a global warming of 3–4 °C by 2100 CE, as run by four General Circulation Models (GCMs). Both scenarios assumed a 'middle of the road' socioeconomic pathway (see 'Methods'). Under this scenario, population increases 9 billion by the end of the century. Most of this population increase occurs in Africa, with increases also simulated in Australia, the Eastern United States and India. Decreases are simulated in Brazil, Southeast Asia, China and Russia. Europe shows both increases and decreases in population (see Figs. S27–S34). Road density increases, with the largest increases in Africa and India. Cropland increases are concentrated in Europe, some regions of Eastern Africa, the Middle East and Southeast Asia (see S27–S34). We also ran a series of sensitivity experiments in which a set of factors (climate, $CO_2$, climate and $CO_2$, human activities) were allowed to change while others were held constant, to diagnose the causes of simulated changes in fire properties under each scenario (see Table 1 and Fig. 1). We show that under the ambitious climate change mitigation scenario reduced burning in tropical regions driven by changes in human activity still dominates the global signal but larger and more intense wildfires in extra-tropical regions driven by changes in climate and $CO_2$. With low climate change mitigation, burnt areas increase greatly across all vegetation types, overwhelming the declining trend currently dominating the global signal, with large and intense wildfires encroaching into the northern high latitudes.

## Results

### Burnt area change under high climate change mitigation
The global decrease in burnt area (see Fig. 2) is primarily driven by a decrease in burnt area in tropical evergreen and deciduous forests, tropical grasslands, tropical shrublands and tropical savannas. These biomes account for 83% of global burnt area under modern conditions but only 77–81% at the end of the twenty-first century (see Table 2 and Fig. 3). This decline was primarily driven by increased human activity, as measured by road density and cropland cover and was particularly concentrated in Africa, India, and Southeast Asia (see Figs. 2 and 4). In these regions, increased landscape fragmentation was the key driver. Globally, there was no modelled global decline in burnt area when socioeconomic variables were held constant, highlighting the importance of human activities in reducing burnt area (see Supplementary

Section 7). However, human activity does not explain the modelled reduction in the tropics entirely. When holding socioeconomic variables constant, reductions in burnt area were still simulated in tropical grasslands, xerophytic shrublands, evergreen and semi-deciduous forests because of increasing dryness (see Supplementary Sections 4–6). Climate changes, for example, in Central America and the Amazon basin (primarily decreasing dry-day seasonality), and vegetation changes (decreasing gross primary production (GPP)), also drove a reduction in wetter tropical regions of South America (see Figs. 2 and 4). Furthermore, even when considering changes in human activity, the decrease in burnt area does not occur homogenously across all tropical regions. For example, an increase in burnt area is simulated in the tropical regions of South America (see Table 2 and Fig. 2). In drier tropical biomes, increased dryness and GPP also led to increases in burnt area, explaining some of the variability between tropical regions.

Whilst the global signal is dominated by the reduction of burnt area in tropical regions, the number of grid cells overall in which the probability of burning exceeded our ignition threshold was 27–30% greater by the end of the twenty-first century than under modern conditions, and most (87–88%) of this increase occurred in the northern extra-tropics. In these regions, the probability of burning increases above a threshold under which no wildfire activity is expected to occur. Thus, these increases represent an encroachment of wildfires into areas where previously there were none. This is the case, for example, in Scandinavia and regions of Siberia.

Increases in burnt area in temperate and boreal forests, and in tundra (see Fig. 3) were driven by decreased precipitation (as reflected by an increase in the number of dry days) and increased atmospheric dryness (vapour pressure deficit (VPD)) (see Fig. 4). This was simulated in both climate-only and $CO_2$-only sensitivity experiments (see Supplementary Sections 5 and 6). However, these regional increases in burnt area were not sufficient to counteract the impact of reductions in the tropics on the global decline in burnt area (see Fig. 2, Table 2).

### Burnt area change under low climate change mitigation
In stark contrast to the high climate change mitigation scenario, under low climate change mitigation, there is an increase (18–60%) in global burnt area. Increased burning is seen across all vegetation types, even in the tropics (see Fig. 2). Human-driven reductions in burnt area in tropical regions are offset by the combined effects of changes in climate and $CO_2$. In temperate regions, vegetation types where burning increased under the high climate change mitigation scenario had an even larger increase under the low climate change mitigation scenario (see Fig. 2, Table 2).

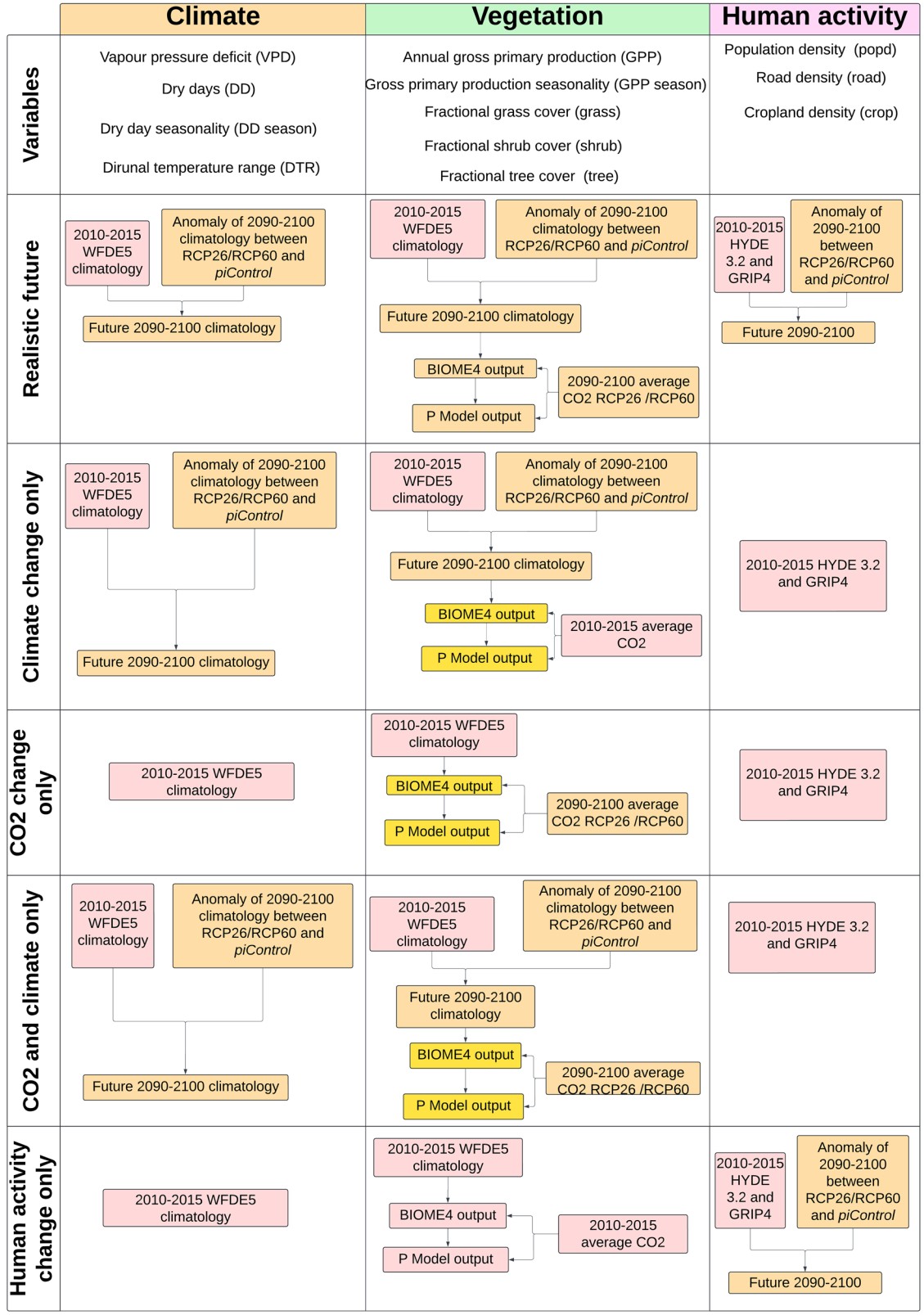

**Fig. 1 | Flowchart of the experimental setup.** Flowchart of the experimental setup showing realistic future experiments and the associated sensitivity experiments. The high climate change mitigation scenario refers to end-of-century RCP2.6 and the low climate change mitigation scenario refers to end-of-century RCP 6.0. Additional information on the source and availability of the data for each experiment can be found in Table 3 (see 'Methods'). Created in Lucid (lucid.co).

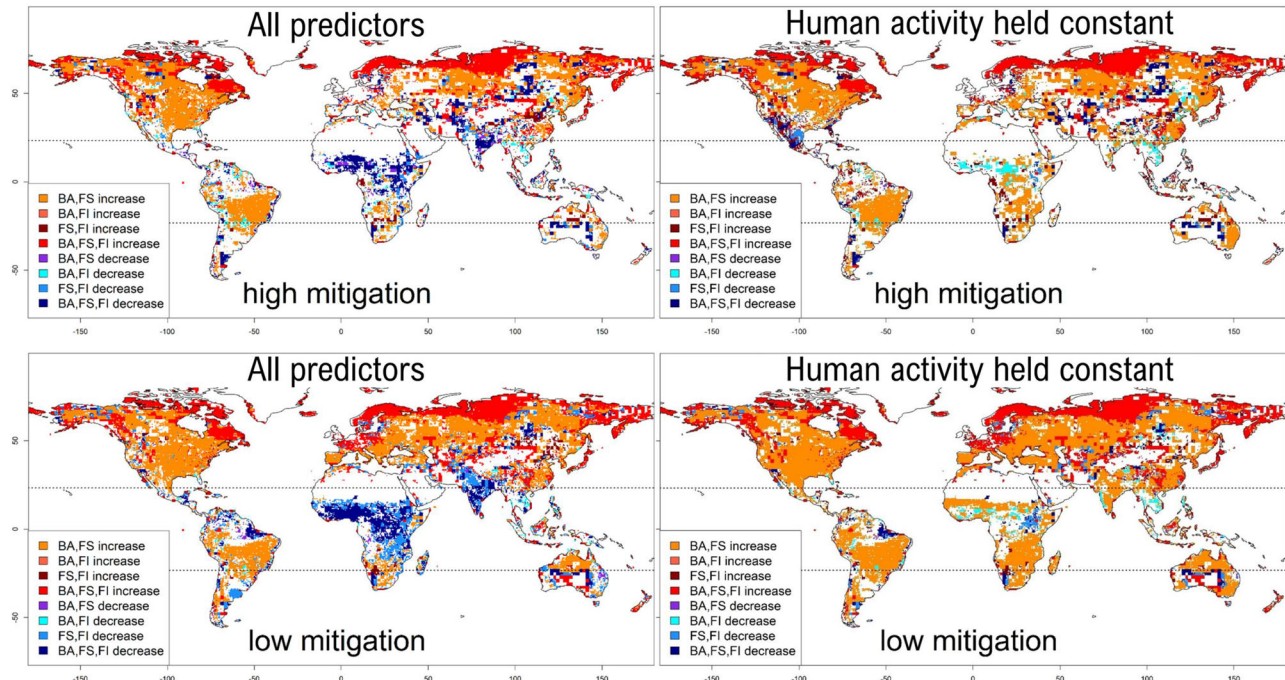

**Fig. 2 | Change in fire regimes by 2100.** Regions of change (>5%) in burnt area (BA), fire size (FS) and fire intensity (FI) when all predictors are varied on the left with the high climate change mitigation scenario (top) and the low climate change mitigation scenario (bottom) and when all predictors are varied except for human activity on the right with the high climate change mitigation scenario (top) and the low climate change mitigation scenario (bottom). The tropics and extra tropics are delimited by dotted horizontal lines. Regions where all three properties increase are shown in red, regions where all three properties decrease are shown in dark blue (only increases of more than 5% in all three properties are considered). Base maps are produced using rnaturalearth.

**Table 2 | Total burnt area per biome and associated global percentage under modern, high and low climate change mitigation scenarios**

| Biome | Modern day | | High climate change mitigation | | Low climate change mitigation | |
|---|---|---|---|---|---|---|
| | Burnt area (km²) | Global percentage | Burnt area (km²) | Global percentage | Burnt area (km²) | Global percentage |
| Tropical evergreen forest | 817,813 | 20.14 | 842,705 | 24.20 | 1,317,980 | 21.56 |
| Tropical deciduous forest | 604,130 | 14.88 | 558,277 | 16.03 | 766,776 | 12.54 |
| Tropical grassland/shrubland/savanna | 2,010,784 | 49.52 | 1,372,585 | 39.42 | 2,281,542 | 37.31 |
| Temperate deciduous forest | 44,004 | 1.08 | 4,9132 | 1.41 | 121,593 | 1.99 |
| Temperate grassland/shrubland/savanna | 135,381 | 3.33 | 219,644 | 6.31 | 259,302 | 4.24 |
| Conifer and mixed forest | 395,355 | 9.74 | 394,149 | 11.32 | 1,284,318 | 21.0 |
| Montane taiga evergreen forest | 12,337 | 0.30 | 23,794 | 0.68 | 36,973 | 0.60 |
| Montane taiga deciduous forest | 38,160 | 0.94 | 19,254 | 0.55 | 31,589 | 0.52 |
| Tundra | 2,795 | 0.07 | 2,413 | 0.07 | 14,355 | 0.23 |
| Global total | 4,059,702 | | 3,481,953 | | 6,114,430 | |

Changes in climate are more important than changes in vegetation and human activity in driving burnt area increases in the low climate change mitigation scenario, although vegetation changes still contribute to increasing burnt area in sub-Saharan Africa (Angola, Zambia, Malawi, Mozambique) (see Fig. 3 and Supplementary Section 5). Human-driven reductions in burnt area in tropical regions are offset by the combined effects of changes in climate and $CO_2$ (see Supplementary Sections 6 and 7). The role of $CO_2$ is important, as the increases in burnt area do not occur in response to changes in climate alone (see Supplementary Section 4). Increases in dryness and grass cover drove an increase in burnt area in North America and increases in GPP drove an increase over much of Eurasia (see Figs. S7–S14). When only socioeconomic changes are considered, we model large

decreases in burnt area globally, and particularly in tropical regions of Africa and India, with much smaller reductions in South America and Southeast Asia (see Supplementary Section 7). This probably reflects deforestation fires in the Amazon and Indonesia, which, though reduced under the future socioeconomic scenario, still occur (deforestation fires are captured through cropland cover and population density in the models).

### Fire size and intensity change under high climate change mitigation conditions

Under the high climate change mitigation scenario, global fire size and mean global fire intensity are increased compared to modern conditions (by 23–36% and 2–5% respectively, see Tables 2 and S4). Global

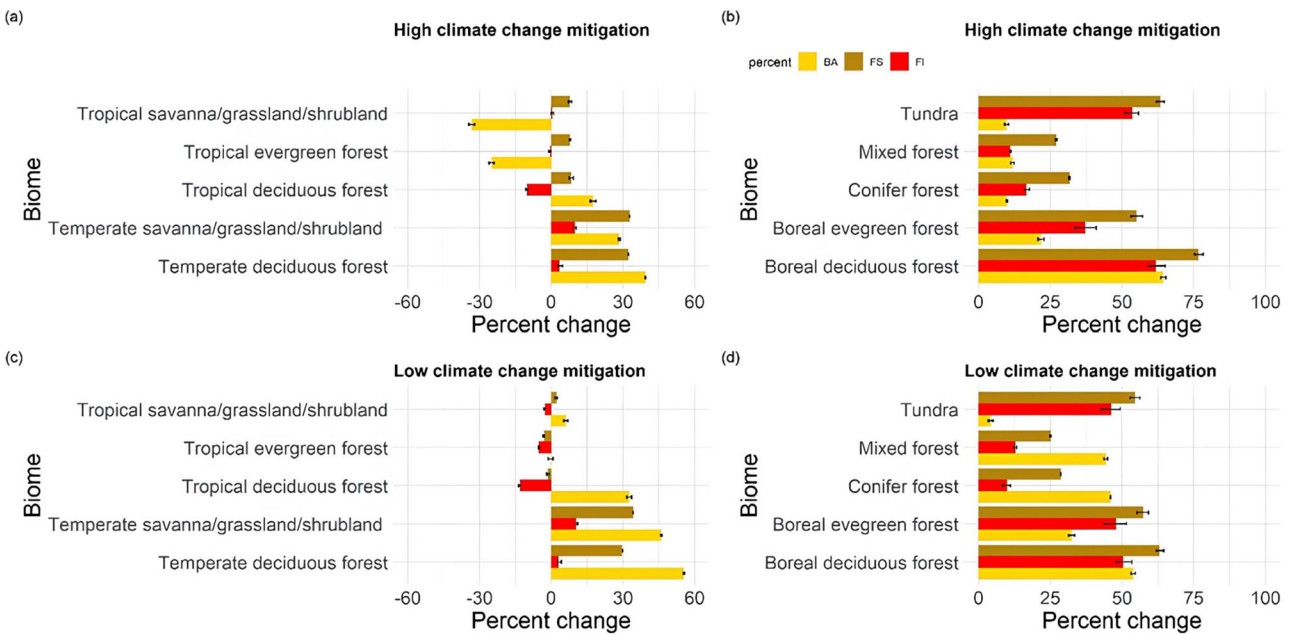

**Fig. 3 | Change in fire regimes by biome.** Percentage change in burnt area (BA), fire size (FS) and fire intensity (FI) per biome for the high climate change mitigation scenario (top) and the low climate change mitigation scenario (bottom). **a**, **c** Tropical biomes and **b**, **d** extra-tropical biomes. The error bars represent the 95% confidence interval. A more detailed version of this figure is available (see Fig. S6).

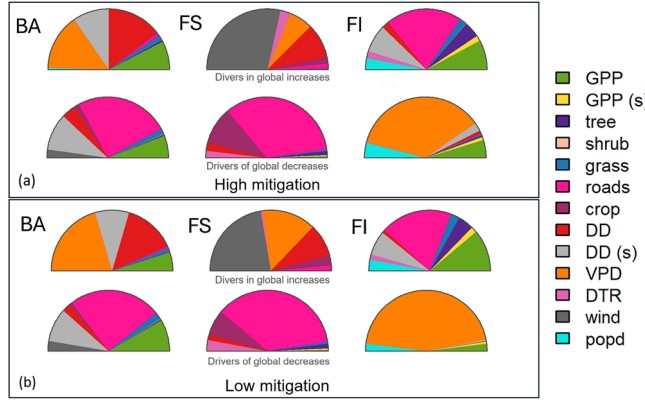

**Fig. 4 | Predictor contribution.** Predictors responsible for driving changes in burnt area (BA), fire size (FS) and fire intensity (FI), respectively, of **a** the high climate change mitigation scenario and **b** the low climate change mitigation scenario. The top row shows predictors driving increases and the bottom row shows predictors driving decreases in the fire properties.

increase in fire size and fire intensity primarily reflects increases in the northern extra-tropics (Fig. 2). The largest increases in fire intensity and fire size occurred in boreal forests and shrub tundra north of 60°N (see Figs. 2 and 3).

Increased fire size in these northern regions was driven by increasing dryness and VPD. Modelling increases in fire intensity in these regions was driven by increased GPP, tree cover, but also road density. Larger fires in South America are driven by increased dryness in the tropical rainforests.

A decrease in fire intensity is modelled in the tropics (see Fig. 3). Fire size also decreases in tropical biomes, though large increases are simulated in South America, dominating the signal when looking at trends within these biomes (see Fig. 2). Modelled decreases in fire size and fire intensity in tropical Africa, the Middle East and India were primarily driven by changes in human activity (see Figs. S7–S14).

Outside these regions, decreases in fire intensity were primarily driven by increasing VPD (see Supplementary Section 3).

## Fire size and intensity change under low climate change mitigation conditions

Under the low climate change mitigation scenario, both global fire size and fire intensity are reduced compared to a high climate change mitigation scenario. Under the low climate change mitigation scenario, global mean fire size increased 19–32% compared to modern, while the trend in global mean fire intensity was inversed, with a global decrease of 1–12% compared to modern. Although the increase in fire intensity was somewhat smaller than in the high-mitigation scenario overall, this was primarily driven by decreases in temperate grasslands and forests, including conifer forests, compared to the high-mitigation scenario (80% decrease), as well as changes in tropical regions. The overall modelled decrease in fire intensity in tropical regions doubled (49%) compared to the high climate change mitigation scenario, except for tropical grasslands, where fire intensity increased (see Fig. 2). Increased GPP was the major cause of increased fire intensity in northern latitudes, though changes in tree cover and grass cover also played a role. For fire size, increases in the number of dry days and dry day seasonality and changes were the main drivers in this region (see Supplementary Section 3). The largest increases in fire intensity and fire size compared to modern were concentrated in taiga forests and tundra above 60°N. The increase in fire size was substantially higher than in the high climate change mitigation scenario across all northern biomes, including temperate forests (50% increase).

## Discussion

Model simulations show major changes in the geographic patterns of wildfire occurrence by the end of the twenty-first century, under both high- and low-climate change mitigation scenarios and middle-of-the-road socioeconomic changes. Both experiments show a reduction in burning in the tropics but increased burning elsewhere. Our results suggest a general trend towards larger and more intense wildfires in northern latitudes, under both high and low climate change mitigation scenarios. The current global trend of decreasing burnt area may

continue under high climate change mitigation efforts, but is unlikely to continue under low climate change mitigation efforts, where changes in climate promoting wildfires overwhelm the effects of human activities in reducing wildfires, particularly in the tropics. Differences in trajectories between the tropical regions can be partly explained as reflecting differing human impacts on wildfire regimes, for example, in South America, deforestation fires are much more prominent than in Africa, where the impact of human activity on wildfire activity is related to cropland expansion[7]. Given that simulations of future fire activity have not been performed in a systematic way, it is difficult to assess agreement between previous studies over the sign of the global trend in future fire probability. However, our conclusions are in line with previous studies[6,25–29] in showing a redistribution of global wildfires under future conditions, with increases in the northern high latitudes and decreases in tropical regions. There are still differences in regional trends, especially in the Amazon, where our simulations show large increases in fire activity, whereas previous studies show either non-significant changes[16,18] or decreases in fire probability[6,26]. Furthermore, although most studies show the largest increases of fire probability in the northern latitudes, some regions show decreasing fire activity despite an overall increasing trend[6,25,26,28].

Our analysis provides insights into key regions in which global fire regimes are sensitive to changing conditions and suggests distinct trajectories for different climate change mitigation scenarios. Furthermore, unlike most projections of future fire regimes, we distinguish between changes in burnt area and changes in fire intensity and size. Our analyses make it clear that some regions will experience substantial changes in fire intensity even though the changes in burnt area are small, whilst other regions will experience important changes in overall burnt area, but the changes in fire size and intensity will be more moderate. It is important to note that these results remain the outputs of a modelled system, and despite the models being extensively trained and evaluated, uncertainties remain in the response of fire regimes to human, climate and vegetation changes. Nevertheless, the relative balance of drivers identified here is consistent with independent analyses, suggesting that the model captures the key processes underlying simulated trends, despite underestimating the sensitivity of the system to human activity. In some regions, increases in burnt area are associated with decreases in fire size and intensity. Direct feedback between these different fire properties (for example, how fire activity may constrain fire intensity[30] or between fire and vegetation) was not considered here. These feedback may potentially offset some of the differences in the diverging trends seen between burnt area, fire size and fire intensity, as well as explain differences with previous studies. To address the impact of feedback on different aspects of the fire regime itself, it will be necessary to move from an offline modelling exercise to a dynamic coupling of wildfires in an Earth System Model. Nevertheless, although the lack of consideration of these feedback may increase the uncertainty of our analysis, it is not surprising to observe divergence in simulated behaviour of the different fire properties–given that burnt area, fire size and fire intensity are driven by different variables. Previous studies have classified fire regimes under modern conditions by identifying distinct associations of fire properties in the observational record and assuming they are coupled by biophysical constraints[31–33]. Whilst we model different fire properties independently, except for the constraint of the probability of fire occurrence being necessary for modelling fire size and intensity, our results suggest that unique combinations of fire properties observed today may not remain constant under a changing climate. Burnt area, fire size and fire intensity have different global controls and the relationships between them may change in the future, reflecting differing sensitivities to environmental drivers. This result has considerable implications for the distribution of fire regimes on Earth and implies that efforts to model future wildfire regimes should not hold the relationships between different fire properties as fixed.

This analysis focuses on mean long-term background rates of fire properties. The models were explicitly designed to capture the longer-term sensitivity of global mean burnt area, median fire size and median fire intensity to large-scale changes in climate, vegetation, land cover and human activity rather than the interannual variability or short-term trends. This study relies on a space-for-time substitution method[34] whereby changes in the spatial patterns of burnt area, fire size and fire intensity are used to infer how these fire properties may change under dramatically different climate, vegetation and human conditions. This method was chosen given that the temporal availability of fire data is currently limited, and that the controls of interannual variability of fire on Earth are not well understood[21,35]. Furthermore, these models have been shown to simulate global burnt area under the radically different forcings of the Last Glacial Maximum realistically[24], which gives us confidence in their ability to respond to large-scale changes in forcing. A consequence of this method is that the models are not designed to capture short-term interannual variability or events such as extreme fire behaviour. Extreme fire events may increase in frequency in the future[12], particularly in regions where conditions become drier[36]. However, given the relatively small number of extreme fire events in the global record, they remain challenging to model[4,37]. Analysis of the interannual performance of the GLM models over the last two decades highlights that the models used here are not sensitive enough to changes in land use and human activity. We argue that this is in part because the human processes by which burned area is modulated are sub-grid cell processes, and as such, the coarse variables and scale used here fail to capture the processes adequately. More research is needed into constraining the influence of human activity on wildfires at these sub-grid cell scales. Furthermore, given that we use a linear model to project future road density, the expansion of roads into the northern high latitude, which is responsible in part for increases in fire intensity, may not be fully realistic. Nevertheless, we see that the impact of fragmentation on landscapes has a very different effect depending on the fire property and on whether the ecosystem is adapted to fire or not. Nevertheless, the fact that there is a large decrease in global burnt area when only changes in human activity are considered, and that no global decline is simulated when only climate and vegetation remain important, results. Furthermore, artificially increasing the models' sensitivity to human activity did not dramatically alter the conclusions that are presented here, highlighting the robustness of these findings.

There is a high level of uncertainty associated with modelling future changes in wildfires. The aim of this analysis was not to provide predictions but to show the sensitivity of wildfires globally to future changes in their different drivers. We have addressed issues of uncertainty in future forcing using a high and a low climate change mitigation scenario, as well as the uncertainty in the response of climate and vegetation inputs to these changes in forcing using multiple GCMs. An additional uncertainty that is more challenging to address is that associated with the fire response itself. We have characterised the uncertainty in this response by considering a 95% confidence interval on the modelled relationships (see Fig. 5), which shows that globally the uncertainties are relatively small. However, GLMs do provide less flexibility in the potential fire response to climate and land use forcing than other non-linear modelling approaches. It has been argued that machine-learning models have greater predictive power than linear modelling tools, but they have limited ability to predict outside of the range on which they were trained. In addition, a major part of this analysis was disentangling the effects of independent variables in driving the change in BA, FS and FI. This is something that is better done through linear modelling than a machine-learning approach. Most process-based models, for example, rely on rate of spread equations to link fire starts to burnt area, and estimate fire intensity as the product of fire spread and fuel consumption[37–39], thus intrinsically linking fire numbers, size, burnt area and intensity. The predictions of

machine-learning approaches do not make these implicit linkages but focus on emergent properties of the fire regime and generally include a large number of predictors, making them more difficult to interpret and not well suited for sensitivity analyses[8]. The GLM approach adopted here, in which burnt area, fire size and fire intensity are modelled as a response to environmental factors independently of one another, provides a powerful way of testing the sensitivity of different fire properties to specific controls and combinations of those controls. It is important to note that these results remain the outputs of a modelled system. Despite extensive model training and evaluation, uncertainties remain in the response of fire regimes to human, climate and vegetation changes. However, the relative balance of drivers identified here is consistent with independent analyses, suggesting that the model captures the key processes underlying observed trends, despite underestimating the sensitivity of the system to them.

We have not considered the potential impact of changes in lightning frequency, and therefore lightning-caused ignitions, on future wildfire trajectories because there are large uncertainties in future lightning projections, even in terms of the sign of the global trend[40,41]. Increased lightning frequency would be expected to lead to even larger increases in burning[42], especially in high-latitude regions, since drier fuels in boreal forest ecosystems would increase lightning ignition efficiency[43–46]. In this analysis, we consider differences between vegetation types through the distinction of fractional tree, grass and shrub cover only. These fractions are obtained by averaging values for each biome globally. As such, we do not observe the response of within-biome responses in different regions of the world. Whilst this could explain some of the regional differences observed, overall, we do not believe that a substantial amount of information is lost through this method. Research has consistently shown that current biomes are not necessarily representative of differences in fire properties and that other plant traits which are not captured in theses classification but that relate to flammability and recovery may be more important[15,47,48]. In addition to this, predictors relating to vegetation amount have been shown to be more important in capturing burnt area than predictors relating to vegetation type[49]. As such, whilst observing these differences could be interesting, we argue that the uncertainty surrounding the simulated trends in within and between biome responses would be large. Furthermore, our vegetation productivity estimates are independent of plant functional type (PFT) parameters and do allow us to observe regional differences between regions with similar conditions, reducing the uncertainty introduced by fixed vegetation categories. We use a single model approach to simulate our vegetation productivity predictors. We therefore cannot assess the uncertainty related to model specification for our vegetation inputs. We were interested in producing a PFT independent analysis, and the optimality-based productivity model used here was the only model that allows this[50]. There are limitations to the model[51], but it has been shown to outperform other vegetation models in predicting seasonal changes in leaf area[52] and can quantitatively explain observed changes in vegetation as a response to environmental change[53] as well as global greening patterns[54]. Our analysis highlights the sensitivity of future fire regimes to human activities. However, there are many assumptions and large uncertainties associated with the socioeconomic scenarios used in these analyses. This analysis only considers one future socioeconomic scenario (SSP2). Projecting how future socioeconomic systems will respond in the future is more challenging than projecting the response of the climate to changes in future forcing. We therefore do not comment on the uncertainty relating to future human activity but work with the 'middle-of-the-road' scenario and focus on the differences between the modern-day and one actualisation of the future SSP2 scenario. The largest uncertainty in future scenarios is how policy changes will affect human activities, including landscape and wildfire management[55].

Shifts in wildfire occurrence, as seen in both low and high climate change mitigation scenarios, will have large ecological and human impacts. There are signs that these are already happening[3,6]. Given the widely differing trajectories of wildfire regimes in different regions, contrasting management strategies are likely to be required. A better understanding of how human activity influences wildfire regimes is needed to implement any successful management strategy[56–58]. In the tropics, for example, a potential consequence of the predicted decrease in burnt area, fire size and fire intensity is woody encroachment, which impacts biodiversity, the carbon cycle, and land-atmosphere interactions and has consequences for local economies, livestock production, tourism, and pastoralism[59–61]. High-intensity fires have been used to limit woody encroachment, but with limited evidence of success[62,63]. Across the African continent, a broad range of fire management strategies are already in place in savanna protected areas[64] and the potential for fire management to help finance local restoration and economies has been highlighted[65]. Under a changing climate, more investment is needed in these regions to ensure that fire management is successful. Involving local communities and knowledge in these efforts will provide the greatest resilience[66–69]. In the extra-tropics and other regions where burnt area, fire size and fire intensity are predicted to increase, increased risks to life, property, and commercial activities are a major concern[6,70]. Previous fire policies may influence future management and what outcomes are desirable. As a result of a long history of fire exclusion in the United States, for example, introducing fire to forests where fire has been previously excluded may reduce high-severity fires but increase total burnt area[71,72]. Regional and landscape level models could be used to address these management issues[73]. In currently fire-prone regions such as Mediterranean ecosystems, the viability of fire suppression policies needs to be reassessed[74–76] Furthermore, management practices may not be well defined in regions where wildfires have been infrequent in the past[74–76]. In the northern latitudes, the most intense wildfires occur under the high climate change mitigation scenario, and it is under these conditions that successful fuel load management and increased preparedness in the wildland urban interface will be most needed[77–80].

Increases in burnt area in these regions under a low climate change mitigation scenario will require moving beyond fire-suppression measures and towards adaptive ones in which communities learn to live with fire[74–79], as they will be more common, though less destructive, than under a high climate change mitigation scenario. Forest planting and restoration projects have received much attention as climate change mitigation tools[80–84] but there has been limited consideration of the effect of wildfires on such projects. Wildfire management will need to be incorporated into the design of afforestation schemes, given that our simulations show substantial changes in fire regimes in the future in most regions of the world, including the tropics under the low climate change mitigation scenario. Indeed, changes in wildfire regimes may preclude implementation of afforestation schemes in some areas[85,86] and wildfire management often conflicts with retaining native ecosystems, cultural landscapes, and local communities[87,88].

## Method

### Generalised linear models for burnt area (BA), fire size (FS) and fire intensity (FI)

We used three GLMs to simulate the spatial patterns of burnt area, fire size and fire intensity[23,89–91]. These models were derived from a common set of sixteen predictors representing vegetation (GPP) land cover (grass cover, shrub cover and tree cover), climate (VPD, dry days, diurnal temperature range and wind speed), topography (vector ruggedness metric, topographical position index), human activity (cropland cover and road density), and ignitions (population density and lightning) and fitted to monthly burnt area (BA), monthly median fire size (FS) and a monthly median fire intensity metric (FI) (see Table 3).

**Table 3 | The GLM models initially used a set of 16 predictors**

| Predictors | Abbreviation | Value | Rationale |
|---|---|---|---|
| **Climate** | | | |
| Dry days | DD | Mean monthly number of dry days | Flammability (To capture drying out of fuel over multiple days) |
| Dry day seasonality | (DD(s)) | Range in value of the monthly number of dry days divided by the mean monthly number of dry days | Fuel-build up(To account for regions with strong wet and dry seasons) |
| Vapour pressure deficit | VPD | Maximum monthly vapour pressure deficit | Fuel-build up Flammability (To capture atmospheric humidity's constraint on fuel-build up, fuel-drying, flammability) |
| Diurnal temperature range | DTR | Mean monthly diurnal temperature range | Flammability(To capture the drying out of fuel in daily time frames) |
| Wind speed | wind | Wind speed of the hottest month | Ignition Fire spread |
| **Vegetation** | | | |
| Annual gross primary production | GPP | Sum of annual gross primary production | Fuel load (To capture fuel accumulation) |
| Gross primary production seasonality | GPP(s) | Range in value of monthly gross primary production divided by the mean monthly gross primary production | Fuel load(To account for seasonal variation in fuel load) |
| Grass cover | grass | Fractional annual grassland cover | Flammability |
| Shrub cover | shrub | Fractional annual shrubland cover | Flammability |
| Tree cover | tree | Fractional annual tree cover | Flammability Fuel load |
| **Human activity** | | | |
| Population density | popd | Annual mean population density | Ignitions |
| Cropland cover | crop | Fractional annual cropland cover | Suppression and land fragmentation |
| Road density | roads | Annual mean road density | Suppression and fragmentation |
| **Topography** | | | |
| Vector ruggedness index | VRM | Mean value (constant) | Land fragmentation and effects of topography |
| Topographical position index | TPI | Mean value (constant) | Land fragmentation and effects of topography |
| **Ignitions** | | | |
| Lightning | light | Mean monthly lightning ground strikes | Ignitions |

**Table 3 (continued) | The GLM models initially used a set of 16 predictors**

| Predictors | Abbreviation | Value | Rationale |
|---|---|---|---|
| Response variables | Abbreviation | Value | Transformation and link function |
| Burnt area | BA | Monthly mean burnt area fraction | None (logit link function) |
| Fire size | FS | Monthly median fire size (km$^2$) | Min-max normalised (log link function) |
| Fire intensity | FI | Monthly median FRP divided by the square root of median fire size (W km$^{-1}$) | Min-max normalised (log link function) |

The individual predictors, as described below, as well as the rationale behind their inclusion in the model.

For all monthly varying variables, a seasonal climatology was derived over 2010–2015 and for each grid cell, the value of the month with (on average) the largest DTR, the highest VPD and the mean number of dry days and mean lightning was selected. Wind speed was taken from the hottest month of the year. Fractional BA is equated with the probability of burning, ranging between 0 and 1, and as such, it is assumed that its probability follows a quasi-binomial distribution. As such, the GLM applies a logit link function. The distribution of FS and FI is hypothesised to follow a power law[35,92]. As such, a quasi-Poisson distribution is assumed for both variables (due to their large overdispersion, with many very small values) and the log link function is applied. GLMs provide interpretable results and, due to their simplicity, require very little computational power, making them an ideal tool for running a large set of sensitivity experiments and providing a way of quantifying independent effects of multiple predictors even when they are correlated[93]. GLMs are useful because they can handle response variables with highly non-Gaussian error distributions, are embedded in a well-established (multiple regression) framework, which allows quantification of the independent effects of multiple predictors even if they are partially correlated with one another and provide partial residual plots showing the effect of each predictor while the others are held constant[93]. The relative importance of each predictor can also be assessed using absolute $t$ values (the fitted regression coefficient of each predictor divided by its standard error). These $t$ values are unitless and scale-invariant[94]. GLMs have previously been used to model burnt area and have been shown to capture both underlying relationships (the relationship between a response and an explanatory variable when all other explanatory variables are held constant, such as the positive relationship between burnt area and productivity) and emergent ones (which are a consequence of combinations of variables, such as the humped-relationship between burnt area and productivity).

Given that BA represents a probability of burning between 0 and 1, the BA model returns a physically meaningful value for each grid cell and does not require further manipulation. However, both the GLM models for FS and FI return values of estimated FS and FI assuming a fire occurs, since the models were fitted to observed data for FS and FI. Given that these models cannot determine if an ignition occurred, they return a value for all grid cells, even those in which the probability of burning is zero.

It is therefore necessary to apply an ignition threshold to study changes in FS and FI[24]. We obtained the ignition threshold value from the distribution of reconstructed BA values in the original model training period (2010–2015) when the observed BA value was zero. There are no systematic biases evident from plotting the residuals of the model, but there is a compression of the range of reconstructed values, leading to apparent over- (under-) prediction at the low (high) extremes. However, the reconstructed values never reach zero, as it is a probabilistic model. Globally, 26% of grid-cells had an observed BA of zero (a total of 14,816 data points and associated fitted values). We took the median value (0.0011) of the fitted values in these grid cells to set the threshold for ignition. The tenth percentile was 0.0001 and the ninetieth percentile was 0.0084. In grid cells in which the burnt area is simulated to be lower than a fractional value of 0.0011, fire size and fire intensity were set to zero. We conducted a True Skill Statistic test[95,96], which balances sensitivity (the ability of the threshold to capture true positives) as well as specificity (how many false negatives the threshold results in) to test the robustness of this threshold value (see Table 4). Since the main concern was to determine under which conditions burning could occur, the selected threshold needed to provide high sensitivity as well as balance specificity. The median value of 0.0011 appears to be a good threshold choice, since it has sensitivity >0.9 and a moderate level of specificity (see 'Methods'). We examined the sensitivity of fire size and fire intensity to the chosen threshold value through testing the 25th percentile and the 75th percentile. Overall,

**Table 4 | Results of the true skill statistic**

| Percentile | Threshold value | TSS | Sensitivity | Specificity | maxTSS |
|---|---|---|---|---|---|
| 0.1 | 0.00013 | 0.094512 | 0.994417 | 0.100095 | 1.094512 |
| 0.2 | 0.00027 | 0.184416 | 0.985105 | 0.199312 | 1.184416 |
| 0.25 | 0.00034 | 0.215538 | 0.979644 | 0.235894 | 1.215538 |
| 0.3 | 0.00047 | 0.266873 | 0.969356 | 0.297516 | 1.266873 |
| 0.4 | 0.00076 | 0.343016 | 0.944393 | 0.398623 | 1.343016 |
| 0.5 | 0.0011 | 0.406785 | 0.914749 | 0.492036 | 1.406785 |
| 0.6 | 0.0018 | 0.460293 | 0.857094 | 0.603199 | 1.460293 |
| 0.7 | 0.0027 | 0.482525 | 0.786251 | 0.696274 | 1.482525 |
| 0.75 | 0.00342 | 0.478716 | 0.728986 | 0.74973 | 1.478716 |
| 0.8 | 0.004 | 0.471159 | 0.687006 | 0.784152 | 1.471159 |
| 0.9 | 0.008 | 0.408702 | 0.514871 | 0.893831 | 1.408702 |

although the absolute values changed, the trends between modern, RCP26 and RCP60 did not. The uncertainty in the 75th percentile was much larger than in the 25th percentile and 50th percentile threshold values (see Table 5).

## Experimental design

The choice of future scenarios was motivated by the future wildfire simulations conducted by the ISIMIP for the United Nations Environment Program[6]. Climate and human activity data from the ISMIP2b protocol were selected because the protocol provides a high (RCP 2.6) and a low (RCP 6.0) mitigation scenario independent of socioeconomic activity pathway scenarios, so that users can make their own coupling of future climate and socioeconomic scenarios[97]. This is not the case for the newer ISIMIP3b protocol, where climate and socioeconomic scenarios are more closely coupled. Given the large uncertainty that remains in building future socioeconomic pathways, our approach minimises uncertainty around future human activity. We selected a single pathway for both the high and the low climate change mitigation scenario, which was most similar to current socioeconomic trajectories, the 'Middle of the road' socioeconomic activity pathway (SSP2). The use of the ISIMIP2b protocol also allowed for a qualitative comparison between the general spatial trends with the outputs from the ISIMIP projections[6]. We conducted five experiments for both scenarios, a realistic experiment, which assumed changes in climate, $CO_2$ levels and human impacts, and four sensitivity experiments (see Table 1). This was done using the climate projections from four different climate models, which participated in the ISIMIP simulations, IPSL-CM5A-LR, GFDL-ESM2M and MIROC5, and HadGEM2-ES[97,98]. We downloaded all the data from the ISIMIP repository: https://data.isimip.org/search/simulation_round/ISIMIP2b/product/InputData/climate_forcing/gfdl-esm2m/subcategory/atmosphere/.

This provided 40 experiments, the spatial patterns of which were compared with a modern-day simulation based on the 2010–2015 period. We explored which predictors each grid cell was responsible for driving the largest increase and decrease in the fire property. For each predictor, we calculated the difference in the model inputs (future experiment – modern-day experiment value) and multiplied this by the corresponding coefficient from the GLM. This approach is analogous to the first-order term of the Stein–Alpert decomposition[99], representing a simplified linear formulation that assumes additive relationships between predictors. This produced spatial maps of predictor contributions, which allowed us to identify the largest positive and negative contributors in each grid cell. This provided maps of the predictors most responsible for increases and decreases in burnt area, fire size and fire intensity between the modern-day and the future. Although the GLM models provide information on the most influential variables globally through the

individual $t$ values of each fitted relationship as through the partial residuals, this grid-cell sensitivity analysis provides a way of quantifying which variable was responsible for the most change within each grid cell between each experiment and the modern-day. This allowed the spatial patterns of the variables responsible for driving the most change to be mapped. Examining the sign of the difference allows us to determine whether the predictor was responsible for an increase or a decrease in the given fire property.

## Data collection and processing

Climate and human activity data from the ISMIP2b protocol were used, providing the high climate change mitigation scenario (SSP2-2.6) and the low climate change mitigation scenario (SSP2-6.0). 'Middle of the road' socioeconomic activity (SSP2) was assumed in both scenarios. To account for potential bias due to differences between models in the varying input datasets and ensure that the simulated changes resulted solely from changes in the climate and vegetation forcing (as opposed to factors such as land-use change), anomalies were calculated between pre-industrial climate and vegetation inputs and the simulated future climate and vegetation inputs under the high and low climate change mitigation scenarios. These anomalies were added to the original climatology data that the model was trained on to provide the future inputs. For socioeconomic changes, we used the anomalies between the mean of the 2090–2100 decade and the original training period (2010–2015) of the SSP2 simulations to account for potential biases in population densities, cropland cover and gross domestic product (GDP). A detailed description of the data collection and processing is provided below.

## Climate inputs

For the modern-day simulation, we used the same climate data (VPD, monthly dry days, dry day seasonality, diurnal temperature range and wind speed as used to construct the original GLM model from WFDE5)[100]. A seasonal climatology was constructed for all variables based on daily data from 2010–2015. For the future experiments, we obtained the climate outputs (daily mean temperature: tas, daily minimum temperature: tmin, daily maximum temperature: tmax, monthly specific humidity: huss, monthly shortwave radiation: rsds, monthly sea level atmospheric pressure: psl, monthly surface wind speed: sfcWind, and daily precipitation: pr) on a daily timestep for the 2091–2100 period for all four climate models from the ISIMIP repository: https://data.isimip.org/ for both the piControl period (extrapolated out to 2091–2100), and the RCP26 and RCP60 respectivelyWe then computed a monthly climatology for mean daily temperature (from tas), diurnal temperature range (from tasmin and tasmax), monthly number of dry days (from daily pr), wind speed (from tas and sfcWind), and VPD (from tas, huss and psl) for all three periods (2091–2100 piControl, 2091–2100 RCP2.6, 2091–2100 RCP6.0) We also collected shortwave radiation (rsds) to obtain cloud cover (described below). Anomalies between the piControl climatologies and the RCP2.6 and RCP6.0 climatologies were then calculated and added the the original modern-day seasonal climatologies from WFDE5. We then followed the methodology in Haas et al.[23] to summarise the climate data to obtain the single-layer variable for each input.

## Vegetation inputs

To design sensitivity experiments which disaggregate the temporal correlation between climate and atmospheric $CO_2$ levels, we used the methodology developed in Haas et al.[24] and summarised below. We obtained modern and future vegetation (tree, shrub, grass) and GPP using the coupled biogeography and biogeochemistry model BIOME4[101,102] and a simple optimality-based model of GPP, the P Model[49,103]. BIOME4 simulates biome distribution given a set of climates (temperature, precipitation, cloud cover) and $CO_2$ conditions

**Table 5 | Fire Size and Fire Intensity under modern-day, RCP26 and RCP60 end-of-century conditions using a 25th percentile threshold, a 50th percentile threshold and a 75th percentile threshold**

| Ignition threshold | Fire size | Fire intensity | Unburnt (%) |
|---|---|---|---|
| 25th quantile | | | |
| Modern | 2.11 [1.98–2.24] | 24.91 [24.02–25.77] | 18.79 [18.18–19.53] |
| RCP26 | 2.81 [2.66–2.97] | 24.69 [23.90–25.50] | 9.60 [9.35–9.87] |
| RCP60 | 2.74 [2.59–2.92] | 24.08 [23.31–24.87] | 6.27 [6.09–6.43] |
| 50th quantile | | | |
| Modern | 1.59 [1.44–1.74] | 20.85 [19.97–21.80] | 31.65 [30.36–32.98] |
| RCP26 | 2.68 [2.51–2.86] | 22.68 [21.88–23.53] | 13.04 [12.43–13.66] |
| RCP60 | 2.62 [2.45–2.78] | 22.23 [21.41–23.06] | 8.01 [7.67–8.42] |
| 75th quantile | | | |
| Modern | 0 [0–0.44] | 0 [0–13.61] | 50.64 [49.37–51.84] |
| RCP26 | 1.89 [1.69–2.13] | 18.20 [17.22–19.20] | 25.44 [24.01–26.99] |
| RCP60 | 2.13 [1.95–2.32] | 18.41 [17.55–19.29] | 15.95 [14.73–17.31] |

and outputs biome classification, fraction of C4 photosynthesis and leaf area index (LAI). We obtained the temperature and pressure inputs from the climate inputs (described above). We obtained cloud cover for the future climate scenarios from shortwave radiation. We fitted a linear model between mean monthly global shortwave radiation (from 2010 to 2015) and mean monthly global fractional cloud cover (from 2010 to 2015) and then multiplied the future monthly shortwave radiation from all four climate models by the slope of the fitted model and adding the intercept.

We derived mean fractional tree, shrub, and grass cover for each of these experiments using the mean values of tree, grass and shrub cover from ESA CCI (Climate Change Initiative) Land Cover[104] for each biome simulated by BIOME4 under modern-day conditions and applying that value to the same biome simulated by BIOME4 for each experiment. Using variable tree, shrub and grass cover was considered; however, the results of including this variability did not fundamentally change the results, whilst introducing additional compounding uncertainty. To calculate grid-level tree, shrub and grass cover, the fraction of each NPP for each PFT to the total simulated NPP per grid cell was used. PFTs were then aggregated to tree, shrub and grass categories. A further rescaling was performed to account for bias in the NPP calculation for tree species using ESA CCI Land Cover. These results are presented in the Supplementary. We also used the biome classification to examine the characteristics of burning within each biome between different experiments.

The first step for obtaining the GPP was to run the BIOME4 model to obtain LAI for each experiment. We then obtained fAPAR (fraction of absorbed photosynthetically active radiation) using the Beer-Lambert law:

$$fAPAR = 1 - e^{(-k.LAI)} \quad (1)$$

where $k \approx 0.5$, a constant extinction coefficient[105] BIOME4 tends to overestimate fAPAR when compared to the NASA/GIMMS3g values, which were used to calculate GPP in the original GLM models. We rescaled fAPAR for each experiment by the ratio between the monthly NASA/GIMMS3g fAPAR averaged from 2010 to 2015 and the simulated BIOME fAPAR for the same period, as was done in Haas et al.[24]. The second step was to compute global monthly $C_3$ and $C_4$ photosynthesis using climate inputs (temperature, VPD, photosynthetic photon flux density: PPFD, and atmospheric pressure) as input to the P Model[50,103]. We derived PPFD from shortwave radiation:

$$PPFD = 60*60*24*10 - 6*kEC*rsds \quad (2)$$

where $= 2.04 \, \mu mol \, J^{-1}$[106]. We used BIOME4-derived fAPAR and $CO_2$ levels appropriate to each experiment. Total GPP was then calculated as follows:

$$GPP_{monthly} = GPP_{c3}(1 - C4_{fraction}) + GPP_{c3}C4_{fraction} \quad (3)$$

with $GPP_{C3}$ and $GPP_{C4}$ representing monthly $C3$ and $C4$ GPP values from the P model and $C4_{fraction}$ representing the fractional C4 cover from BIOME4.

**Human activity inputs**

Human activity is represented in the GLM models by three input parameters: population density, cropland density, and road density. To minimise uncertainty related to changes in human activity and to account for potential biases with the models, we use the same anomaly approach as with the climate variables. In this case, however, we compute the anomalies between the modern-day-period population density, cropland cover and road density from all our input data and then add the anomalies to the original modern-day data used to build the GLM models. This allowed the differences between the input datasets to be accounted for and ensured that the change reported was due to changes in the input variables and not due to differences in model assumptions.

Gridded annual global population data at 0.5° × 0.5° resolution used in the ISIMIP2b protocol and based on the national SSP2 population projections for 2010 to 2015 and 2090 to 2100 was obtained from the ISIMIP data repository: https://data.isimip.org/. These data were averaged to obtain the mean gridded population density for the modern period and the end-of-the-century period; their anomaly was calculated and added to the original GLM population density input

Gridded global cropland fraction at 0.5° × 0.5° resolution was obtained from the ISIMIP data repository: https://data.isimip.org/. The ISIMIP2b protocol used transient land-use patterns based on patterns generated by the land-use model MAgPIE according to the SSP2 scenario of economic development and population growth. This model also accounts for climate and atmospheric $CO_2$ fertilisation effects, forced with data associated with RCP2.6 and RCP6.0, respectively. We downloaded outputs for 2010 to 2015 and 2090 to 2100 from MagPIE/REMIND based on IPSL-CM4A-LR, MagPIE/REMIND based on GFDL-ESM2M, MagPIE/REMIND based on MIROC5, and MagPIE/REMIND based on HadGEM2-ES. These outputs were averaged to obtain gridded population density for the modern period and the

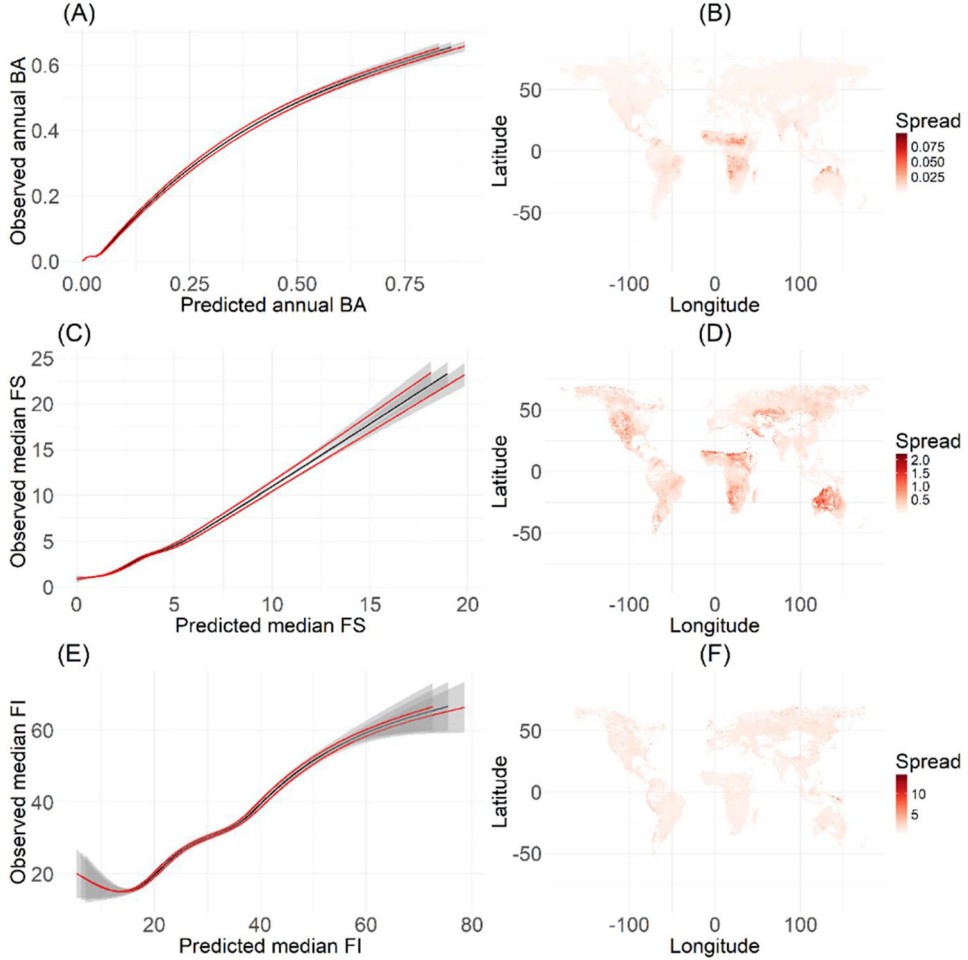

**Fig. 5 | Model performance.** Partial residual relationships between predicted vs observed fire properties (in black) and the associated 95% confidence interval (in red) for **A** burnt area (BA), **C** fire size (FS) and **E** fire intensity (FI). The spatial patterns of the spread of the confidence interval (range between upper and lower bounds of the 95% confidence intervals) are shown for **B** BA, **D** FS and **F** FI. These plots show that the model uncertainties are small over most of the observed range of each fire property.

end-of-the-century period. Their anomaly was calculated and added to the original GLM population density input

Gridded future road density was derived using the method discussed in Meijer et al.[107] and using the predictors identified by the Global Roads Inventory Project, specifically gridded population density and gridded GDP. Gridded GDP data, obtained from downscaling country-level estimates using ISIMIP2b gridded population density, for 2010–2015 and 2090–2100, were downloaded from the ISMIP repository at 0.5° × 0.5° resolution, and their anomaly was calculated and added to the original GLM population density input. Total GDP per grid cell was divided by population density data to obtain GDP per capita. The yearly data were averaged to provide mean values for each time period.

A linear model developed using the population density for 2010–2015, the GDP for 2010–2015, OECD membership (yes/no)[107] and total area (in km$^2$) (obtained with the area function in the R raster package) was used to simulate the spatial pattern of road density[24]. The model performs less well than the country-level model developed from the GRIP dataset (ref. [107]; adjusted-$R^2$ = 0.90; ref. [24]; adjusted-$R^2$ = 0.47), but this is to be expected. It has a good correlation with the GRIP dataset ($r$ = 0.68). We use this model to predict road density under the SSP2 scenario, using the population density and GDP per capita data for the mean 2090–2100 period (see ref. [24] for full model description).

### Lightning and topographic inputs
Lightning ignitions were kept constant at the 2010–2015 WGLC WWLLN background rate[108] in all experiments, following the ISIMIP2b protocol in which lightning is prescribed.
Topographic position index (TPI) and vector ruggedness measure (VRM) were held constant.

### Uncertainty associated with BA, FS and FI GLM models
To assess the uncertainties associated with the response of fire to climate, vegetation and human activity, we calculated the 95% predictive confidence intervals for the BA, FS and FI models separately by obtaining the upper and lower interval values by calculating plus and minus the fitted value multiplied by 1.96 times the standard error (see Fig. 5). We also examined the spatial patterns of these uncertainties (distance between upper and lower bounds of the 95% confidence interval) to determine whether there were spatial biases in the predictions. These analyses show that the uncertainty both along environmental gradients and in terms of spatial distribution (Fig. S3) is relatively small.

### Data availability
The processed data and the generated data in this study have been deposited in the repository: https://figshare.com/account/home#/projects/131081. The climate and socioeconomic data for the future

scenarios used in this study are available at the ISIMIP repository: https://data.isimip.org/search/simulation_round/ISIMIP2b/product/InputData/climate_forcing/gfdl-esm2m/subcategory/atmosphere/at

## Code availability

The description of how the GLM models were constructed can be found in Haas et al.[23]. BIOME4 code is available at: https://pmip2.lsce.ipsl.fr/synth/biome4.shtml. The P model description and code are available at https://pyrealm.readthedocs.io/en/latest/users/pmodel/module_overview.html. The code is available at the following DOIs: Haas et al.[24]. results scripts. figshare. Software. https://doi.org/10.6084/m9.figshare.24764571.v4; Haas et al.[24]. run models. figshare. Dataset. https://doi.org/10.6084/m9.figshare.24764577.v2; Haas et al.[24]. set up the models. figshare. Dataset. https://doi.org/10.6084/m9.figshare.24764583.v2; Haas et al.[24]. final data. figshare. Dataset. https://doi.org/10.6084/m9.figshare.24764589.v1.

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

## Acknowledgements

This work is a contribution to the LEMONTREE (Land Ecosystem Models based On New Theory, obseRvations and ExperimEnts) project, funded through the generosity of Eric and Wendy Schmidt by recommendation of the Schmidt Futures programme (S.P.H., I.C.P.) and to the HORIZON-MSCA FIRE-ADAPT (The Role of Integrated Fire Management on Climate Change Adaptation for Ecosystem Services in Tropical and Subtropical Regions) programme (S.P.H.). Thanks also go to the members of the Leverhulme Wildfire Centre and to David Orme for help with BIOME4. The authors acknowledge funding from NERC Centre for Doctoral Training in Quantitative and Modelling skills in Ecology and Evolution (NE/S007415/1) (O.H.) and Leverhulme Centre for Wildfires, Environment and Society (RC-2018-023) (O.H., I.C.P., S.P.H.). And European Research Council, Re-inventing Ecosystem And Land-surface Models (787203) (I.C.P.).

## Author contributions

Conceptualisation: O.H., S.P.H., I.C.P. Investigation: O.H., S.P.H., I.C.P. Methodology: S.P.H., I.C.P., O.H. Data curation and analysis: O.H. Funding acquisition: I.C.P., S.P.H. Supervision: I.C.P., S.P.H. Writing—original draft: O.H. Writing—review and editing: O.H., S.P.H., I.C.P.

## Competing interests

The authors declare no competing interests.
