## [Transparent Peer Review file · Nature Communications]

Wildfires on a changing planet

Corresponding Author: Ms Olivia Haas

Version 0:

Reviewer comments:

Reviewer #1

(Remarks to the Author)

The study by Haas et al. applied simple Generalized Linear Models (GLMs) to future climate projections for the end of the 21st century. Even under high mitigation efforts, these projections show a substantial shift in future fire regimes. Most parts of the world are expected to experience increases in burnt area, fire size, and intensity. However, tropical areas are projected to see a reduction in burnt areas, particularly in Africa and the Indian subcontinent, and stronger under high mitigation. The study also identified the main drivers of these changes.

The study marks a refreshing departure from just looking at projections in burnt area by including multiple fire regime metrics. Analysing mitigation potential for different fire measures and how it varies across different biomes also provides valuable information, and determining the driving causes of future fire regime change is also a significant step towards making fire research relevant to the real world. The manuscript is well-written and easy to follow, which is commendable considering the complexity of the analysis. As a dyslexic reviewer, I appreciated the accessibility of the language in the paper.

I do have some major concerns with the modelling setup and experimental design. Many modelling choices require better explanation, stronger justifications, or new analysis. There are also several instances of misinterpretation or miscomparison with previous studies on future burnt areas that the authors need to address thoroughly. I've highlighted the main ones I noticed in bold in the specific comments. Responses and changes to the m/s, and possible revised analysis, will need re-reviewing before I can recommend publication. However, if the authors are able to address these concerns in the next round of reviews, I foresee this paper making a really important and novel contribution to how we perform future fire projections.

****Major comments****

Future driving data

The newer ISIMIP3b climate inputs have been available for some time now and are starting to be used in studies projecting future wildfire activity (e.g., Jones et al. 2024). ISIMIP2b uses 4 bias-corrected GCMs over 2 RCPs, transitioning from historical to future in 2006. On the other hand, ISIMIP3b uses 5 GCMs, allowing sampling of a broader range of uncertainty in climate response to forcing, more SSPs, a longer bias-correction period and a historical/future transition at 2014. Would it be worth using ISIMIP3b instead? The justification for using ISIMIP2b outlined in lines 612-614 is that using ISIMIP2b allows for comparison vs UNEPs projections. This, however, seems rather weak, as there is no direct comparison to the projections UNEP reported. There may be other justifications. For example, ISIMIP2b is still being used in some studies exploring future fire due to the current absence of corresponding future land use and population density, which might also be a factor here - though it appears that the study uses one land use projection for both high and low mitigation scenarios, which *might* be useable in conjunction with ISIMIP3b.

Jones et al. State of Wildfires 2023–24, Earth Syst. Sci. Data Discuss, <https://doi.org/10.5194/essd-2024-218>, in review, 2024.

Historical comparison

I may have misinterpreted the model set-up, but was the difference calculated against historical isimip2b 2010-2015 or WFDE5 2010-2015? The text in the methods and numbers in Table S4 implied that you are comparing against WFDE5 2010-2015.

ISIMIP2b bias correcting preserves GCM trends and variability, which means that there is still some important remaining differences in key fire-related variables even for the present day (Mathison et al. 2023). So when calculating future anomalies vs present day, you do need to calculate the difference compared to individual isimip2b model 2010-2015 fire simulations.

Why 2010-2015 for historical? Should you use 10 years (i.e. 2010-2020 or 2005-2015) to match the same period as the future simulation?

Mathison et al. Description and evaluation of the JULES-ES set-up for ISIMIP2b, *Geosci. Model Dev.*, 16, 4249–4264, <https://doi.org/10.5194/gmd-16-4249-2023>, 2023.

Uncertainty quantification/confidence intervals

The uncertainty ranges seem to be based on the different GCMs' responses, thereby sampling the uncertainty in forcing (via high and low mitigation) and the uncertainty in climate response (via multiple GCMs). Is that correct?

Fire response to climate and land use forcing is also a substantial part of uncertainty, and should, if possible, be considered. GLMs come with a number of ways to quantify uncertainty (confidence/predictive intervals, error, residuals, bootstrapping techniques, Bayesian Inference etc) many of which are relatively easy to implement. It would be nice to also have some level of confidence/uncertainty quantification of modelled fire responses.

** Specific comments **

L30-31: ref 16 (UNEP 2022) seems like an obvious reference for this point as well.

L39: Warmers and dry conditions impact on fuel dynamics may be more complicated than the simple “reduced fuel load and fuel continuity” listed here (i.e. Different vegetation composition, fuel types and flammability, delayed decomposition, vegetation and fuel height structure, etc). Maybe rephrase to point out that the relationship is complex. Though the point is valid – warmer & drier don't necessarily mean more fire activity.

L42-44: As far as I'm aware, no process-based global fire model study has used ISIMIP2b data to perform future projections in burnt area, intensity or fire size (though there might be individual model studies I don't know about). Ref 16, though, definitely does not use a process-based fire model. UNEP (2022) actually uses a very similar experimental design presented here: an empirical model trained on observed burnt area and driven by ISMIP2b climate data, and using a vegetation model to provide maps of potential vegetation cover as proxy for fuel. It is still worth including this as past work, and this study still has plenty of weaknesses (i.e. only looking at changes in very high burnt areas, not looking at intensity and fire size, etc), which makes your analysis still very exciting. But its worth making sure the description of this (and other previous studies) is accurate.

L44-46: the point that process-based models haven't been shown to have the agreement or skill to reproduce trends and therefore future projections is definitely valid. However, it is worth noting that these studies cited here are now using fire models from 10 years ago, and newer models when combined with observations and advanced statistical techniques are starting to constrain around long term trends (i.e Burton & Lampe et al. 2023).

Burton C, Lampe S, et al. Global burned area increasingly explained by climate change. Research Square, preprint <https://doi.org/10.2023/Dec/14;21203>.

Line 59: As ISIMIP2b uses CMIP5 models, I don't think “state-of-the-art” applies here.

Line 78: Do you need to refer to Figure S3 here for an individual fire property breakdown?

Line 94-96 (looking at the number of gridcells where fire's occurred). Is this a relevant or useful measure? For one thing, gridcell areas are, by their nature, somewhat arbitrary and the number would change depending on the resolution. Also, the GLMs have been trained on observations that are known to miss some unobservable fires (even GFED5 won't capture all burning in e.g. closed campy forests). And because of the lack of information of grid cell heterogeneity, these simulations don't say if the same locations of these grid cells are burning each year or if fires can be randomly distributed. Also, it seems like it is not a very important point anyway. Unless I've misinterpreted, I would take it out.

Line 133-134: Does the model capture the possible transition to less but faster drying fuel seen in some grassier ecosystems? If not, would this impact your results?

Line 139: “increased drying and GPP” is this due to CO2 effects or something else?

Line 161-162: Is this first sentence in reference to Burnt Area?

Line 206/207: ref 16 tends to show increasing fire activity over the southern Amazon and Cerrado, in line with this studies BA and FS.

Line 209: ref 16 and (I think) 30 shows increasing fire activity at northern high latitudes.

Line 211: ref 16 does not report changes in the total global burnt area under high mitigation (it instead reports changes in probability in the occurrence of locally unusually high burnt areas). The higher mitigation in ref 29 and 30 appears to be B1 used in CMIP3, which is roughly the equivalent of RCP4.5 and, therefore, does not have the same level of mitigation as RCP2.6 tested here. They also use an older generation of climate models, which could also explain the disagreement. And I'm not sure how to ascertain a sensible trend from ref 33, though if anything, it looks like RCP2.6 shows increases in fire activity.

Line 220: Are the different GLMs connected somehow, i.e. using some over-arching hierarchical sampling of parameters? Or are they entirely independent (besides using similar driving data?) If they are not formally connected, is it legitimate to test and comment on the breakdown of fire clusters? These could be dependent on feedbacks (including indirect feedbacks) between fire regime aspects that are not represented in the model.

On a similar point...

Line 246-250: Empirical models, such as this, lack fire-fuel feedbacks. Woody encroachment under reduced burning mentioned here, could effect fire regimes, but also the removal of fuel or shift in ecosystem composition from fire will not be represented. Some discussion on how this might impact your results, and as a source of possible disagreement with previous work, would be great somewhere.

Line 577: Can you supply a version-controlled repository or doi with the model code/configuration rather than just a link to the current code?

Line 609: See comment on lines 94-96: do you need this threshold? And if so, why 0.0011?

Line 666-671: This sounds like a generally a good experimental design to produce vegetation/fuel information into the future. But it is a single model approach and therefore wont capture uncertainty in responses of fuel load to future climate. Some discussion on how using that single realisation might affect your uncertainty analysis would be interesting.

Line 678-683: Similarly, using that same mean fractional covers across biomes might wash out a lot of the interesting within-biome responses. A brief discussion on how that impacts your results would be great.

Line 704: A similar point – I like the setup here for human activity. I don't think we're at a point yet where we can test the uncertainty in future human activity that leads to similar climate pathways, so following the ISIMIP protocol for these variables makes pragmatic sense. But somewhere (probably in the main) some acknowledgment that this will have an impact on your projections would be useful.

Supplementary Info

Line 777: I don't see any quantification of the difference/agreement in trends between the model and observations. This is important because one of the few ways to infer if a model might capture future out of sample projections from present day observations is if they are getting the correct present-day trend (I would trust this over NME comparisons), and it's hard to see if there is agreement in overall trend across the time period in Figures S1 and S2.

Line 778: This is the first round of fireMIP? It might be worth noting these fire models are now quite old.
Fig S1, S2: Can you use a more accessible colour scheme than the green-red combination?

Just to reiterate: this is a very interesting study. Once these points are addressed, I look forward to it eventually being published.

Best of luck with the rest of the reviews

(note my previous collaborations with some of the co-authors have been checked and cleared by the editorial team)

(Remarks on code availability)

The link to the code appears not to work I think

Reviewer #2

(Remarks to the Author)

I commend the authors on their manuscript assessing future potential global wildfire activity. Though I believe this manuscript would be a strong addition to nature communications, I believe the article would be strengthened with a discussion of regional models that can more intimately address additional mitigation policies, which include both land and fire management. In the attached manuscript you will find detailed comments. Below are some previous research that should be considered in the discussion.

Ager, Alan A.; Barros, Ana M. G.; Houtman, Rachel; Seli, Rob; Day, Michelle A. 2020. Modelling the effect of accelerated forest management on long-term wildfire activity. *Ecological Modelling*. 421: 108962.

Barros, Ana M. G.; Ager, Alan A.; Day, Michelle A.; Krawchuk, Meg A.; Spies, Thomas A. 2018. Wildfires managed for restoration enhance ecological resilience. *Ecosphere*. 9(3): e02161.

Dye, Alex W., et al. "Simulated future shifts in wildfire regimes in moist forests of Pacific Northwest, USA." *Journal of Geophysical Research: Biogeosciences* 129.2 (2024): e2023JG007722.

Prichard, Susan J.; Salter, R. Brion; Hessburg, Paul F.; Povak, Nicholas A.; Gray, Robert W. 2023. The REBURN model: simulating system-level forest succession and wildfire dynamics. *Fire Ecology* 19:38.

Young, Jesse D.; Ager, Alan A.; Thode, Andrea E. 2022. Using wildfire as a management strategy to restore resiliency to ponderosa pine forests in the southwestern United States. *Ecosphere*. 13: e4040.

Young, Jesse D.; Ager, Alan A. 2024. Resource objective wildfire leveraged to restore old growth forest structure while stabilizing carbon stocks in the southwestern United States. *Ecological Modelling*. 488: 110573.

(Remarks on code availability)

Reviewer #3

(Remarks to the Author)

This manuscript projects changes in key aspects of fire regimes under alternative climate futures. The paper extends the authors' earlier empirical models of satellite observed burned area, fire size, and fire intensity, and projects how these important metrics of fire are likely to change under several alternative future scenarios. The topic is significant and of high importance, but I am afraid I do find myself with reservations over some of the results.

Major comments

I find myself questioning the validity of some of the findings, stemming primarily from reservations over the original models (<https://iopscience.iop.org/article/10.1088/1748-9326/ac6a69>) on which this paper is based. I recognise that paper is already published, but I need to evaluate the validity of those models in order to evaluate the validity of this paper. Those models didn't seem to include any interactions between explanatory variables, and the models assumed linear relationships between explanatory variables and fire metrics. Both decisions, in my opinion, are not justified, because interactive effects (e.g., dryness x fuel load) and non-linearities (e.g., hump-shaped effect of productivity) are likely to occur. The intermediate

productivity hypothesis, for example, explains that fire activity is most common at intermediate productivity, but your GLMs assume a linear relationship, resulting in the effect that environments with extremely low productivity have the highest predicted fire intensities. This results, in my opinion, with quite questionable and unrealistic predictions (especially of fire intensity) in polar regions with low productivity. Further, there are some quite large extrapolations, with fire intensity being predicted in areas that haven't experienced fire in the satellite record. These effects can be seen in Fig 3, bottom panel, of the original paper, in which the highest fire intensities are predicted for the high arctic tundra, far southern South America, and a large section of the Tibetan plateau, all of which are regions with very little to no observed fire. Because I find those findings implausible, I am afraid I also question the validity of the findings in this current manuscript because it is based on those models. Further, the models don't really account for the fact that different vegetation types burn differently (e.g. boreal forest burns differently than temperate broadleaf forests, even if productivity is similar). I welcome the authors to correct me if I am wrong.

A second major concern is that the projections do not adequately consider uncertainty. In Fig 1, for example, there is no category for "unchanged", and relatedly, there is no acknowledgement of uncertainty. It seems there has been no attempt to quantify the confidence associated with a given change. It's possible many of these changes are slightly positive or negative with large confidence intervals spanning "no change".

Line-by-line comments

L 1. Minor point, but perhaps consider tweaking the title, because "global wildfires" reads as if there are wildfires that are global, whereas the study was global but the fires aren't.

L37. And there is recent empirical evidence (not just projections) that indicate very intense fires are increasing <https://doi.org/10.1038/s41559-024-02452-2>

L38. Not clear from intro how your approach improves on this pitfall of not taking "account of the potential for changes in vegetation". More info needed in the intro on how your approach ensures a relevant veg state in the future, as well as relevant road and population density. (given the method-at-the-end format, I think there is a need to provide a bit more methodological info up front so a reader can understand the major steps as we read through)

L49-51: I think some more explanation would help here. I went and read your earlier work (ref 19) and understand those models, but some more text may help to make what you did in that work clear in this paper, so that it's not necessary for a reader to have to read that paper to understand this one. I suggest adding that the "models" themselves were generalised linear models, and explain the response variables more precisely, including temporal scale and units. As far as I gather from the other paper, the models were of summarised versions of the fire data as opposed to individual fires/hotspots, I believe mean burnt area per month, median fire size per month, and median fire intensity.

Table 1. Can the right-hand column be turned into multiple columns, so that there's a column for climate, CO2, popdensity etc, with a few plain language words describing the various experiment components? Can these "experiments" be explained in a general way in the text; it's not clear in the main body of the paper whether the future scenarios, for example, also include changes to human population density/road density, and how vegetation is modelled in the future.

Fig 1. It's quite hard to interrogate the finer-scale patterns in the fig because of size and resolution, e.g. I found it impossible to evaluate the area where I live. I know the fig will be higher res in final version, but I would advocate for a larger figure with the maps stacked vertically, taking up the full width. You only have 3 fairly small figs as is, so there should be plenty of space.

L 89. Are the "modern conditions" the empirical obs, or are they the model predictions for the modern conditions? Both are important, because if the model is projecting incorrectly to the present day, then some of the differences reported in this para could be due to that.

Table 2. I find it quite surprising that the current burned area lies in between two climate scenarios that are both warmer (2 and 3-4 deg); i.e., 2 degrees of warming (high mitigation) is predicted to cause a ~15% reduction in BA, while 3-4 degrees of warming (low mitigation) will lead to a 50% increase in BA. How do scenarios that both involve warming lead to such divergent patterns from the current day?

Fig 2. It's hard to compare the same biome class in the side-by-side panels. I think it would be easier to do that if the two sub plots were stacked vertically, using the same axis ranges, so that it's visually easier to line up the bars.

L128-157: these paragraphs are dense slabs of text. Can you break them up a bit?

L173-174: how reliable is a finding like this? This implies that the vegetation is currently not burning solely because of a lack of human ignitions; that doesn't seem plausible to me.

Fig 3. For this to be understandable, the acronyms of the predictors need to be described in the main text, or here in the figure caption - e.g. what is "DD (s)". Also, explain in the captions how the values for each of the slices were determined.

Somewhere in Discussion, I think it would be good to acknowledge that these results reflect predictions of the mean/median fire. But many, if not most, fires on earth are human orchestrated. So, this focus on the central tendency may overlook how the extremes are changing, which are arguably quite important (raised by <https://doi.org/10.1038/s41559-024-02452-2>)

L257-258. I find it quite surprising, if not unlikely, that high mitigation rather than low mitigation will lead to the most intense fires. [Same for “though less destructive” on L263]. Why is this the case?

L577: this URL for data and code doesn't work for me; need to provide the DOI version, not the link with you logged in because it takes me to the projects page.

L585-586. Same comment as last one.

L589. A diagram of the workflow of the whole project would be helpful.

L607: I don't understand the ignition threshold; more detail needed, as well as rationale for how it solves the problem.

L608: I don't understand why you convert $BA < 0.0011$ to 0, when they are both valid numbers for BA.

L628-636: I find this quite hard to follow and think it needs to be explained more clearly. I was under the impression the GLMs were already fitted, and that you were projecting them to future conditions. Here it gives the impression you are refitting them, sequentially excluding predictors. I don't get why you would refit the GLMs, or how this actually helps determine which variables are driving increases and decreases. Couldn't the partial effects of the variables show that?

Several times in the main body, it refers to “see Supplementary Materials” or similar. Please be specific, because the supplement is quite long.

(Remarks on code availability)

The manuscript lists URLs for accessing the code, but these links did not work for me. I believe the authors may have posted the link from their login page rather than the publicly accessible DOI. That means I cannot evaluate the data/code.

Version 1:

Reviewer comments:

Reviewer #1

(Remarks to the Author)

Thank you for your detailed responses to the first-round review. The manuscript has improved, and I appreciate the additional clarifications and methodological adjustments. However, I still have some major concerns regarding the modelling framework's ability to reproduce observed changes in burnt area, as well as a few instances where the results may still be somewhat overstated.

Anything I haven't commented on from the previous round I am happy to consider as fully addressed.

Major Concern: Trend Evaluation

One of the central themes of the manuscript is the projection of future fire trends. In several parts of the discussion, future changes are linked to the current global declining trend in burnt area (e.g., “The current global trend of decreasing burnt area may continue under high-mitigation efforts but is unlikely to continue under low-mitigation efforts where changes in climate promoting wildfires overwhelm the effects of human activities in reducing wildfires, particularly in the tropics”). However, the current evaluation does not demonstrate that the modelling framework adequately captures global or regional trends in burnt area when compared with observations. The addition of the trend analysis is a positive step forward, but it reveals a key weakness in the framework: the model does not capture the global decline nor regional trends in burnt area. While I appreciate the authors' acknowledgment of this issue, the explanation provided does not sufficiently address the underlying causes. Specifically:

- As stated, the GLMs were designed to capture spatial patterns of fire regimes. However, since they are now being used to infer changes over time – which the model has not necessarily been optimised for, it is crucial to evaluate their ability to reproduce long-term trends. While I am less concerned about the global results, the model also seems to struggle to reproduce trends in burnt area in the regional breakdown.
- The rebuttal points out that GFED5 shows a significant decrease in burnt area and GFED4s shows a non-significant decrease, and uses this to infer observational disagreement. This does not constitute a valid statistical argument or evidence of observational disagreement. Rather, it highlights a change in the confidence of the trend, which is not surprising given the updated nature of GFED5. Additionally, it is important to consider that over a longer time period than the 10 years considered here, studies have shown that GFED4s does indeed indicate a significant decline in burnt area. Moreover, there are trend benchmarking metrics that can account for uncertainty in observed trends, should the authors feel this is an important consideration.
- The argument that the modelling framework reproduces changes on paleo timescales is useful, but it would be more impactful as an additional line of evidence. It would be more compelling if the model first demonstrated that it could reproduce contemporary changes.

Given the emphasis on projecting future fire activity, a more rigorous assessment of why the model fails to reproduce historical trends is necessary before publication. Without a more thorough investigation, there is a risk that the modelling framework may not support the conclusions regarding future trends. The authors should conduct a more detailed evaluation of the divergence between observed and simulated trends, explaining where the disagreement originates—e.g., whether

observed decreases in trends in some regions are influenced by land use effects, which, while the model should capture, are less relevant for this study which primarily focuses on climate-driven fire trends. Evaluation of the changes in fire size and intensity, more detailed regional break down to show where observation and model may disagree, etc, may all help too. Additionally, a clearer discussion of the limitations of the modelling approach will probably be needed to ensure that future projections are better substantiated.

For evaluating model trends in the context of high observational trend uncertainty, I recommend consulting Burton et al.'s (2024) supplement, which includes methods for this purpose. While I was involved in that paper (and I may have suggested it to many of my own papers already), I suggest it as a starting point rather than something that must be cited. There may well be more straightforward benchmarking techniques. Additionally, Spuler et al. (2024) might also provide useful insights.

Burton, C.A., Kelley, D.I., Burke, E. et al. Fire weakens land carbon sinks before 1.5 °C. *Nat. Geosci.* 17, 1108–1114 (2024). <https://doi.org/10.1038/s41561-024-01554-7>

Spuler, F. R., Wessel, J. B., Comyn-Platt, E., Varndell, J., & Cagnazzo, C. (2024). *ibicus*: a new open-source Python package and comprehensive interface for statistical bias adjustment and evaluation in climate modelling (v1. 0.1). *Geoscientific Model Development*, 17(3), 1249-1269.

Other Key Comments

1. Use of ISIMIP2b Instead of ISIMIP3b.

The justification for using ISIMIP2b over ISIMIP3b looks great. One potential addition (if the authors find it useful) could be a brief discussion on what improvements in ISIMIP3b would make it more suitable for future studies. For example, would additional land-use scenarios or longer bias-corrected periods enhance its utility? Though if you feel this is a distraction, then you don't have to include it.

2. Uncertainty Representation.

The addition of uncertainty bounds in Supplementary Figure 3 is useful, though it appears that you are calculating a confidence rather than a predictive interval? Which is fine, just that the terminology may need reviewing.

While BA and FS seem to show a relatively narrow uncertainty range, the uncertainty for FI appears to be larger. It would be helpful to quantify the implications for your results rather than visually assess the uncertainty. Could you use these uncertainty estimates to indicate statistical significance—perhaps by masking non-significant trends in Figure 2 or adding error bars to Figure 3?

The authors acknowledge the limitations of plant functional type classification in their rebuttal. A short addition to the main text or methods would be helpful in clarifying this.

It would be helpful to explicitly reference Supplementary Figures 6 and 7 in the main text, with a brief mention of potential limitations in biome-specific confidence levels.

3. Decoupling of Fire Property Clusters.

I remain concerned about the phrasing regarding the decoupling of fire clusters in the future. The current wording suggests that burnt area, fire size, and intensity are presently observed as clusters but may decouple over time. This framing is problematic because it implies an existing interdependence that will break down, whereas the models are explicitly independent by design. If the models are independent, referring to a "decoupling" of clusters could mislead readers into thinking that fire properties are currently linked in a way that is not reflected in the modeling framework.

While all three fire properties share an ignition threshold, this does not create a structured interdependence between them regarding their response to environmental drivers. Instead, the results indicate that the relationships between these properties may evolve in the future due to their differing sensitivities to changing climate, land use, and vegetation conditions. A clearer way to convey this might be: "Our results suggest that the relationships between fire properties may change in the future, reflecting differing sensitivities to environmental drivers. However, these relationships are not modeled as interdependent feedbacks."

4. Threshold Selection for Ignition Probability.

The choice of the median value (0.0011) as the threshold for ignition seems reasonable, but could the authors provide a reference supporting this approach?

5. Comparison to fireMIP models.

The latest FireMIP round is available at ISIMIP (<https://data.isimip.org/10.48364/ISIMIP.446106>), with summary NME scores in Table S3 of Burton & Lampe et al (2024) (<https://www.nature.com/articles/s41558-024-02140-w>). I appreciate that these results weren't available at the time of initial submission. That said, the broader comparison between statistical models trained on observations performing better than process-based models without direct training has already been well covered across multiple studies from many research groups. Given this, it's not too surprising that a GLM approach outperforms older process-based models for specific tasks like modelling spatial patterns of BA.

What might be more relevant is to focus on the strengths of your approach relative to all other methods—whether process-based (like fireMIP), statistical (BASE, MaxEnt approaches etc), or hybrid (ConFire, decision tree/RF etc). Highlighting aspects beyond just burnt area, such as fire intensity or size, is a great example already in the revised m/s.

6. Re: "Empirical analyses have considered vegetation and human impacts on fire risk and are being used in the Inter-Sectoral Impact Model Intercomparison Project (ISIMIP) to assess potential future wildfire regimes under climate change (16, Jones et al., 2022)" I think reference to Jones et al. (2022) should be corrected to Jones et al. (2024).

<https://doi.org/10.5194/essd-16-3601-2024>

7. New Figure 1 is very helpful, but also feels like a supplementary figure.

Overall, the manuscript is shaping up well, and the responses to the first round of review have strengthened the study. However, the issue of trend evaluation remains a significant concern that needs to be addressed before the manuscript can be accepted.

I appreciate the authors' efforts and look forward to seeing the revised manuscript.

(Remarks on code availability)

I have only lightly reviewed the code to check the GLM model construction, which seems okay.

Reviewer #2

(Remarks to the Author)

My concerns have been addressed or discussed in detail that is satisfactory for publication.

(Remarks on code availability)

Reviewer #3

(Remarks to the Author)

The authors have generally done a good job responding to reviewer comments. I appreciate the consideration of multiple metrics of fire activity.

The main thing that I think could be improved relates to the complexity of the experiments: with multiple moving parts, it makes it challenging for readers to discern what drives the broad-scale results. A key issue lies in comparing present-day conditions with future scenarios, as these involve simultaneous changes in input variables beyond climate mitigation—such as population densities, land cover, and other factors. For example, present day global population is roughly 8 billion whereas SSP2 simulates roughly 9.5 billion I believe This complexity makes it difficult to intuitively understand or visualize the differences between present-day and future scenarios.

For example, the quite unintuitive prediction that a 2°C warming scenario leads to less fire than present-day (lines 100–102) might reflect changes in population or land use rather than the climate mitigation itself. This means comparisons between scenarios (e.g., present-day vs. mitigation) are not purely driven by changes in climate but also by shifts in human and land-use factors. It would be good to explain that when introducing that finding. Another example: in response to my earlier comment about the surprising prediction that high mitigation leads to more intense fires than low mitigation, you explained that this is due to fuel loading and shifts in productivity-dryness limitations.

I think this challenge would benefit from clearer visualization and explanation of how all of the drivers of fire differ under the scenarios (incl present day). For example, supplementary tables and maps showing how productivity, dryness, population size, land cover and other variables differ across scenarios would make it easier for readers to grasp these effects. I think this additional context would make it quite a lot easier to interpret the model results.

Clarifying “mitigation”: Since many readers will approach this paper as fire ecologists or practitioners, it’s crucial to explicitly clarify what “mitigation” refers to. In this paper, you generally and implicitly use it to mean climate mitigation, but for fire ecologists, the term typically implies vegetation management to mitigate fire risk. I recommend adding “climate” before “mitigation” throughout the manuscript to avoid confusion, especially since you’ve occasionally used “mitigation” to mean fire mitigation (e.g., L56).

In your response to my request for clear definition of the units of the response variables, you mentioned adding a table for response and input variables. However, the response variables still do not seem to be there and remain somewhat vaguely defined. It is essential to clearly define them, including their units, the time scale of measurement, and spatial resolution. Fire intensity seems to be undefined entirely. I know BA is the proportion burned per month, but at what resolution (i.e., proportion of what)? Each variable needs clear definitions, units, and temporal dimension, so that it’s crystal clear to reader what the models themselves were actually modelling.

L816–818: I now understand your rationale for selecting the median predicted BA value (0.0011) as a threshold to determine whether modelled fire should occur in grid cells where fire had not been empirically observed. However, selecting the median value as a threshold means that half of the grid cells without observed fire will have fire assigned, which seems problematic. In this case, 26% of grid cells had an observed BA of zero, meaning approximately 13% will have been assigned as having fire even though they didn’t, which represents a significant proportion. I suggest revisiting this threshold to ensure it aligns better with empirical observations. Rather than selecting the median value, you could use an approach that balances sensitivity and specificity (e.g., True Skill Statistic).

L43–44: “Physical indices such as measures of fire weather equally do not take account of the potential for human activities to mitigate wildfire risk.” Sure, but this work also does not evaluate fire mitigation.

L130: “this mitigation scenario” is unclear—specify which scenario you mean.

L137: Similarly, clarify what “this scenario” refers to, as the prior sentence mentions both scenarios.

L281: Please provide specific references to the 95% CIs.

L286–288: Consider dropping the sentence about overfitting. The argument is weak since machine learning models can be tuned to minimize overfitting, and the cited reference even points out that linear models can suffer from even worse specification bias.

L312: Spell out “PFT” at first use, as it may not be familiar to all readers.

Table 2 issue: There is no Table 2 in the main text (it goes from Table 1 to Table 3), but it is referenced in the text.

As I noted last time, at several points, the text says "see Supplementary Material". This is too vague because the Supplement is about 50 pages long. Please be specific as it took me considerable time to locate the relevant part each time.

(Remarks on code availability)

Version 2:

Reviewer comments:

Reviewer #1

(Remarks to the Author)

Apologies for the slow review. The request came in the middle of summer, and a combination of annual leave, fieldwork, and conferences delayed my response. Also, I've spent the past couple of weeks working through the bias correction methods. While I suspect there may be some issues with the mathematics, I wanted to be thorough because I don't want the authors to undertake another major round of revisions or discard the hard work they've already put in unnecessarily. Especially because, aside from the model evaluation for burned area, I'm very impressed with the revised manuscript, and the evaluation appears to be in - and possibly beyond - the right direction.

Main concern: Bias correction

1. Purpose of the test: The main objective seems to be to show that the non-human driven component of the model is plausible (or the converse, as expressed by the authors, “we tested the hypothesis that the model underestimates the influence of human activity”). I don't think there is an absolute need to carry out a full bias correction, particularly for future projections where the human contribution has already been heavily simplified and the difference between scenarios is essentially due to non-human components. However, if an appropriate correction is applied, it certainly wouldn't harm the findings. But...

2. Problems with the current correction: I think (though might be wrong) the correction as implemented essentially doubles the human trend and removes the none-human trend. If I'm right, this is not a reliable test if the none-human component is already correct, and it is not a defensible approach for future projections. I might have misunderstood how this works, so I've attached my derivation (for historic evaluation only) to check whether my interpretation matches what the authors implemented.

Recommendation: If I'm wrong, feel free to clarify the bias correction and how it evaluates the model under recent changes in burned area. Otherwise, I would remove it and take a simpler evaluation approach using your new sensitivity experiments. I think that simply showing that your non-human sensitivity run yields plausible results comparable to independent analysis, while the human component is very flat, is enough in itself (both globally and for your sub regions). But if you want to do something a bit clever, similar to the technique you've already started, I one suggestion is that you could obtain another point of evaluation by: assuming the non-human (climate/vegetation) component is correct, infer the implied human contribution to the trend directly from the full model, the non-human model, and the observations. The attachment has the math for this. The derivation is a bit long, and probably longer than necessary. But the actual comparison is very simple: essentially the difference between observed, modelled, and modelled-without-none-humans trends. This approach provides two clear evaluation points:

- Does the non-human component of the model behave plausibly? From the sensitivity runs.
- If the climate component is correct, what does the “corrected” human component look like, and is it consistent with independent evidence?

Both non-human and inferred human-driven trends could be compared to the standard literature already cited in the manuscript (e.g., Jones 2022; Burton & Lampe 2024; Andela 2017).

Either of these alternatives seem simple to implement and avoids the problematic assumptions of the current correction, while still addressing the underlying question: the plausibility of the model's none-human-driven trends.

After this, it's mostly a matter of tightening the discussion language to make it clear this is a modelled (albeit heavily trained) system and that some unknowns remain in the response to human and bioclimatic variables.

This feels like minor revisions, but I'd suggest to the editor a final quick check with a reviewer (doesn't have to be me) to ensure the literature comparisons are fully consistent.

Minor comments

- No changes needed here, but I really like the revised sentence:

“This result has implications for the distribution of fire regimes on Earth and implies that efforts to model future wildfire regimes should not assume that the relationships between different fire properties are fixed.”

This phrasing is much stronger for implications across future fire metrics.

- I may have missed it, but what trend evaluation (regression technique?) Did you perform to get your trend coefficients?

- The trend table S1 shows zero trend in other fire metrics for the model result (which I *think* correspond to the time series in Figure S1?). While the trends appear very small, I'm surprised it's exactly zero. Could you check? This might be an artefact of the regression technique (e.g., if outliers are downweighted), which is fine, but it would be good to note.
- In your sensitivity runs, I've assumed you hold variables that aren't varying to their climatological mean. Please just confirm that's correct and state it explicitly somewhere.
- Line 1167: these might be a small formatting problem here

That's all I could spot. As mentioned, the last part of the evaluation should be a quick fix, but could do with one last glance over. Otherwise, well done on the revisions, and good luck with Reviewer 3!

(Remarks on code availability)

Reviewer #3

(Remarks to the Author)

I am pleased to see a bit more explicit acknowledgement in the Results of the role of changes in human / socioeconomic variables in driving the predicted outcomes. However, I still believe there needs to be more effort on this point in other areas so that the results are not misinterpreted. For example:

- Abstract: says there will be generally reduced burning in the tropics. The way the abstract is worded, I think readers will interpret that as a result of changes in the climate scenarios, because there has been no mention yet that population and roads etc were also allowed to change in the predictions. The abstract would essentially lead a reader to think all of the changes are due to climate change and varying degrees of mitigation. Thus, in the Abstract, I think it should be explicitly stated that you compared two climate scenarios with X and X degrees warming, in which rising population and other socioeconomic variables were also projected, and that this particular effect of reduced burning in the tropics was strongly driven by those socioeconomic variables. At present, I think a reader would interpret the abstract as these effects all stem from climate mitigation differences.

- Introduction (last para): You explain that you compare two climate scenarios, involving 2 and 3-4 C warming. But the socioeconomic variables are just as important (even more so for some results), and they are only given a cursory mention. So, given their importance, I think it's absolutely imperative that you summarise those changes here too, e.g., under the middle of the road socioeconomic pathway in which human population and road density increase by X. Please explicitly state in the main text how much the population, or road density etc, is projected to increase under those scenarios.

- Opening paragraph of the Discussion: "under both high- and low-climate change mitigation scenarios. Both scenarios show a reduction in burning in the tropics but increased burning elsewhere." But it's not the climate mitigation that led to the predicted reduction, it's the simultaneous change in socioeconomics. I feel these sentences would lead a reader to believe the reduction in burning in the tropics is driven by the climate mitigation scenarios; please be explicit at this point that this effect was strongly driven by humans/roads.

- I think it may help to add 2 panels to Fig 2, in which human activity variables are held at present values, so a reader can easily discern the separate effects of climate change and climate mitigation. That way, it will be more explicable why we see such large reductions in fire in much of Africa, for example. I believe that is already shown in Supplementary Figure S20. I think this is quite important because it helps explain some of the less intuitive findings in the Results.

Significant (but easily rectified) concern:

Fig 4 and L896-911: I have concerns about the method used to identify the dominant driver of projected change in each grid cell. As I understand it, the authors re-fit the GLM multiple times, each time excluding one predictor, and then compare the resulting projections to those from the full model. This appears to be intended as a sensitivity analysis i.e., identifying which variable drives the largest change in predictions for a given cell. However, in its current form, this approach does not cleanly achieve that aim. Specifically: (1) Apples-to-oranges comparison: Comparing predictions from different models with different predictor sets does not constitute a controlled sensitivity analysis of those predictors. It introduces confounding due to changes in model structure. (2) Lack of marginal effect isolation: This approach does not isolate the marginal contribution of the excluded variable. Instead, the observed differences in predictions may arise from a combination of the variable's absence (globally from the model, not cell-wise), shifts in the coefficients of other predictors, or even differences in overall model performance.

This makes interpretation difficult and potentially misleading, as the observed changes could reflect changes in the model or instability rather than the true influence of the omitted variable. This is particularly problematic in the presence of variables with some degree of correlation (as is the case here), where excluding a predictor can redistribute explained variance among correlated variables. As a result, the spatial attribution of projected changes may reflect changes in the model rather than the independent contribution of each predictor.

My suggestion: I think that a more interpretable and statistically coherent alternative would be to retain the full GLM model and conduct a post hoc sensitivity analysis by perturbing one predictor at a time e.g., by permuting its values randomly, breaking its association with the response. This would allow the influence of each variable on future predictions to be assessed directly, without altering the fitted model structure, and would provide a clearer basis for spatial attribution of

change in projected fire risk.

Minor points:

L105: sentence is ambiguous because it could be interpreted that the global decrease in burnt area is primarily driven by decreases in tropical evergreen and deciduous forests themselves, or in the area burned in those veg types. Minor tweak to wording would help.

L106: "Increased road density in previously fire-free regions also caused increases in burnt area in Europe and the east coast of the United States across all four models". I find it *very* hard to believe that an increase in roads in eastern USA and northern Europe, where road density is already high, would be the dominant driver of such an effect. It implies that a lack of roads in these heavily populated regions is currently limiting burned area more than the climate. These regions currently have relatively little fire (compared to other parts of those continents) because of more benign climate conditions, not because of a lack of roads. I think this is a likely a spurious finding. Moreover, the fact that the finding held across the four climate models is no evidence of reliability, because (1) it could easily be an issue with the GLMs, (2) road density would have been held constant in each climate scenario, or (3) it could also be an issue with the iterative approach to assigning the most important variable.

L788: I raised in my first review that the Figshare URL for the code is not accessible and this has not been changes. The URL links to the private project (which is not accessible to others), whereas it is possible to provide a private link for reviewers. As a result, I have not looked at the code.

(Remarks on code availability)

Code is still unavailable because the authors have provided a link to their own FigShare project page, which is not accessible to others. They should provide a private link for reviewers, which is possible in Figshare.

Version 3:

Reviewer comments:

Reviewer #1

(Remarks to the Author)

The authors have now addressed all my comments, apart from a couple of very small points. I therefore recommend the paper be accepted, barring some tweaks outlined below that do not need to be re-reviewed:

1. On bias correction: even though you've chosen not to refine this here (which I'm fully happy with - you've now done plenty of model evaluation!), a correct implementation would make for an excellently nerdy follow-up.

2. There are a few instances, mainly in the new text, where "observed" is used for model results. You should instead use terms like "simulated", "modelled", or "reconstructed". (I always get picked up on that when my papers go to review, so I'll pass on the pain ;)).

3. In Supplementary Table 6, your response says: "The values are not exactly zero, however, they become zero when rounded up." Given that some of these zeros look significant, it would be better to use a different unit that avoids rounding to zero. e.g. ha or m² for fire size. I'm not sure what the small-number equivalent would be for W km⁻¹, but there should be a way to avoid zeros in significant results just due to rounding.

Finally, just to note that future projections for fire are always hard to get right, so any attempt to do this in a robust framework was inevitably going to invite significant, multiple rounds of review. I suspect this paper will encourage a lot of future studies moving away from burned area and towards more meaningful fire regime measures. Thanks for persevering with this and for going all out in your responses. It'll be an important paper for the rest of the community to test and build on, and I look forward to seeing what comes from it.

(Remarks on code availability)

I have reviewed the code before, but didn't this time.

Response to reviewers' comments for the manuscript “*Global wildfires on a changing planet*”

We thank all the reviewers for their comments on this manuscript. We would especially like to thank reviewer 1 for their constructive feedback. We have addressed the issues raised by all reviewer's point-by-point. Some issues arose multiple times; we have addressed these once, and subsequently refer back to them. We have also ensured that the data and code are available in a public link which should be accessible. Our responses are in **blue text** and the changes made to the manuscript in *blue italics*. New references have also been highlighted in the text.

Reviewer 1

“The newer ISIMIP3b climate inputs have been available for some time now and are starting to be used in studies projecting future wildfire activity (e.g., Jones et al. 2024). ISIMIP2b uses 4 bias-corrected GCMs over 2 RCPs, transitioning from historical to future in 2006. On the other hand, ISIMIP3b uses 5 GCMs, allowing sampling of a broader range of uncertainty in climate response to forcing, more SSPs, a longer bias-correction period and a historical/future transition at 2014. Would it be worth using ISIMIP3b instead? The justification for using ISIMIP2b outlined in lines 612-614 is that using ISIMIP2b allows for comparison vs UNEPs projections. This, however, seems rather weak, as there is no direct comparison to the projections UNEP reported. There may be other justifications. For example, ISIMIP2b is still being used in some studies exploring future fire due to the current absence of corresponding future land use and population density, which might also be a factor here - though it appears that the study uses one land use projection for both high and low mitigation scenarios, which **might** be useable in conjunction with ISIMIP3b.”

Whilst we agree that using ISIMIP3b would allow us to sample more uncertainty through using five GCMs instead of four and providing a longer bias-corrected period, these advantages are outweighed by those of staying with ISIMIP2b. In ISIMIP3b, the socio-economic and climate scenarios are more tightly coupled than in ISIMIP2b. We would therefore have to use either SSP126 climate and vegetation and SSP1 human activity or SSP370 climate and vegetation and SSP3 human activity. There is large uncertainty about how human activity influences wildfires (both in the present and the future) and considering multiple SSPs would complicate our analysis. In contrast, using a single SSP that corresponds to multiple climate mitigation scenarios – and could be held constant without any confounding effects in the climate and vegetation drivers – provides a more easily analysed experiment. The ISIMIP2b protocol was well suited, precisely because it only considers one SSP, which is decoupled from the RCP scenarios (Frieler et al., 2017). The future socio-economic SSP2 scenario was implemented to provide a consistent analysis (SSP2 is consistent with both RCP26 and RCP60) of the sensitivity of burnt area, fire size and fire intensity to future climate, vegetation and human activity.

We have added the following justification to the manuscript:

“The choice of future scenarios was motivated by the future wildfire simulations conducted by the Inter-Sectoral Impact Model Intercomparison Project (ISIMIP) for the United Nations Environment Programme (16). Climate and human activity data from the ISMIP2b protocol were selected because the protocol provides a high (RCP 2.6) and a low (RCP 6.0) mitigation scenario independent of socioeconomic activity pathway scenarios, so that users can make

their own coupling of future climate and socio-economic scenarios (Frieler et al., 2017). This is not the case for the newer ISIMIP3b protocol, where climate and socio-economic scenarios are more closely coupled. Given the large uncertainty that remains in building future socio-economic pathways, our approach minimizes uncertainty around future human activity. We selected a single pathway for both the high and the low mitigation scenario, which was most similar to current socioeconomic trajectories, the ‘Middle of the road’ socioeconomic activity pathway (SSP2). The use of the ISIMIP2b protocol also allowed for a qualitative comparison between the general spatial trends with the outputs from the ISIMIP projections (16).”

Frieler, K., Lange, S., Piontek, F., Reyer, C.P., Schewe, J., Warszawski, L., Zhao, F., Chini, L., Denvil, S., Emanuel, K. and Geiger, T., 2017. Assessing the impacts of 1.5°C global warming—simulation protocol of the Inter-Sectoral Impact Model Intercomparison Project (ISIMIP2b). *Geoscientific Model Development*, 10(12), pp.4321-4345.

“I may have misinterpreted the model set-up, but was the difference calculated against historical isimip2b 2010-2015 or WFDE5 2010-2015? The text in the methods and numbers in Table S4 implied that you are comparing against WFDE5 2010-2015.

ISIMIP2b bias correcting preserves GCM trends and variability, which means that there is still some important remaining differences in key fire-related variables even for the present day (Mathison et al. 2023). So, when calculating future anomalies vs present day, you do need to calculate the difference compared to individual isimip2b model 2010-2015 fire simulations.”

We apologise for the lack of clarity in the initial manuscript. The future anomalies were calculated as follows. Firstly, the anomaly between the *piControl* 2091-2100 to the relevant climate scenario 2091-2100 simulation for each GCM was calculated (from a common 10-year period). This was done to ensure that we were isolating the climate and vegetation signals of the impacts of global warming of a high and a low mitigation scenario above pre-industrial level for each of the four GCMs. These anomalies were then added to the climatology on which the fire model is trained. This was done to ensure that the change in each variable represents solely the difference between the modern-day empirical models and a high or low mitigation future climate, allowing us to account for potential differences in the underlying GCMs as well as potential biases in the empirical models themselves.

We will add this clarification to the manuscript:

“To account for potential bias due to differences between the GCMs and to eliminate the effect of any other external drivers, anomalies were calculated between pre-industrial climate and vegetation inputs and the simulated future climate and vegetation inputs under high (RCP2.6) and low (RCP6.0) mitigation for all four GCMs. These anomalies were then added to the original climate data used to train the fire models to provide the future inputs for the simulations. For socio-economic changes, we used the anomalies between the mean of the 2090-2100 decade and the original training period (2010-2015) of the SSP2 simulations to account for potential biases in population density, cropland cover and GDP. This ensures that the observed changes resulted solely from changes in the climate and vegetation forcing as opposed to factors such as land-use change.”

“The uncertainty ranges seem to be based on the different GCMs' responses, thereby sampling the uncertainty in forcing (via high and low mitigation) and the uncertainty in climate response

(via multiple GCMs). Is that correct? Fire response to climate and land use forcing is also a substantial part of uncertainty, and should, if possible, be considered. GLMs come with a few burways to quantify uncertainty (confidence/predictive intervals, error, residuals, bootstrapping techniques, Bayesian Inference etc) many of which are relatively easy to implement. It would be nice to also have some level of confidence/uncertainty quantification of modelled fire responses.”

We are indeed using the different GCMs under both a high and a low mitigation scenario to sample the uncertainty in forcing and climate response. To address the uncertainty in fire response, we have computed a 95% predictive confidence interval with lower and upper bound values for all the experiments and the GLMs themselves to quantify the uncertainty in the models. We have added the 95% predictive confidence intervals to the Supplementary tables and also in the main text when simulated values are given. We have also added a plot showing the spread of the lower and upper bound estimates, which provides insights into the spatial uncertainties, to the Supplementary.

We have added a new section to the online methods to describe how uncertainty was evaluated, as follows:

4. Uncertainty associated with BA, FS and FI GLM models

To assess the uncertainties associated with the response of fire to climate, vegetation and human activity we calculated the 95% predictive confidence intervals for the BA, FS and FI models separately by obtaining the upper and lower interval values through calculating plus and minus the fitted value multiplied by 1.96 the standard error. We also examined the spatial patterns of these uncertainties (distance between upper and low bound of the 95% confidence interval) to determine whether there were spatial biases in the predictions. These analyses show that the uncertainty both along environmental gradients and in terms of spatial distribution (Supplementary Figure 3) is relatively small.

Supplementary Figure 3. Partial residual relationships between predicted vs observed fire properties (in black) and the associated 95% confidence interval (in red) for (A) burnt area (BA), (C) fire size (FS) and (E) fire intensity (FI). The spatial patterns of the spread of the confidence interval (range between upper and lower bounds of the 95% confidence intervals) are shown for (B) BA, (D) FS and (F) FI. These plots show that the model uncertainties are small over most of the observed range of each fire property.

We have modified the text (line 272 onwards) to include a discussion of these uncertainties as follows:

“There is a high level of uncertainty associated with modelling future changes in wildfires. The aim of this analysis was not to provide predictions but to show the sensitivity of wildfire globally to future changes in their different drivers. We have addressed issues of uncertainty in future forcing using a high and a low mitigation scenario as well as the uncertainty in the response of climate and vegetation inputs to these changes in forcing using multiple GCMs. An additional uncertainty that is more challenging to address is that associated with the fire response itself. The GLMs used to model this response have been shown to simulate realistic patterns under Last Glacial Maximum conditions, which increases our trust in their ability to perform out-of-sample (Haas et al., 2023). We have also characterised the uncertainty in this response by considering a 95% confidence interval on the modelled relationships, which shows

that globally the uncertainties are relatively small. However, GLMs do provide less flexibility in the potential fire response to climate and land use forcing than other non-linear modelling approaches. It has been argued that machine-learning models have greater predictive power than linear modelling tools, but they have limited ability to predict outside of the range on which they were trained. In addition, their tendency to overfit is challenging in the case of wildfires, which involve highly stochastic processes (Boulanger et al., 2018). Thus, it is not clear that using these alternative modelling approaches would have reduced prediction uncertainty substantially.”

“L30-31: ref 16 (UNEP 2022) seems like an obvious reference for this point as well.”

We have added the reference to the UNEP report here.

“L39: Warmers and dry conditions impact on fuel dynamics may be more complicated than the simple “reduced fuel load and fuel continuity” listed here (i.e. Different vegetation composition, fuel types and flammability, delayed decomposition, vegetation and fuel height structure, etc). Maybe rephrase to point out that the relationship is complex. Though the point is valid – warmer & drier don’t necessarily mean more fire activity.”

We have expanded the text line 38:

“However, these physical indices do not take account of the potential for changes in vegetation to moderate the effect of future climate changes: warmer and drier conditions increase the incidence of fire weather, for example, but the effect that changes in climate may have on vegetation is complex, and its impact on fire not straightforward. On the one hand, warmer conditions can reduce fuel loads and fuel continuity, decreasing fire risk (11-14,8). However, warmer and drier conditions can also have indirect effects vegetation composition and type, which in turn influences landscape flammability. They also influence individual species, changing plant traits and vegetation structure (Enright et al., 2014; Pausas et al 2017). Physical indices such as measures of fire weather equally do not take account of the potential for human activities to mitigate wildfire risk (15).”

Pausas, J.G., Keeley, J.E. and Schwilk, D.W., 2017. Flammability as an ecological and evolutionary driver. *Journal of Ecology*, 105(2), pp.289-297.

Enright, N.J., Fontaine, J.B., Lamont, B.B., Miller, B.P. and Westcott, V.C., 2014. Resistance and resilience to changing climate and fire regime depend on plant functional traits. *Journal of Ecology*, 102(6), pp.1572-1581.

“L42-44: As far as I’m aware, no process-based global fire model study has used ISIMIP2b data to perform future projections in burnt area, intensity or fire size (though there might be individual model studies I don’t know about). Ref 16, though, definitely does not use a process-based fire model. UNEP (2022) actually uses a very similar experimental design presented here: an empirical model trained on observed burnt area and driven by ISMIP2b climate data and using a vegetation model to provide maps of potential vegetation cover as proxy for fuel. It is still worth including this as past work, and this study still has plenty of weaknesses (i.e. only looking at changes in very high burnt areas, not looking at intensity and fire size, etc), which makes your analysis still very exciting. But its worth making sure the description of this (and other previous studies) is accurate.”

“L44-46: the point that process-based models haven’t been shown to have the agreement or skill to reproduce trends and therefore future projections is definitely valid. However, it is worth noting that these studies cited here are now using fire models from 10 years ago, and newer models when combined with observations and advanced statistical techniques are starting to constrain around long-term trends (i.e Burton & Lampe et al. 2023).”

We have corrected this mistake and rephrased this paragraph (line 52) to include the point about process-based models:

“Empirical analyses have considered vegetation and human impacts on fire risk and are being used in the Inter-Sectoral Impact Model Intercomparison Project (ISIMIP) to assess potential future wildfire regimes under climate change (16, Jones et al., 2022) However, these studies tend to focus on extreme fire behaviour as well as changes in number of fires or changes in very high burnt areas (16) with little attention paid to other fire properties such as size or intensity. Process-based global models, which simulate full fire behaviour, still differ considerably in their predictions of historic trends (17) and do not skilfully capture wildfire seasonality and interannual variability (18), although recent efforts to improve these models have led to greater constraint in the longer-term trends (Burton & Lampe et al. 2023). Nevertheless, the ability of these process-based models to provide insights into future trends remains limited.”

Jones, M.W., Abatzoglou, J.T., Veraverbeke, S., Andela, N., Lasslop, G., Forkel, M., Smith, A.J., Burton, C., Betts, R.A., van der Werf, G.R. and Sitch, S., 2022. Global and regional trends and drivers of fire under climate change. *Reviews of Geophysics*, 60(3), p.e2020RG000726.

Burton, C., Lampe, S., Kelley, D., Thiery, W., Hantson, S., Christidis, N., Gudmundsson, L., Forrest, M. and Burke, E., 2023. Global burned area increasingly explained by climate change. *Research Square*, preprint [https://doi.org/10, 21203](https://doi.org/10.21203).

“Line 59: As ISIMIP2b uses CMIP5 models, I don’t think “state-of-the-art” applies here.”

We have deleted the phrase “state-of-the-art”.

“Line 78: Do you need to refer to Figure S3 here for an individual fire property breakdown?”

We have added a reference to the figures in the supplementary which provide the maps of the individual fire property breakdowns and altered the caption to read:

“Fig 1. Regions of change in burnt area (BA), fire size (FS) and fire intensity (FI) under the high mitigation scenario (top) and the low mitigation scenario (bottom). Regions where all three properties increase are shown in red, regions where all three properties decrease are shown in dark blue. The global maps of the anomalies (in total change) in each individual fire property under both high and low mitigation are given in Fig S3”

“Line 94-96 (looking at the number of gridcells where fire’s occurred). Is this a relevant or useful measure? For one thing, gridcell areas are, by their nature, somewhat arbitrary and the number would change depending on the resolution. Also, the GLMs have been trained on observations that are known to miss some unobservable fires (even GFED5 won’t capture all burning in e.g. closed campy forests). And because of the lack of information of grid cell heterogeneity, these simulations don’t say if the same locations of these grid cells are burning

each year or if fires can be randomly distributed. Also, it seems like it is not a very important point anyway. Unless I've misinterpreted, I would take it out."

We agree that this is not a useful measure for fire size and intensity. In the case of burnt area, it provides a measure of how many more regions are experiencing wildfires due to the inclusion of an ignition threshold in this analysis (a point which is also discussed below) and thus provides a way of showing that fire is occurring in new areas. This is the case, for example, in the northern higher latitudes, where more grid cells experience burning in the future compared to the present. We will add a sentence here to make the utility of this metric clearer:

"Whilst the global signal is dominated by the reduction of burnt area in tropical regions, globally, the number of grid cells in which the probability of burning exceeded our ignition threshold was 27-30% greater by the end of the 21st century than under modern conditions, and most (87-88%) of this increase occurred in the northern extra-tropics. In these regions, the probability of burning increases above a threshold under which no wildfire activity is expected to occur. Thus, these increases represent an encroachment of wildfires into areas where previously there were none."

Line 133-134: Does the model capture the possible transition to less but faster drying fuel seen in some grassier ecosystems? If not, would this impact your results?

Our results do not account for this effect explicitly since the only plant functional type classification we have is percent of tree, shrub and grass. We do distinguish between C3 and C4 photosynthesis, but this only affects overall GPP in our framework. However, it is unlikely that adding this process would substantially modify the results whereas it would add an extra layer of uncertainty, which we are trying to avoid – given the large number of uncertainties already associated with this exercise.

Line 139: "increased drying and GPP" is this due to CO₂ effects or something else?

The increased dryness is a result of precipitation changes in the GCM projections. The increased GPP is a combination of both CO₂ and climate effects, although we expect the CO₂ effect will dominate when we compare the changes simulated in the climate-only and the CO₂-only experiments.

Line 161-162: Is this first sentence in reference to Burnt Area?

Yes, we will make this clear as follows:

"Human-driven reductions in burnt area in tropical regions are offset by the combined effects of changes in climate and CO₂."

Line 206/207: ref 16 tends to show increasing fire activity over the southern Amazon and Cerrado, in line with this studies BA and FS. Line 209: ref 16 and (I think) 30 shows increasing fire activity at northern high latitudes.

Line 211: ref 16 does not report changes in the total global burnt area under high mitigation (it instead reports changes in probability in the occurrence of locally unusually high burnt areas). The higher mitigation in ref 29 and 30 appears to be B1 used in CMIP3, which is roughly the equivalent of RCP4.5 and, therefore, does not have the same level of mitigation as RCP2.6 tested here. They also use an older generation of climate models, which could also explain the

disagreement. And I'm not sure how to ascertain a sensible trend from ref 33, though if anything, it looks like RCP2.6 shows increases in fire activity.

We have rewritten this part of the discussion:

“Given that simulations of future fire activity have not been performed in a systematic way, it is difficult to assess agreement between previous studies over the sign of the global trend in future fire probability. However, our conclusions are in line with previous studies (16, 29-33) in showing a redistribution of global wildfires under future conditions, with increases in the northern high latitudes and decreases in tropical regions. There are still differences in regional trends, especially in the Amazon where our simulations show large increases in fire activity whereas previous studies show either non-significant changes (16) or decreases in fire probability (16,30). Furthermore, although most studies show the largest increases of fire probability in the northern latitudes, some regions show decreasing fire activity despite an overall increasing trend (16,29-30,32).”

Line 220: Are the different GLMs connected somehow, i.e. using some over-arching hierarchical sampling of parameters? Or are they entirely independent (besides using similar driving data?) If they are not formally connected, is it legitimate to test and comment on the breakdown of fire clusters? These could be dependent on feedback (including indirect feedbacks) between fire regime aspects that are not represented in the model.

The models are independent GLMs and were not connected in any way when they were constructed. However, we do apply a manual threshold value for ignition to the fire size and fire intensity models. When the probability of burning is below this threshold, the predicted fire size and fire intensity are also zero. Therefore, all three fire properties cannot be completely decoupled since they do still rely on the same initial conditions for ignition to occur. We do not consider feedback between different fire properties (for example the effect of the intermediate fire occurrence-intensity hypothesis (Luo et al., 2017)). However, not including this effect does not prevent us commenting on the breakdown of fire clusters, since we are not commenting on tightly coupled clusters per se (with clearly defined boundary conditions) but on the overall direction of the trend in the three fire properties, showing that they may decrease or increase in the future due to different sensitivity and relationships to the predictor variables. Successfully fitting three independent models to these three fire properties also shows that they are driven by different predictors. The changes made to the manuscript are embedded with the changes that we address in the next point, but we have underlined the specific reference to this comment below.

On a similar point...

Line 246-250: Empirical models, such as this, lack fire-fuel feedbacks. Woody encroachment under reduced burning mentioned here, could effect fire regimes, but also the removal of fuel or shift in ecosystem composition from fire will not be represented. Some discussion on how this might impact your results, and as a source of possible disagreement with previous work, would be great somewhere.

Whilst we agree with the reviewer that fire-fuel feedback will play an important role on the future patterns of wildfire on Earth, getting those feedbacks right would require accurately representing vegetation properties in land surface models as well as the relationship between

fire properties and vegetation properties accurately in a dynamic coupling, which is out of the scope of this analysis.

We have added this sentence to the discussion:

“In some regions, increases in burnt area are associated with decreases in fire size and intensity. Direct feedback between these different fire properties (for example how fire activity may constrain fire intensity (Luo et al., 2017) or between fire and vegetation) were not considered here. These feedbacks may potentially offset some of the differences in the diverging trends seen between burnt area, fire size and fire intensity as well as explain differences with previous studies. To address the impact of feedback on different aspects of the fire regime itself it will be necessary to move from an offline modelling exercise to a dynamic coupling of wildfires in an Earth System Model. Nevertheless, although lack of consideration of these feedbacks may increase the uncertainty of our analysis, it is not surprising to observe divergence in simulated behaviour of the different fire properties— given that burnt area, fire size and fire intensity are driven by different variables.”

Luo, R., Hui, D., Miao, N., Liang, C. and Wells, N., 2017. Global relationship of fire occurrence and fire intensity: A test of intermediate fire occurrence-intensity hypothesis. *Journal of Geophysical Research: Biogeosciences*, 122(5), pp.1123-1136.

Line 577: Can you supply a version-controlled repository or doi with the model code/configuration rather than just a link to the current code?

We apologise for this. The project was public, but the individual folders were not published. This has now been corrected, and all the data, scripts and results are available here:

- Haas, Olivia (2023). final data. figshare. Dataset.
<https://doi.org/10.6084/m9.figshare.24764589.v1>
- Haas, Olivia (2023). set up the models. figshare. Dataset.
<https://doi.org/10.6084/m9.figshare.24764583.v1>
- Haas, Olivia (2023). run models. figshare. Dataset.
<https://doi.org/10.6084/m9.figshare.24764577.v1>
- Haas, Olivia (2023). results scripts. figshare. Software.
<https://doi.org/10.6084/m9.figshare.24764571.v1>

Line 609: See comment on lines 94-96: do you need this threshold? And if so, why 0.0011?

All three reviewers have raised questions about the ignition threshold, so we give our overall response to these questions here. The threshold was introduced because the fire size and fire intensity models have been trained on existing fires. If a fire does not occur, there is no data documenting its size or intensity. This fact introduces a bias in the models because they cannot simulate a realistic size or intensity under conditions where no burning takes place. We introduced a threshold based on the BA model to indicate grid cells in which the probability of burning is zero, and fire size and fire intensity should therefore also be zero. We obtained this threshold from the BA GLM, looking at the distribution of predicted values when observed BA was zero.

We have added the following explanation to the online methods to make this point clear:

“Given that BA represents a probability of burning between 0 and 1, the BA model returns a physically meaningful value for each grid cell and does not require further manipulation. However, both the GLM models for FS and FI return values of estimated FS and FI assuming a fire occurs, since the models were fitted to observed data for FS and FI. Given that these models cannot determine if an ignition occurred, they return a value for all grid cells, even those in which the probability of burning is zero.

It is therefore necessary to apply an ignition threshold to study changes in FS and FI. We obtained the ignition threshold value from the distribution of reconstructed BA values in the original model training period (2010-2015) when the observed BA value was zero. There are no systematic biases evident from plotting the residuals of the model; but there is a compression of the range of reconstructed values, leading to apparent over- (under-) prediction at the low (high) extremes. However, the reconstructed values never reach zero, as it is a probabilistic model. Globally, 26% of grid-cells had an observed BA of zero (a total of 14,816 data points and associated fitted values). We took the median value (0.0011) of the fitted values in these grid cells to set the threshold for ignition. In grid cells in which burnt area is simulated to be lower than a fractional value of 0.0011, fire size and fire intensity were set to zero.

Fig S1. Histograms showing the distribution of the fitted values by the GLM BA model when observed BA values are 0 in the 2010-2015 period for (a) the whole range and (b) the range up to the 95th percentile. The red line shows the median value, and the black lines show the 10th and 90th percentile values.

Table 1. Statistics for the fitted BA distribution when observed BA is 0 for the 2010-2015 climatology.

	10 th percentile	50 th percentile	90 th percentile
Fitted BA	0.0001	0.0011	0.0084

[Figure Redacted]

Figure 2. Maps of BA ignition mask (where no burning is assumed to occur) under modern-day conditions (2010-2015 climatology) showing in red (a) where the observational BA values are 0 and (b) where the fitted₂ BA values are equal or lower to 0.0011.”

Line 666-671: This sounds like a generally a good experimental design to produce vegetation/fuel information into the future. But it is a single model approach and therefore wont capture uncertainty in responses of fuel load to future climate. Some discussion on how using that single realisation might affect your uncertainty analysis would be interesting.

We will add the following to the discussion:

“We use a single model approach to simulate our vegetation productivity predictors. We therefore cannot assess the uncertainty related to model specification for our vegetation inputs. We were interested in producing a PFT-independent analysis, and the optimality-based productivity model used here was the only model that allows this (Stocker et al., 2020). There are limitations to the model (Bloomfield et al., 2023) but it has been shown to outperform other vegetation models in predicting seasonal changes in leaf area (Zhou et al., 2024) and can quantitatively explain observed changes in vegetation as a response to environmental change (Zhu et al., 2023) as well as global greening patterns (Cai et al., 2023).”

Bloomfield, K.J., Stocker, B.D., Keenan, T.F. and Prentice, I.C., 2023. Environmental controls on the light use efficiency of terrestrial gross primary production. Global change biology, 29(4), pp.1037-1053.

Zhu, Z., Wang, H., Harrison, S.P., Prentice, I.C., Qiao, S. and Tan, S., 2023. Optimality principles explaining divergent responses of alpine vegetation to environmental change. Global Change Biology, 29(1), pp.126-142.

Cai, W., Zhu, Z., Harrison, S.P., Ryu, Y., Wang, H., Zhou, B. and Prentice, I.C., 2023. A unifying principle for global greenness patterns and trends. bioRxiv, doi: <https://doi.org/10.1101/2023.02.25.529932>

Zhou, B., Cai, W., Zhu, Z., Wang, H., Harrison, S.P., Prentice, I.C., 2024. A general model for the seasonal to decadal dynamics of leaf area. *bioRxiv* doi: <https://doi.org/10.1101/2024.10.23.619947>

Line 678-683: Similarly, using that same mean fractional covers across biomes might wash out a lot of the interesting within-biome responses. A brief discussion on how that impacts your results would be great.

It is true that some within-biome detail is potentially being lost. We have reviewed the analysis and accounted for within-biome variation by obtaining grid-cell values of tree, shrub and grass cover from the BIOME4 model. This was done by calculating the fraction of NPP for each PFT in BIOME4 relative to the total NPP in a grid cell to obtain the relative coverage of this grid cell by this PFT. The PFTs were then aggregated to tree/grass and shrub categories and scaled to the ESA CCI land cover data (to account for the bias introduced by shading of grasses by trees, which is not accounted for in the NPP calculation). We then performed all the analyses again and have accordingly adjusted the numbers. Accounting for this within-biome variation did not make any difference to the overall conclusions.

We have added the following to the methods:

“

3. Realistic experiments when considering grid-cell level tree, shrub and grass cover

In this analysis we consider differences between vegetation types through the distinction of fractional tree, grass and shrub cover only. These fractions are obtained by averaging values for each biome globally. Grid-level fractional covers were computed, but did not show significant differences regionally in simulated burnt area, fire size or fire intensity, both under modern and future conditions. Accordingly, the impact of within-biome differences was removed to ensure a more rigorous analysis and limit uncertainties. Research has consistently shown that current biomes are not necessarily representative of differences in fire properties; and that other plant traits that are not captured in this classification, relating to flammability and recovery, may be more important (Belcher et al., 2018; McLauchlan et al., 2019; Harrison et al., 2021). Moreover, predictors relating to vegetation amount have been shown to be more important in capturing burnt area than predictors relating to vegetation type (Haas et al., 2024). Our vegetation productivity estimates are independent of plant functional type parameters and do allow us to observe regional differences between regions with similar conditions, reducing the uncertainty introduced by the use of fixed vegetation categories.

Figure 4. Burnt area, fire size and fire intensity simulated using constant grass, tree and shrub cover within each BIOME4 biome (top panel) and using variable grass, tree and shrub cover within each BIOME4 biome (bottom panel).

Figure 5. The percentage difference in modelled burnt area between constant tree, shrub and grass cover within each biome and varying tree, shrub and grass cover within each biome.

Table 4. Global burnt area ($M km^2$), fire size (km^2) and fire intensity (km^{-1}) using ESA landcover, BIOME4 simulated constant land cover and BIOME4 simulated varying land cover, along with the percentage change between the modern and end-of-century simulations for RCP2.6 and RCP6.0

	Modern	Percentage change RCP2.6	Percentage change RCP6.0
Burnt Area			
ESA	4.44		
Constant	3.96	-11.39	-9.20

Variable	3.81	27.68	29.08
Fire Size			
ESA	2.59		
Constant	3.86	27.46	20.59
Variable	2.80	29.53	22.23
Fire Intensity			
ESA	31.25		
Constant	26.54	5.87	-2.30
Variable	26.30	5.19	-2.15

Belcher, C.M., Archibald, S., Lehmann, C.E.R., Bond, W.J., Bradstock, R.A., Daniau, A.L., Dexter, K.G., Ferrelstel, E.J., Greve, M., He, T. and Higgins, S.I., 2018. Biological and geophysical feedbacks with fire in the Earth system.

*McLauchlan, K.K., Higuera, P.E., Miesel, J., Rogers, B.M., Schweitzer, J., Shuman, J.K., Tepley, A.J., Varner, J.M., Veblen, T.T., Adalsteinsson, S.A. and Balch, J.K., 2020. Fire as a fundamental ecological process: Research advances and frontiers. *Journal of Ecology*, 108(5), pp.2047-2069.*

*Harrison, S.P., Prentice, I.C., Bloomfield, K.J., Dong, N., Forkel, M., Forrest, M., Ningthoujam, R.K., Pellegrini, A., Shen, Y., Baudena, M. and Cardoso, A.W., 2021. Understanding and modelling wildfire regimes: an ecological perspective. *Environmental Research Letters*, 16(12), p.125008.*

Line 704: A similar point – I like the setup here for human activity. I don't think we're at a point yet where we can test the uncertainty in future human activity that leads to similar climate pathways, so following the ISIMIP protocol for these variables makes pragmatic sense. But somewhere (probably in the main) some acknowledgment that this will have an impact on your projections would be useful.

We agree with the reviewer, and will add the following to the discussion at line 232:

“This analysis only considers one future socio-economic scenario (SSP2). Projecting how future socio-economic systems will respond in the future is more challenging than projecting the response of climate to changes in future forcing. We therefore do not comment on the uncertainty relating to future human activity but work with the ‘middle-of-the-road’ scenario and focus on the differences between the modern-day and one actualisation of the future SSP2 scenario”.

Supplementary Info

Line 777: I don't see any quantification of the difference/agreement in trends between the model and observations. This is important because one of the few ways to infer if a model might capture future out of sample projections from present day observations is if they are getting the correct present-day trend (I would trust this over NME comparisons), and it's hard to see if there is agreement in overall trend across the time period in Figures S1 and S2.

The GLM models were not designed to capture interannual variability or seasonal cycles but rather the spatial patterns of fire regimes given a decadal average background climate, land cover and human activity. We have used the NME step 1 score to compare the spatial

performance of the GLM models to both observational products and FireMIP simulations. We have reported the NME score averaged over the 10-year period, but the NME score is below 1 for each individual year for both GFED4s and GFED5. This shows that the GLM model performs well in terms of simulating the spatial patterns of burnt area compared to the FireMIP models. We have added a table with the trends in burnt area (in Mha) for the modelled burnt area, GFED4s and GFED5 over the 2006-2015 period. The model does not capture the significant decreasing trend shown in GFED5. However, GFED4s does not show a significant trend globally over the same period, highlighting the disagreement between the observational products.

Table S4. Trend in burnt area (in Mha) between 2006 and 2015 between modelled BA and GFED4s and GFED5. Significant trends are reported with * for $p < 0.05$ and *** for $p < 0.0001$

	Modelled BA	GFED4s	GFED5
Global	0.13	-0.16	-0.23*
Northern hemisphere	0	-0.02	-0.08
Above 60°N	0.11	0.10*	0.14
Southern hemisphere	0.18***	-0.17	-0.26***
Tropics (between 30°N and 30°S)	0.12	-0.15	-0.16
African continent	0.18***	-0.18	-0.26***

Table S5. NME scores (step 1) for modelled burnt area against GFED4s and GFED5 for the 2006 to 2015 period.

Year	GFED4s	GFED5
2006	0.67	0.87
2007	0.66	0.88
2008	0.67	0.94
2009	0.61	0.84
2010	0.65	0.83
2011	0.65	0.82
2012	0.65	0.8
2013	0.61	0.76
2014	0.66	0.85
2015	0.67	0.83

Line 778: This is the first round of fireMIP? It might be worth noting these fire models are now quite old.

This is indeed the first round of FireMIP experiments, but the latest experiments are not available. We will add this to the caption:

“Table S3. Burnt area correlations and benchmarking scores of modelled burnt area and the global fire models included in the first phase of the Fire Modelling Intercomparison Project (FireMIP) compared to GFED4s and GFED5. The normalized mean error (NME) scores for the spatial patterns (18, 85) are for the overlapping period (2006-2013). A perfect score is 0, and a score below 1 is better than the mean model. The mean value and the range (in brackets)

are given for the first round of the FireMIP models (2014) because the phase2 FireMIP results are not yet available.”

Fig S1, S2: Can you use a more accessible colour scheme than the green-red combination?

We have made a new version of the figure.

Fig S1. Global 10-year trend in burnt area (BA) from 2006-2015 in Mha from the model (red), GFED4(s) (blue) and GFED5 (black).

Fig S2. Global 10-year trend in burnt area (BA) for the (A) northern hemisphere, (B) above 60°N, (C) southern hemisphere, (D) tropics, (E) northern hemisphere America and (F) southern hemisphere America from the model (in red), GFED4(s) (blue) and GFED5 (black)

Reviewer 2

Major points:

“I believe the article would be strengthened with a discussion of regional models that can more intimately address additional mitigation policies, which include both land and fire management. In the attached manuscript you will find detailed comments. Below are some previous research that should be considered in the discussion.”

We agree with the reviewer that a consideration of management policies would be useful, and that regional models provide tools through which the outcomes of such policies can be simulated. However, the point of our analysis was to highlight the sensitivity of different aspects of the fire regime to changing conditions because this is a pre-requisite for addressing mitigation policies and management. We have reworked the last paragraph of the discussion as follows to include greater reference to these different scales:

“In the tropics for example, a potential consequence of the predicted decrease in burnt area, fire size and fire intensity is woody encroachment, which impacts biodiversity, the carbon cycle, and land-atmosphere interactions and has consequences for local economies, livestock production, tourism, and pastoralism (Eldridge et al., 2011; Shackleton et al., 2013; Gray and Bond., 2013). High-intensity fires have been used to limit woody encroachment, but with limited evidence of success (47-48). Across the African continent, a broad range of fire management strategies are already in place in savanna protected areas (Nieman et al., 2021) and the potential for fire management to help finance local restoration and economies has been highlighted (Tear et al., 2021). Under a changing climate, more investment is needed in these regions to ensure that fire management is successful. Involving local communities and knowledge in these efforts will provide the greatest resilience (49-52). In the extra-tropics and other regions where burnt area, fire size and fire intensity are predicted to increase, increased risks to life, property, and commercial activities are a major concern (16,53). Previous fire policies may influence future management and what outcomes are desirable. As a result of a long history of fire exclusion in the United States, for example, introducing fire to forests where fire has been previously excluded may reduce high-severity fires but increase total burnt area (Young et al., 2021). Regional and landscape level models could be used to address these management issues (Prichard et al., 2023).”

“What is the difference between high and low mitigation. I see this detail is in the methods with regards to the different warming scenarios. Be specific that it is mitigation of global warming. I understand the desire to limit methods, but this information needs to be worked into the introduction to set up the reader for success. Anything that is new to this paper/implementation of the model should be discussed in more detail within the introduction. At least having a conversation around key drivers and expectations.”

We have added a flowchart describing the set-up of the different experiments and a table explaining the hypothesis driving the inclusion of each predictor to the main text which we hope will help clarify this.

Table 1. The GLM models initially used a set of sixteen predictors. The individual predictors as described below, as well as the rationale behind their inclusion in the model.

Predictors	Abbreviation	Value	Rationale
-------------------	---------------------	--------------	------------------

Climate			
Dry days	DD	Mean monthly number of dry days	Flammability (To capture drying out of fuel over multiple days)
Dry day seasonality	(DD(s))	Range in value of monthly number of dry days divided by the mean monthly number of dry days	Fuel-build up (To account for regions with strong wet and dry seasons)
Vapour pressure deficit	VPD	Maximum monthly vapour pressure deficit	Fuel-build up Flammability (To capture atmospheric humidity's constraint on fuel-build up, fuel-drying, flammability)
Diurnal temperature range	DTR	Mean monthly diurnal temperature range	Flammability (To capture the drying out of fuel daily time frames)
Wind speed	wind	Wind speed of the hottest month	Ignition Fire spread
Vegetation			
Annual gross primary production	GPP	Sum of annual gross primary production	Fuel load (To capture fuel accumulation)
Gross primary production seasonality	GPP(s)	Range in value of monthly gross primary production divided by the mean monthly gross primary production	Fuel load (To account for seasonal variation in fuel load)
Grass cover	grass	Fractional annual grassland cover	Flammability
Shrub cover	shrub	Fractional annual shrubland cover	Flammability
Tree cover	tree	Fractional annual tree cover	Flammability Fuel load
Human activity			
Population density	popd	Annual mean population density	Ignitions
Cropland cover	crop	Fractional annual cropland cover	Suppression and land fragmentation
Road density	roads	Annual mean road density	Suppression and fragmentation
Topography			
Vector ruggedness index	VRM	Mean value (constant)	Land fragmentation and effects of topography

Topographical position index	TPI	Mean value (constant)	Land fragmentation and effects of topography
-----	-----------------------	--

Ignitions

Lightning	light	Mean monthly lightning ground-strikes	Ignitions
-----------	-------	---------------------------------------	-----------

Figure 1. Flowchart of the experimental setup showing realistic future experiments and the associated sensitivity experiments. The high mitigation scenario refers to end of century RCP2.6 and the low mitigation scenario refers to end of century RCP 6.0. Additional information on the source and availability of the data for each experiment can be found in the Supplementary Materials.

“Consider speaking to how the GCM handles the effects of wildfire activity on ecologies. It would help ground you subsequent modelling, or at a minimum would be fully transparent.”

In ISIMIP2b, wildfires are treated as external drivers. By taking the difference between the piControl period and the end of century period, we are removing the impacts of external drivers, and only accounting for changes in climate (Frieler et al., 2017). We will make this point clearer when discussing how the future climate was obtained:

“To account for potential bias due to differences between models in the varying input datasets and eliminate the effect of any other external drivers and ensure that the observed changes resulted solely from changes in the climate and vegetation forcing as opposed to of factors such as land-use change, anomalies were calculated between pre-industrial climate and vegetation inputs and the simulated future climate and vegetation inputs under high (RCP2.6) and low (RCP6.0) mitigation.”

“General shortfalls of the modeling that should be explored in this deployment of the models.

- Mean wind speed of the hottest month is likely not a good measure of the influence of wind on burned area. Wind driven fires are generally in the spring and fall, and not in the heat of the season. Try the mean wind speed of the windiest month between spring and fall months...or above a certain VPD where ignitions are more likely to occur.
- Fitting TPI as a quadratic should be explored (...TPI, TPI_squared,...). This would enable the model to capture the full spectrum of fire activity ranging from valleys to slopes to ridge tops. Including slope may also prove to be beneficial as it would enable the model to tease apart slopes from flat lands (in both cases TPI = 0). Try interaction terms between TPI and slope. Seasonality of GPP and DD may also benefit from being fit quadratically to enable finding maximums by enabling the model to find the waxing and waning of these metrics. This would improve your 'skill' of finding seasonality.”

We agree that alternative indices, and interactions, could be explored within a GLM framework. However, our goal here is to explore the sensitivity of **existing** models of burnt area, fire size and fire intensity under future conditions. We use three previously published models which have been shown to reproduce modern fire patterns reasonably (Haas et al., 2022) and have also been tested under past climates, specifically the Last Glacial Maximum (Haas et al., 2023). This makes us confident that the outcomes under future climates are meaningful and can provide insights that are helpful.

Minor points (from pdf):

Line 53: The reviewer suggested including a reference to the Dye et al., 2024 but this is a regional study and does not address past climate regimes. Our comment here that global fire models have not demonstrated the ability to simulate changes in fire under large climate changes in the past. We will stress this by rewording the text in the following way (line 71):

*“In addition, they have shown to have the ability to simulate the impact of large changes in climate at the last glacial in a realistic way, which other **global** fire models have yet to demonstrate (20)”*

Line 60: We have removed the reference to the Supplementary Materials and expanded the introduction to include more information about the models used as well as the set-up of the sensitivity analysis.

“In this study, we use recently developed empirical generalized linear models of monthly mean burnt area, median fire size and median fire intensity driven by lightning ignitions, climate, vegetation properties, topography, and human factors (see Table 1) (19). A quasi-binomial model was fitted to burnt area, and a quasi-Poisson model to fire size and intensity. These models have been shown to capture modern spatial patterns in fire properties (19).”

Line 61: We have removed all reference to counterfactual tests and replaced them with the term sensitivity analysis.

Line 75: We have changed the subheading titles, they now read:

- **Contrasting fire regimes under high (RCP 2.6) and low (RCP 6.0) mitigation**
 - o *Burnt area*

- *Fire size and intensity*
- *Key causes of fire regime change under high mitigation (RCP2.6)*
- *Key causes of fire regime change under low mitigation (RCP6.0)*

Line 78: We have added horizontal lines to this figure at 23.26°N and 23.26°S

Line 84: These numbers represent the range in the simulated outputs from the four different GCMs. We have added a table with the average values from the GCMs in the main text for all three fire variables and will refer to the supplementary table where the detail for each GCM (along with predicted confidence intervals) can be found.

Table 2. Summary of burnt area, fire size and fire intensity global values under modern-day and high and low mitigation by the end of the century. Ensemble values from all four GCMs are reported here. Detailed information as well as prediction confidences intervals for the individual GCMs can be found in Table S4.

	Modern-day (realistic)	High mitigation	Percentage change	Low mitigation	Percentage change
BA (km²)	4,059,433	3,436,795	-14.28	6,110,862	50.53
FS (km²)	2.84	3.74	32.05	3.64	28.37
FI (W.km⁻¹)	26.42	27.45	3.87	24.71	-6.49

Line 88: Yes, this is referring to the tropical regions of South America, we will add this to the sentence. The reference should also be to the maps in Fig 2 not Fig 3:

“This decrease does not occur homogenously across all tropical regions, however, and an increase in burnt area is simulated in the tropical regions of South America (see Table 3 and Fig 2).”

Line 89: We will refer to the newly added table and the table in the Supplementary that has these absolute values reported:

“Global fire size and mean global fire intensity are increased compared to modern conditions under this mitigation scenario (by 23-36% and 2-5% respectively, see Table 2 and S4)”

Line 92: We will refer to the different tropical biomes directly, and reference Fig 3 to make this clearer. These sections have been reworked.

Line 93: The land cover changes when the conditions are changed.

Comments from line 94 to 124:

We have reworked the results section. The two sub-headings “**Future fire regimes under high mitigation**” and “**Future fire regimes under low mitigation**” have been condensed into the heading “**Contrasting fire regimes under high (RCP 2.6) and low (RCP 6.0) mitigation**”, addressing first the trends in burnt area and then in fire size and intensity. Please refer to the newest version of the manuscript.

Comments from line 126 to 190:

These sections relate to the sensitivity analysis as well as interpretation of the realistic experiment, which have been reworked. Please refer to the newest version of the manuscript.:

Comments from line 190 to 270:

The discussion has been reworked and expanded to include reflections on potential limitations and we have expanded the discussion on policy implications, including the potential use of regional models. Please refer to the newest version of the manuscript.

Reviewer 3

Major comments

“I find myself questioning the validity of some of the findings, stemming primarily from reservations over the original models (<https://iopscience.iop.org/article/10.1088/1748-9326/ac6a69>) on which this paper is based. I recognise that paper is already published, but I need to evaluate the validity of those models in order to evaluate the validity of this paper. Those models didn't seem to include any interactions between explanatory variables, and the models assumed linear relationships between explanatory variables and fire metrics. Both decisions, in my opinion, are not justified, because interactive effects (e.g., dryness x fuel load) and non-linearities (e.g., hump-shaped effect of productivity) are likely to occur. The intermediate productivity hypothesis, for example, explains that fire activity is most common at intermediate productivity, but your GLMs assume a linear relationship, resulting in the effect that environments with extremely low productivity have the highest predicted fire intensities. This results, in my opinion, with quite questionable and unrealistic predictions (especially of fire intensity) in polar regions with low productivity. Further, there are some quite large extrapolations, with fire intensity being predicted in areas that haven't experienced fire in the satellite record. These effects can be seen in Fig 3, bottom panel, of the original paper, in which the highest fire intensities are predicted for the high arctic tundra, far southern South America, and a large section of the Tibetan plateau, all of which are regions with very little to no observed fire. Because I find those findings implausible, I am afraid I also question the validity of the findings in this current manuscript because it is based on those models. Further, the models don't really account for the fact that different vegetation types burn differently (e.g. boreal forest burns differently than temperate broadleaf forests, even if productivity is similar). I welcome the authors to correct me if I am wrong.”

We have addressed this major comment in three parts:

- “Interaction” between variables
- Fire intensity extending into the northern polar regions
- Differences in vegetation types

“Interaction” between variables

The reviewer raised about concerns about the methodology in this analysis,. These concerns may result from a misunderstanding of our modelling approach. The reviewer's concern was the ability of generalised linear models (GLMs) to capture interactions between predictors. Whilst generalized linear models assume linear relationships between each response variable and each predictor variable, this partial relationship is fitted **when all other predictors are held constant**. The model thus captures both the underlying linear relationships (when all other predictors are held constant) and the emerging relationships (when all predictors are allowed to change). The example given by the reviewer – the intermediate productivity hypothesis – arises from the combination of two linear relationships: all else being equal, increasing fire occurrence with increasing productivity, and decreasing fire occurrence with increasing moisture. The humped relationship between fire occurrence and productivity emerges from these two underlying relationships (Bistinas et al., 2014). GLMs reproduce this humped relationship at a global scale (Bistinas et al., 2014), and so does our burnt-area GLM (see Figure

below). Furthermore, this modelling technique is one of the most popular for dealing with issues of empirical fire modelling (Sergi Costafreda-Aumedes et al, 2017; Vilar et al., 2016; Bistinas et al., 2014).

Whilst issues of multicollinearity are a potential concern, we have used variance inflation factors to show that this is not an issue for our analyses.

Underlying vs emergent relationships

Figure 1. These plots were produced using climate data from the WFDE5 ERA5 dataset (Cucchi et al., 2020). GPP outputs are from the P model (Wang et al., 2017; Stocker et al., 2020) and burnt area data from GFED4 (Randerson et al., 2018). A GLM was fitted between GPP, precipitation and BA and the plots above are the resulting underlying and emergent relationships. On the left: the monotonically positive partial residual relationship fitted between annual GPP (annual mean from 2010-2015) and BA (annual mean from 2010-2015) when precipitation is held constant. In the middle: the monotonically negative partial residual relationship fitted between monthly precipitation (monthly mean from 2010-2015) and BA when GPP is held constant. On the right: the emergent, humped relationship between monthly precipitation and BA.

In order to make the methodology clearer for the reader, we have added the following to the Methods:

“GLMs provide interpretable results and due to their simplicity require very little computational power, making them an ideal tool for running a large set of sensitivity experiments and providing a way of quantifying independent effects of multiple predictors even when they are correlated (70). GLMs are useful because they can handle response variables with highly non-Gaussian error distributions, are embedded in a well-established (multiple regression) framework, which allows quantification of the independent effects of multiple predictors even if they are partially correlated with one another and provide partial residual plots showing the effect of each predictor while the others are held constant (Larsen & McCleary, 1972). The relative importance of each predictor can also be assessed using absolute t -values (the fitted regression coefficient of each predictor divided by its standard error). These t -values are unitless and scale-invariant (Grömping, 2015). GLMs have previously been used to model burnt area and have been shown to capture both underlying relationships (the relationship between a response and an explanatory variable when all other explanatory variables are held constant, such as the positive relationship between burnt area

and productivity) and emergent ones (which are a consequence of combinations of variables, such as the humped-relationship between burnt area and productivity)”

Fire intensity extending into the northern polar regions

The reviewer is correct to highlight the implausible fire intensity values in the northern polar regions; however, this is not due to the GLM models but rather to the fact that the probability of a fire occurring in the northern polar regions is zero. Whilst the burnt-area training data has meaningful zero values, the fire size and fire intensity models do not. So when projecting future trends, neither the fire size nor the fire intensity GLM model are sufficient alone; the probability of burning occurring must be considered first. This is why we introduced the ignition threshold, both in our previous study (Haas et al., 2023) and in this study. The threshold emerges from the fact that the BA model will never simulate a zero value, given that it is a probabilistic model. We studied the distribution of the predicted BA values when the observed BA values were zero. Taking the median of these values allowed us to set an ignition threshold of predicted BA < 0.0011 as indicating the absence of fire.

We have added this explanation to the Methods section of the manuscript, which can be found in our response to Reviewer 1. When this threshold is applied to present-day predictions, the spatial distributions look as follows:

Figure 2. From left to right: predicted burnt area, fire size and fire intensity for the 2010-2015 period, applying a 0.0011 threshold to fire size and intensity

Differences in vegetation types

Research in fire ecology has shown that current classifications of vegetation types (and plant functional types more generally) are not representative of differences in fire properties; plant traits relating to flammability and recovery are but cut across the boundaries of traditionally defined vegetation and plant functional types (Belcher et al., 2018; Harrison et al., 2021). Furthermore, predictors relating to vegetation amount have been shown to be more important in capturing burnt area than predictors relating to vegetation type (Haas et al., 2024). Given the uncertainty around which vegetation traits influence global wildfire patterns, one of the strengths of this analysis is that it does not consider vegetation type beyond the distinction of trees, shrubs and grasses. Given that our productivity estimates are independent of plant functional type parameters, they are much more flexible to a changing climate, reducing the uncertainty introduced by fixed vegetation categories.

We will add the following to the manuscript:

“In this analysis we consider differences between vegetation types only through the distinction of fractional tree, grass and shrub cover. These fractions are obtained by averaging values for each biome globally. We therefore do not consider within-biome responses, and this could potentially explain some regional differences. However, there are reasons to expect that little information is thereby lost. Research has consistently shown that current biomes are not representative of differences in fire properties and that other plant traits which are not captured in this classification, but that relate to flammability and recovery, are more important (Belcher et al., 2018; Harrison et al., 2021). Moreover, predictors relating to vegetation amount have been shown to be more important in capturing burnt area than predictors relating to vegetation type (Haas et al., 2024). Our vegetation productivity estimates are independent of plant functional type parameters and do allow us to observe regional differences between regions with similar conditions, reducing the uncertainty introduced by fixed vegetation categories.”

A second major concern is that the projections do not adequately consider uncertainty. In Fig 1, for example, there is no category for “unchanged”, and relatedly, there is no acknowledgement of uncertainty. It seems there has been no attempt to quantify the confidence associated with a given change. It's possible many of these changes are slightly positive or negative with large confidence intervals spanning "no change".

We have changed Figure 1 (which is now Figure 2) to only consider changes larger than 5% in all three fire properties. We have also added a map showing the spread of the confidence intervals in each experiment.

The figure and caption now read as below:

Editorial Note: Base maps in Fig 1 in this Peer Review file are produced using rnaturalearth.

Fig 1. Regions of change ($>5\%$) in burnt area (BA), fire size (FS) and fire intensity (FI) under the high mitigation scenario (top) and the low mitigation scenario (bottom). The tropics and extra tropics are delimited by dotted horizontal lines. Regions where all three properties increase are shown in red, regions where all three properties decrease are shown in dark blue (only increases of more than 5% in all three properties are considered).

Line-by-line comments

L 1. Minor point, but perhaps consider tweaking the title, because "global wildfires" reads as if there are wildfires that are global, whereas the study was global, but the fires aren't.

This point was also brought up by reviewer 2. We suggest changing the title to "Wildfires on a changing planet".

L37. And there is recent empirical evidence (not just projections) that indicate very intense fires are increasing <https://doi.org/10.1038/s41559-024-02452-2>

We have added this reference line 32:

"Weather conditions promoting fire, as measured by fire danger indices, consistently increase in future climate-change projections (1,9-10) — contributing to the concern raised by recent

megafires, which have been shown to be increasing in frequency in some regions (Cunningham et al., 2024)."

L38. Not clear from intro how your approach improves on this pitfall of not taking "account of the potential for changes in vegetation". More info needed in the intro on how your approach ensures a relevant veg state in the future, as well as relevant road and population density. (given the method-at-the-end format, I think there is a need to provide a bit more methodological info up front so a reader can understand the major steps as we read through)

In this analysis we make use of two models (the P Model: Wang et al., 2017; Stocker et al., 2020; and BIOME4: Kaplan et al., 2003) to simulate our vegetation predictors. For both these vegetation models, given that the climate and CO₂ conditions can be inputted manually, we are able to explore the (realistic) interaction of CO₂-induced changes and climate-induced changes and fire regimes as well as the sensitivity of fire regimes to each factor. BIOME4 is a coupled biogeography-biogeochemistry model that simulates potential natural vegetation types solely from information on latitude, CO₂ levels, climate, and soils. The P model (Wang et al., 2017; Stocker et al., 2020) is a data-driven vegetation model that provides robust gross primary production (GPP) estimates in a PFT- and biome-independent way. These two models together provide a flexible methodology to explore the effect of vegetation under a changing climate. For the human activity predictors, cropland, population density and GDP were taken directly from the ISIMIP2b SSP2 projections. Road density was estimated using a linear model which was based of the GRIP4 database and is presented in the Methods.

L49-51: I think some more explanation would help here. I went and read your earlier work (ref 19) and understand those models, but some more text may help to make what you did in that work clear in this paper, so that it's not necessary for a reader to have to read that paper to understand this one. I suggest adding that the "models" themselves were generalised linear models, and explain the response variables more precisely, including temporal scale and units. As far as I gather from the other paper, the models were of summarised versions of the fire data as opposed to individual fires/hotspots, I believe mean burnt area per month, median fire size per month, and median fire intensity.

We thank the reviewer for this point and reworked the introduction. A table with the variables (both response and input variables) has been added as well as a flowchart explaining the experimental set up.

Table 1. Can the right-hand column be turned into multiple columns, so that there's a column for climate, CO₂, pop density etc, with a few plain language words describing the various experiment components? Can these "experiments" be explained in a general way in the text; it's not clear in the main body of the paper whether the future scenarios, for example, also include changes to human population density/road density, and how vegetation is modelled in the future.

We have added a table/flowchart figure to describe each experiment and a table summarizing the rationale behind each predictor (already attached in the response for reviewer 2).

Fig 1. It's quite hard to interrogate the finer-scale patterns in the fig because of size and

resolution, e.g. I found it impossible to evaluate the area where I live. I know the fig will be higher res in final version, but I would advocate for a larger figure with the maps stacked vertically, taking up the full width. You only have 3 fairly small figs as is, so there should be plenty of space.

We have changed the figures to the layout suggested by the reviewer. All figures will be provided in 300 dpi resolution.

L 89. Are the "modern conditions" the empirical obs, or are they the model predictions for the modern conditions? Both are important, because if the model is projecting incorrectly to the present day, then some of the differences reported in this para could be due to that.

We thank the reviewer for this point, which was also raised by reviewer 1. We realise that the way in which our different input layers for the future were calculated was not explained clearly enough. The flowchart describing each sensitivity analysis should clarify this point.

We have added the following to the online methods section:

“To account for potential bias due to differences between models in the varying input datasets and ensure that the observed changes resulted solely from changes in the climate and vegetation forcing (as opposed to factors such as land-use change) anomalies were calculated between pre-industrial climate and vegetation inputs and the simulated future climate and vegetation inputs under the high and low mitigation scenarios. These anomalies were added to the original climatology data that the model was trained on, to provide the future inputs. For socio-economic changes, we used the anomalies between the mean of the 2090-2100 decade and the original training period (2010-2015) of the SSP2 simulations to account for potential biases in population densities, cropland cover and GDP.”

Table 2. I find it quite surprising that the current burned area lies in between two climate scenarios that are both warmer (2 and 3-4 deg); i.e., 2 degrees of warming (high mitigation) is predicted to cause a ~15% reduction in BA, while 3-4 degrees of warming (low mitigation) will lead to a 50% increase in BA. How do scenarios that both involve warming lead to such divergent patterns from the current day?

Both warming scenarios simulate increased global burnt area by the end of the century when human activity is held constant at modern day levels. It is only when socio-economic changes (which include increased population density and cropland expansions) are considered that the high mitigation scenario sees a global reduction in total burnt area, so both warming scenarios lead to the same pattern in burnt area **all else being equal**. However, the influence of human activity on wildfire patterns is driving the divergent patterns. This is in line with the literature highlighting a departure climate-driven trends in biomass burning after 1750 (Marlon et al., 2008). After a sharp increase in wildfire activity from 1750 to 1850, global wildfire activity declined (Marlon et al., 2008). This decline has more recently been shown in the satellite record and has been attributed to cropland expansion and large-scale agricultural activities (Andela et al., 2016). An increase in global burnt area under the low mitigation scenario signals that the suppressing effect of human activity may no longer outweigh the increased flammability of the Earth system under 3-4 degrees of warming.

Fig 2. It's hard to compare the same biome class in the side-by-side panels. I think it would be easier to do that if the two sub plots were stacked vertically, using the same axis ranges, so that it's visually easier to line up the bars

We have reworked this figure to include the feedback from the reviewer.

Fig 3. Percentage change in burnt area (BA), fire size (FS) and fire intensity (FI) per biome for (a) the high mitigation scenario and (b) the low mitigation scenario. See Supplementary materials for more detail.

L128-157: these paragraphs are dense slabs of text. Can you break them up a bit?

We have reworked these paragraphs, please see the revised manuscript.

L173-174: how reliable is a finding like this? This implies that the vegetation is currently not burning solely because of a lack of human ignitions; that doesn't seem plausible to me.

“Increased road density in previously fire-free regions also lead to increases in burnt area in Europe and the east coast of the United States across all four models. This finding implies that in fire-sensitive regions, increasing fragmentation leads to an increase in the probability of burning”.

We do not imply that burning is not occurring in these regions solely because of a lack of human ignitions. Changes in climate and vegetation will also lead to increases in the probability of burning, but they do not emerge as the primary driver in these locations. This finding does imply, however, that in these fire-sensitive regions, increasing fragmentation leads to an increase in the probability of burning. This does not appear to us to be implausible – indeed this effect has been documented in the literature (see Harrison et al., 2021 for a review).

Fig 3. For this to be understandable, the acronyms of the predictors need to be described in the main text, or here in the figure caption - e.g. what is "DD (s)". Also, explain in the captions how the values for each of the slices were determined.

We have made sure that all acronyms are described in the main text.

Somewhere in Discussion, I think it would be good to acknowledge that these results reflect predictions of the mean/median fire. But many, if not most, fires on earth are human orchestrated. So, this focus on the central tendency may overlook how the extremes are changing, which are arguably quite important (raised by <https://doi.org/10.1038/s41559-024-02452-2>)

There is growing evidence that at a global scale, ignitions are not limiting for burnt area in much of the world (**Bistinas et al., 2014; Haas et al., 2022**). This interpretation is supported by the fact that global fire models that assume an ignition-saturated world do not perform better to models that assume an ignition-limited one (**Hantson et al., 2020; Li et al., 2024; Haas et al., 2024**).

It may be that the greatest **number** of fires are started by people. This is certainly true for some densely settled regions, including most of Europe. It is also true for regions where, for example, field-scale burning of crop residues is widely practised. However it does not follow from this that the incidence of large and/or intense fires – even in these areas – is “orchestrated” by people. The abundance of small fires does not affect this type of modelling effort significantly because the controls of the fire regime still lie in the environmental realm, rather than being in human hands (**Kasoar et al., 2024**). Kasoar, M., Perkins, O., Millington, J.D., Mistry, J. and Smith, C., 2024. Model fires, not ignitions: Capturing the human dimension of global fire regimes. *Cell Reports Sustainability*, 1(6).

The distinction between mean/median background rates and extreme wildfire is a separate issue from that of human agency. Our analysis focuses on mean/median behaviour; trends in the extremes of the distribution are outside the scope of the study. For clarity, we will add the following text to the discussion:

*“This analysis focuses on the background rates of different fire properties. The models were designed to capture the climatological mean burnt area and median fire size and intensity given a set of longer-term climate, vegetation and human activity conditions. The analysis thus does not capture the most extreme fire behaviour. Extreme fire events may increase in frequency in the future (**Cunningham et al., 2024**) particularly where conditions become drier (**Hanston et al. 2017**). However, given the relatively small number of extreme fire events in the global record, they remain challenging to model (**Hantson et al., 2024; Jones et al. 2024**).”*

Kasoar, M., Perkins, O., Millington, J.D., Mistry, J. and Smith, C., 2024. Model fires, not ignitions: Capturing the human dimension of global fire regimes. *Cell Reports Sustainability*, 1(6).

L257-258. I find it quite surprising, if not unlikely, that high mitigation rather than low mitigation will lead to the most intense fires. [Same for “though less destructive” on L263]. Why is this the case?

This is a question of fuel loading. Under the high mitigation scenario, a greater accumulation of fuel is stimulated by elevated CO₂. Under the low mitigation scenario, the same process is in play – but increasing temperature and dryness become a limiting factor on vegetation productivity, reducing the accumulation of fuel overall. It is important to note, however, that even if simulated fires are globally less intense under the low mitigation scenario, more fires do still occur in regions that do not experience intense wildfires today..

We have added the following to the discussion:

“Although it may be surprising that a high mitigation scenario leads to more intense wildfires than a low mitigation scenario, this result reflects increasingly limited fuel loads under the greatest levels of warming. Indeed, whereas productivity change was the most important reason for increasing fire intensity under a high mitigation scenario, this changes to dryness under a low mitigation scenario, constraining fire intensity. Furthermore, although the overall fire intensity is lower under a low mitigation scenario, there are more regions that experience fire events, meaning that overall more intense fire events will be observed.”

L577: this URL for data and code doesn't work for me; need to provide the DOI version, not the link with you logged in because it takes me to the projects page.

L585-586. Same comment as last one.

Addressed above.

L589. A diagram of the workflow of the whole project would be helpful.

This flowchart has been added to the introduction.

L607: I don't understand the ignition threshold; more detail needed, as well as rationale for how it solves the problem. L608: I don't understand why you convert $BA < 0.0011$ to 0, when they are both valid numbers for BA.

Addressed above.

L628-636: I find this quite hard to follow and think it needs to be explained more clearly. I was under the impression the GLMs were already fitted, and that you were projecting them to future conditions. Here it gives the impression you are refitting them, sequentially excluding predictors. I don't get why you would refit the GLMs, or how this actually helps determine which variables are driving increases and decreases. Couldn't the partial effects of the variables show that?

We did refit the GLM models excluding one predictor at a time, re-running the 2100 realistic future. The aim of refitting the models was to identify which excluded variable caused the greatest change in each individual grid cell between in the simulated patterns of burnt area, fire size and fire intensity when that variable was excluded. The aim was to identify differences in

the spatial importance of each predictor. This information cannot be obtained from the partial effects alone. We have added this paragraph to the manuscript:

*“This provided 40 experiments, the spatial patterns of which were compared with a modern-day simulation based on the 2010-2015 period. We explored which predictors were driving changes between each experiment and the modern-day simulation by excluding one predictor at a time from the GLM models, re-running all the future experiments, and identifying which excluded variable caused the greatest change in the burnt area, fire size, and fire intensity respectively when compared to the modern-day simulated change in each grid cell. Although the GLM models provide information on the most influential variables globally through the individual *t*-values of each fitted relationship as through the partial residuals, this iterative method provides a way of quantifying which variable was responsible for the most change within each grid cell between each experiment and the modern-day. This allowed the spatial patterns of the variables responsible for driving the most change to be mapped. Similar methods have been used previously (e.g. Forkel et al. 2019). The same method was also used in a previous analysis on past climate change (Haas et al., 2022). By comparing these results to the burnt area, fire size, and fire intensity observed changes between the experiments and the modern-day simulation with the full GLM models we could determine whether the predictor was responsible for an increase or decrease in the given fire property.”*

Several times in the main body, it refers to “see Supplementary Materials” or similar. Please be specific, because the supplement is quite long.

We will refer to individual sections of the supplementary material to make this easier for the reader.

Response to the second round of reviewers' comments for the manuscript "*Wildfires on a changing planet*"

We thank all the reviewers for their additional comments on this manuscript. We have addressed the issues raised by all reviewer's point-by-point. Our responses are in **blue text** and the changes made to the manuscript in *blue italics*. The edits made to the original manuscript following the first round of reviews is implemented in *brown italics*.

Reviewer 1

Thank you for your detailed responses to the first-round review. The manuscript has improved, and I appreciate the additional clarifications and methodological adjustments.

However, I still have some major concerns regarding the modelling framework's ability to reproduce observed changes in burnt area, as well as a few instances where the results may still be somewhat overstated.

Anything I haven't commented on from the previous round I am happy to consider as fully addressed.

Major Concern: Trend Evaluation

One of the central themes of the manuscript is the projection of future fire trends. In several parts of the discussion, future changes are linked to the current global declining trend in burnt area (e.g., "The current global trend of decreasing burnt area may continue under high-mitigation efforts but is unlikely to continue under low-mitigation efforts where changes in climate promoting wildfires overwhelm the effects of human activities in reducing wildfires, particularly in the tropics"). However, the current evaluation does not demonstrate that the modelling framework adequately captures global or regional trends in burnt area when compared with observations. The addition of the trend analysis is a positive step forward, but it reveals a key weakness in the framework: the model does not capture the global decline nor regional trends in burnt area. While I appreciate the authors' acknowledgment of this issue, the explanation provided does not sufficiently address the underlying causes. Specifically:

- As stated, the GLMs were designed to capture spatial patterns of fire regimes. However, since they are now being used to infer changes over time – which the model has not necessarily been optimised for, it is crucial to evaluate their ability to reproduce long-term trends. While I am less concerned about the global results, the model also seems to struggle to reproduce trends in burnt area in the regional breakdown.

We have conducted the following additional exploration and analyses:

- Extended the time frame from 2006-2015 to 2002-2019. This extended period represents the overlapping period of the four observational products used: the reprocessed version of burnt area from MODIS MCD64A1 Collection 6.1 dataset (Giglio et al., 2021), GFED5, the extended version of the Global Fire Atlas from 2002-

2021 (Andela et al., 2024), and the MODIS Collection 6.1 active fire dataset as downloaded from the FIRMS LANCE server.

- Computed the interannual NME scores (step 2) for each fire property and year, globally and for each GFED region.
- Conducted a sensitivity analysis over 2002–2019 by varying different groups of predictors—climate, gross primary productivity (GPP), natural vegetation, and human activity (roads, crop area, and population density)—to assess their individual contributions to observed global trends in burnt area.
- As part of this analysis, we ran a simulation in which all predictors varied interannually *except* the human activity predictors, which were held constant. This allowed us to isolate the effect of changes in human activity on burnt area.
- Calculated anomalies by taking the difference between the predicted burnt area when *only human activity predictors* varied and when *only other predictors* (climate and vegetation) varied. These anomalies were added to the yearly predictions from the full model (with all predictors varying), effectively applying a *within-model bias correction*.
- This approach allowed us to disentangle the contribution of human activity to burnt-area trends from that of climate and vegetation, while remaining consistent with the model structure and inputs. By doing so, we tested the hypothesis that the model underestimates the influence of human activity— which may explain its failure to fully reproduce the observed declining trend in burnt area.
- We applied the mean fractional anomaly calculated over the 2002-2019 period to both the RCP2.6 and RCP6.0 realistic simulations to assess the effect of underestimating the influence of human activity on the future projections.

The details of these analysis are provided below and will be presented in a supplementary section. These analyses demonstrate that the GLMs can adequately capture the sensitivity of burnt area, fire size and fire intensity to the driving variables, although the models underestimate the influence of human activity on burnt area. This underestimation does not affect our principal findings. We have reworked the third paragraph of the discussion to address these various concerns:

Line 268 - *“This analysis focuses on mean long-term background rates of fire properties. The models were explicitly designed to capture the longer-term sensitivity of global mean burnt area, median fire size and median fire intensity to large-scale changes in climate, vegetation, land cover and human activity, rather than interannual variability or short-term trends. This study thus relies on a space-for-time substitution (Pickett, 1989) whereby changes in the spatial patterns of burnt area, fire size and fire intensity are used to infer how these fire properties may change under substantially different conditions. This approach was chosen because the temporal availability of fire data is currently limited, and the controls on the interannual variability of fire are not well understood (20, 92). Furthermore, these models have been shown to realistically simulate global burnt area under the radically different conditions of the Last Glacial Maximum (23). This gives us confidence in their ability to respond to large-scale changes in forcing.*

A consequence of the space-for-time substitution approach, however, is that the models may not capture all aspects of interannual variability, nor extreme fire behaviour. Extreme fire events may increase in frequency in the future (12), particularly in regions where conditions become drier (36). However, given the relatively small number of extreme fire events in the global record, they remain challenging to model (4,37). Analysis of the interannual performance of the GLMs over the last two decades highlights that the models used here are not sensitive enough to changes in land use and other aspects of human activity. We suggest that this is in part because the human processes modulating burnt area are sub-grid cell processes not captured at the relatively coarse spatial scale adopted here. More research is needed into constraining the influence of human activity on wildfires at sub-grid cell scales. Nevertheless, the fact that there is a large, modelled decrease in global burnt area when only changes human activity are considered, and that no global decline is observed when only climate and vegetation are considered, are strong results. Furthermore, artificially increasing the models' sensitivity to human activity did not substantially alter the conclusions presented here, highlighting the robustness of our findings.”

Given the emphasis on projecting future fire activity, a more rigorous assessment of why the model fails to reproduce historical trends is necessary before publication. Without a more thorough investigation, there is a risk that the modelling framework may not support the conclusions regarding future trends. The authors should conduct a more detailed evaluation of the divergence between observed and simulated trends, explaining where the disagreement originates—e.g., whether observed decreases in trends in some regions are influenced by land use effects, which, while the model should capture, are less relevant for this study which primarily focuses on climate-driven fire trends. Evaluation of the changes in fire size and intensity, more detailed regional break down to show where observation and model may disagree, etc, may all help too. Additionally, a clearer discussion of the limitations of the modelling approach will probably be needed to ensure that future projections are better substantiated.

We have conducted a further analysis on the sensitivity of the global models to climate, vegetation, land cover and human activity from 2002-2019 (the overlapping period of the observational data for burnt area, fire size and fire intensity). The key reason that the burnt-area model does not reproduce the declining trend adequately is that it is insufficiently sensitive to land-cover change and human activity (represented by cropland cover change, road density changes and human population change). A very strong declining burnt-area trend (double the observed trend, see Table S7) is modelled when only these predictors are varied. There is no clear trend in either fire size or fire intensity over the same period in the observational products, or in the predicted outputs (see Table S7). We have added the sensitivity analysis to the interannual variability plot and discussion.

Line 1067 - “ 1. BA inter-annual variability performance and trends

The modelled mean annual burnt area from 2002-2019 was 755 Mha, intermediate between the values of 418 Mha for the MODIS BA (Giglio et al., 2021) and 769 Mha for the GFED5

(Chen et al., 2023) burnt-area data products. The model captured the global and regional observed year-to-year variability in global burnt area reasonably well; however it failed to capture the declining global trend over the period of 2002 to 2019 (See Fig. S1 and Fig S2). We investigated what caused this divergence in the predicted and observed trend by conducting four sensitivity analyses: 1) holding GPP, GPP seasonality, tree, shrub and grass constant over the period (GPP and natural vegetation sensitivity) and varying all other predictors, 2) holding DD, DD seasonality, VPD, DTR and wind constant over the period (climate sensitivity) and varying all other predictors, 3) holding everything constant except for cropland cover, road density and population density (GPP, natural vegetation and climate sensitivity) and 4) varying only cropland cover, road density and population density (human activity sensitivity) and holding all other predictors constant (see Fig. S1, Fig. S2 and Table S1).

Figure S1. Global 18-year trend in burnt area (BA) and fire size (FS) in km^2 and fire intensity (F) in Wkm^{-1} during 2002–2019. Predicted trends where all predictors were varying are shown in red, and observations are shown in black. The trend in the GPP and natural vegetation sensitivity run is shown in green, the trend in the climate sensitivity analysis is shown in orange, the trend in the GPP and natural vegetation and climate sensitivity run is shown in dark green and the trend in the human activity sensitivity analysis is shown in purple.

Table S1. NME scores of trends in interannual variability (step 2) for modelled burnt area against FIRMS MODIS and GFED5 for the 2002 to 2019 period.

Observational Product	Burnt Area			Fire Size		Fire Intensity	
	Predicted	MODIS	GFED5	Predicted	GFA	Predicted	MODIS
	Trend (km²)	NME	NME	Trend (km²)	NME	Trend (W.km-1)	NME
All predictors	176.86***	1.51	1.45		0.87	-0***	1.10
Fragmentation and human activity only	-248.72***	0.87	0.91	0	1.11	0	1.00
Climate predictors only	83.74***	1.17	1.15	0*	1.81	0	0.90
GPP and natural vegetation only	87.86***	1.17	1.25	0	1.00	0**	1.00
GPP, natural vegetation and climate only	20.07***	1.65	1.55	0*	4.72	-0***	1.10

The trend in fire size was best represented when changes in all predictors were applied; the trend in fire intensity was best represented when only changes in climate predictors were applied. The declining global trend in burnt area is modelled when only changes in human activity are considered, We hypothesize that the failure to fully reproduce the observed declining trend in burnt area when all predictors are considered is due to the model underestimating the influence of human activity. We tested this hypothesis by performing a within-model bias correction to assess the impact that this underestimation may have on the overall results. The contribution of human activity to the trend in burnt area was quantified by computing the difference between the predicted burnt area when only human activity predictors varied and when only the non-human predictors (climate and vegetation) varied. The resulting anomalies were added to the yearly predictions from the full model (with all predictors varying). This approach allowed us to disentangle the contribution of human activity to burnt-area trends from those of climate and vegetation, while remaining consistent with the model structure and inputs. Trend analysis shows that applying this method dramatically improves the model's performance to reproduce regional burnt area trends, suggesting that we can account for the contribution of human activity to burnt area after a suitable correction (see Fig. S3). However, the modelled total global burnt area was thereby reduced to 343 Mha, a 54% reduction.

Table S2. Interannual trends in burnt area and NME scores (step 2) against observations for the original model and the model with the within-model bias correction

GFED Region	Original BA			Corrected BA		
	Predicted trend (km ² -yr ⁻¹)	NME step 2 (GFED5)	NME step 2 (MODIS)	Predicted trend (km ² -yr ⁻¹)	NME step 2 (GFED5)	NME step 2 (MODIS)
BONA	1047384.58	1.37	1.52	1643902.86	0.99	0.97
TENA	35.19	4.10	4.70	-41.77	2.17	2.29
CEAM	1269593.78	1.10	1.87	1842987.17	1.08	1.26
NHSA	37.89	1.61	1.76	-51.67	1.46	1.66
SHSA	108078.38	1.48	1.53	24212.08	0.87	1.09
EURO	11.33	1.91	3.36	1.92	1.19	1.50
MIDE	116160.18	0.80	1.08	142855.26	1.17	1.28
NHAF	9.55	1.15	1.27	-4.05	0.84	0.72
SHAF	214827.38	2.14	1.64	282555.94	1.15	0.76
BOAS	1.53	1.18	1.33	-10.30	1.02	1.03
CEAS	95469.28	1.11	1.12	13166.16	0.96	0.94
SEAS	-0.49	0.90	1.03	-1.27	1.09	1.54
EQAS	477143.27	0.48	0.58	-20969.65	1.40	1.54
AUST	17.95	0.94	1.00	26.86	1.13	1.13

Performing this correction for the influence of human activity on fire properties had the largest effect on burnt area. Neither FS or FI were strongly influenced by the correction (see Table S3).

Table S3. Interannual-trends in fire size and fire intensity and NME scores (step 2) against observations for the original model and the model with the within-model bias correction

GFED Region	FI NME score (step 2)	FI corrected NME score (step 2)	FS NME score (step 2)	FS corrected NME score (step 2)
BONA	2.46	1.11	1.00	1.00
TENA	1.41	1.02	0.92	0.99
CEAM	1.08	1.01	0.97	1.01
NHSA	1.26	0.98	0.95	1.01
SHSA	0.78	0.98	1.04	0.91
EURO	0.88	0.92	1.07	1.01
MIDE	1.23	1.02	0.94	1.08
NHAF	1.29	1.04	1.04	1.08
SHAF	0.97	0.94	1.03	1.00
BOAS	1.08	1.03	1.11	1.01
CEAS	1.05	1.00	1.07	0.98
SEAS	1.06	1.03	1.00	0.99
EQAS	1.04	0.96	0.99	0.99
AUST	1.02	1.03	1.03	0.94

– Corrected BA – GFED5 – MODIS – Predicted BA

Supplementary Figure S5. Trends in observed and modelled burnt area for all GFED regions over the 2002-2019 period for the original model (in red) and the corrected BA (in purple).

To assess the impact of accounting for this human signal in the future, we computed the fractional anomaly (mean absolute anomaly divided by the mean predicted BA from 2002-2019 in each grid cell) and added it to the realistic RCP2.6 and RCP6.0 realistic simulations, creating a scaled anomaly (see Fig S3) such that:

Future corrected = Future predicted + Future anomaly , where

*Future anomaly = Future predicted * Scaled anomaly and*

Scaled Anomaly =

$$\frac{(\text{Modern predicted}_{\text{human activity only}} - \text{Modern predicted}_{\text{GPP, vegetation and climate only}})}{\text{Modern predicted}}$$

Supplementary Figure S6. Anomaly between the original and corrected predictions applied to the modern, high mitigation and low mitigation climate change scenarios for BA, FS and FI.

As expected, correcting the predictions in this way lead to large reductions in burnt area under all three scenarios across much of the globe. Increases in fire size and intensity in the northern high latitudes were also observed. Whilst the decreases in burnt area were somewhat larger in the future scenarios, the spatial patterns remained consistent. Accounting for this effect reduced the overall amount of burnt area (see Table S2) but the overall trends remained unchanged, with a continued decrease under the high climate-change mitigation scenario and an increase under the low climate-change mitigation scenario. The amplitude of these trends differed between the models, and overall, there was a large reduction in the total annual global burnt area for all scenarios compared to the original model (Table S3). Nevertheless, this exercise gives us confidence that the models’ underestimation of the human signal does not affect the main conclusions of this work.

Table S4. Predicted global annual burnt area (BA) under present conditions, the high climate-mitigation experiment and the low climate-mitigation experiment, for the original model and with the within-model bias correction applied

	Original BA	Corrected BA
--	-------------	--------------

BA (Mkm²)	Predicted burnt area	Percent change	Predicted burnt area	Percent change
Modern	4.06		2.48	
High climate mitigation				
GFDL-ESM2M	3.05	-24.89	2.36	-5.02
IPSL-CM5A-LR	3.9	-3.84	2.11	-15.07
HadGEM2-ES	3.24	-20.19	1.9	-23.59
MIROC5	3.23	-20.34	1.86	-25.11
Low climate mitigation				
GFDL-ESM2M	6.1	50.31	3.48	38.25
IPSL-CM5A-LR	6.39	57.46	3.57	43.85
HadGEM2-ES	6.49	59.81	2.24	-9.69
MIROC5	4.62	13.9	2.71	9.02

- The rebuttal points out that GFED5 shows a significant decrease in burnt area and GFED4s shows a non-significant decrease and uses this to infer observational disagreement. This does not constitute a valid statistical argument or evidence of observational disagreement. Rather, it highlights a change in the confidence of the trend, which is not surprising given the updated nature of GFED5. Additionally, it is important to consider that over a longer time period than the 10 years considered here, studies have shown that GFED4s does indeed indicate a significant decline in burnt area. Moreover, there are trend benchmarking metrics that can account for uncertainty in observed trends, should the authors feel this is an important consideration.

We take the reviewer’s point and have removed this argument from the manuscript. We have now extended the analysis to cover the overlapping period of the three fire datasets, 2002-2019. We have also redone the trend analysis on a longer period and using the GFED regions, as this is more consistent with other studies (e.g. Burton et al., 2024).

- The argument that the modelling framework reproduces changes on paleo timescales is useful, but it would be more impactful as an additional line of evidence. It would be more compelling if the model first demonstrated that it could reproduce contemporary changes.

The GLMs were designed to capture the spatial patterns of fire regimes given a background climate and vegetation state. The models were constructed from a seasonal climatology to eliminate the effect of inter-annual variability and show the spatial distribution of fire on Earth given a specific climate forcing. Thus, the models are expected to capture the longer-term sensitivity of global mean burnt area, median fire size and median fire intensity to large-scale changes in climate, vegetation, land cover and human activity, rather than the interannual variability of these fire properties. The fact that the models have been shown to realistically simulate global fire regimes under the radically different conditions of the Last Glacial Maximum (Haas et al., 2023) is thus a key test of their performance. The global and regional trend analysis serve as an additional argument to show that, whilst the magnitude of change may not be fully captured, the models are able to respond appropriately to contemporary changes in climate, vegetation and land cover (see details of the global and regional trend

analysis and sensitivity testing above). We have added some argument to the first paragraph of the discussion addressing this (see above).

For evaluating model trends in the context of high observational trend uncertainty, I recommend consulting Burton et al.'s (2024) supplement, which includes methods for this purpose. While I was involved in that paper (and I may have suggested it to many of my own papers already), I suggest it as a starting point rather than something that must be cited. There may well be more straightforward benchmarking techniques. Additionally, Spuler et al. (2024) might also provide useful insights.

Burton, C.A., Kelley, D.I., Burke, E. et al. Fire weakens land carbon sinks before 1.5 °C. *Nat. Geosci.* 17, 1108–1114 (2024). <https://doi.org/10.1038/s41561-024-01554-7>
Spuler, F. R., Wessel, J. B., Comyn-Platt, E., Varndell, J., & Cagnazzo, C. (2024). ibicus: a new open-source Python package and comprehensive interface for statistical bias adjustment and evaluation in climate modelling (v1. 0.1). *Geoscientific Model Development*, 17(3), 1249-1269.

We believe the analysis presented above provides a sufficient and robust assessment of the models' strengths and limitations. Given that the models were not designed to reproduce inter-annual variability, we do not believe that assessing the uncertainty in the simulated trends is necessary.

Other Key Comments

1. Use of ISIMIP2b Instead of ISIMIP3b.

The justification for using ISIMIP2b over ISIMIP3b looks great. One potential addition (if the authors find it useful) could be a brief discussion on what improvements in ISIMIP3b would make it more suitable for future studies. For example, would additional land-use scenarios or longer bias-corrected periods enhance its utility? Though if you feel this is a distraction, then you don't have to include it.

The article is already quite complex, and the discussion covers many topics. We do not think a discussion of how to improve ISIMIP is central to our argument.

2. Uncertainty Representation.

The addition of uncertainty bounds in Supplementary Figure 3 is useful, though it appears that you are calculating a confidence rather than a predictive interval? Which is fine, just that the terminology may need reviewing.

These bounds represent the fitted value plus or minus the product of the critical value (from the t-distribution, here we are using 95%) and the standard error of the fit, which does represent confidence interval. The terminology has been changed.

While BA and FS seem to show a relatively narrow uncertainty range, the uncertainty for FI

appears to be larger. It would be helpful to quantify the implications for your results rather than visually assess the uncertainty. Could you use these uncertainty estimates to indicate statistical significance—perhaps by masking non-significant trends in Figure 2 or adding error bars to Figure 3?

Figure 2 already implicitly masks non-significant trends, as it only shows changes of over 5% (based on the lowest value in the predictive interval). Given that we are simulating the difference between two periods and not time series, we believe masking out areas where change does not exceed 5% accounts for some of this uncertainty. We have added error bars to Figure 3.

The authors acknowledge the limitations of plant functional type classification in their rebuttal. A short addition to the main text or methods would be helpful in clarifying this. It would be helpful to explicitly reference Supplementary Figures 6 and 7 in the main text, with a brief mention of potential limitations in biome-specific confidence levels.

We have added this paragraph (which was part of the supplementary) to the main discussion:

Line 325 - *“In this analysis we consider differences between vegetation types through the distinction of fractional tree, grass and shrub cover only. These fractions are obtained by averaging values for each biome globally. This means we do not address the response to within-biome differences in different regions of the world. Whilst this could explain some of the regional differences observed, overall, it is unlikely that a substantial amount of information is lost through this method. Research has consistently shown that current, conventionally defined biomes are not representative of differences in fire properties, while other plant traits are not captured in these classifications — relating to flammability and recovery — may be more important (15, 48-49). In addition, predictors relating to vegetation amount have been shown to be more important in capturing burnt area than predictors relating to vegetation type (50).”*

3. Decoupling of Fire Property Clusters.

I remain concerned about the phrasing regarding the decoupling of fire clusters in the future. The current wording suggests that burnt area, fire size, and intensity are presently observed as clusters but may decouple over time. This framing is problematic because it implies an existing interdependence that will break down, whereas the models are explicitly independent by design. If the models are independent, referring to a "decoupling" of clusters could mislead readers into thinking that fire properties are currently linked in a way that is not reflected in the modeling framework. While all three fire properties share an ignition threshold, this does not create a structured interdependence between them regarding their response to environmental drivers. Instead, the results indicate that the relationships between these properties may evolve in the future due to their differing sensitivities to changing climate, land use, and vegetation conditions. A clearer way to convey this might be:

We have rephrased this discussion point in the manuscript:

Line 257 - "Previous studies have classified fire regimes under modern conditions by identifying distinct associations of fire properties in the observational record and assuming they are coupled by biophysical constraints (33-35). Whilst we model different fire properties independently, except for the constraint that fire occurrence is necessary for modelling fire size and intensity, our results suggest that unique combinations of fire properties observed today may not remain constant under a changing climate. Burnt area, fire size and fire intensity have different controls and the relationships between them may change in the future, reflecting differing sensitivities to environmental drivers. This result has implications for the distribution of fire regimes on Earth and implies that efforts to model future wildfire regimes should not assume that the relationships between different fire properties are fixed."

4. Threshold Selection for Ignition Probability.

The choice of the median value (0.0011) as the threshold for ignition seems reasonable, but could the authors provide a reference supporting this approach?

This issue was also raised by the third reviewer; please see response below.

5. Comparison to fireMIP models.

The latest FireMIP round is available at ISIMIP (<https://data.isimip.org/10.48364/ISIMIP.446106>), with summary NME scores in Table S3 of Burton & Lampe et al (2024) (<https://www.nature.com/articles/s41558-024-02140-w>). I appreciate that these results weren't available at the time of initial submission. That said, the broader comparison between statistical models trained on observations performing better than process-based models without direct training has already been well covered across multiple studies from many research groups. Given this, it's not too surprising that a GLM approach outperforms older process-based models for specific tasks like modelling spatial patterns of BA.

What might be more relevant is to focus on the strengths of your approach relative to all other methods—whether process-based (like fireMIP), statistical (BASE, MaxEnt approaches etc), or hybrid (ConFire, decision tree/RF etc). Highlighting aspects beyond just burnt area, such as fire intensity or size, is a great example already in the revised m/s.

Given that the reviewer does not find the comparison with the FireMIP models particularly helpful, we have removed this and focused on more detailed analysis of GLMs from 2002-2019 (shown above). We have added the following paragraph to the discussion on the model choice:

Line 306 - *“In addition, a major part of this analysis was disentangling the effects of independent variables in driving the change in BA, FS and FI. This is something that is better done through linear modelling than a machine-learning approach. Most process-based models, for example, rely on rate of spread equations to link fire starts to burnt area, and estimate fire intensity as the product of fire spread and fuel consumption (37-40), thus intrinsically linking fire numbers, size, burnt area and intensity. The predictions of machine-learning approaches do not make these implicit linkages but focus on emergent properties of the fire regime and generally include a large number of predictors, making them more difficult to interpret and not well suited for sensitivity analyses (8). The GLM approach adopted here, in which burnt area, fire size and fire intensity are modelled as a response to environmental factors independently of one another, provides a powerful way of testing the sensitivity of different fire properties to specific controls and combinations of those controls.”*

Re: “Empirical analyses have considered vegetation and human impacts on fire risk and are being used in the Inter-Sectoral Impact Model Intercomparison Project (ISIMIP) to assess potential future wildfire regimes under climate change (16, Jones et al., 2022)” I think reference to Jones et al. (2022) should be corrected to Jones et al. (2024). <https://doi.org/10.5194/essd-16-3601-2024>

We have changed this reference.

7. New Figure 1 is very helpful but also feels like a supplementary figure.

Other reviewers asked for this figure to be in the main text.

Reviewer 3:

The authors have generally done a good job responding to reviewer comments. I appreciate the consideration of multiple metrics of fire activity.

The main thing that I think could be improved relates to the complexity of the experiments: with multiple moving parts, it makes it challenging for readers to discern what drives the broad-scale results. A key issue lies in comparing present-day conditions with future scenarios, as these involve simultaneous changes in input variables beyond climate mitigation—such as population densities, land cover, and other factors.

The concern about simultaneous changes in input variables is why we have now made sensitivity experiments holding climate, vegetation, and human activity constant, allowing us to identify which predictor is driving the largest changes in fire activity. These sensitivity

analyses are a key strength of the paper, as they show that the impacts of climate, vegetation and human activity change under different climate change scenarios. We have reworked the results section of the manuscript, so that we discuss the changes and drivers of burnt area first and then continue by discussing the changes and drivers of fire size and fire intensity. We hope that this will make it easier for the reader to discern what is driving the changes. This reorganisation means the results section reads as follows:

Line 100 - “*Burnt Area change under high climate change mitigation*”

Under ambitious climate mitigation, the annual global burnt area is reduced from 406 Mha under present conditions to 36-335 Mha by the end of the century (17-22% decrease for three of the four GCMs with one showing a negligible decrease, see Table 2 and S4). The global decrease in burnt area is primarily driven by decreases in tropical evergreen and deciduous forest, grassland, shrubland and savanna, which account for 83% of global burnt area under modern conditions but only 77-81% at the end of the 21st century (see Table 2). The key driver of the reduction in tropical burnt area, particularly in Africa, India and South-East Asia, was landscape fragmentation driven by increased human activity, as measured by road density and cropland cover. There was no modelled global decline in burnt area when socioeconomic variables were held constant: highlighting the importance of human activities in reducing burnt area (see Supplementary Section 7). However, human activity does not entirely explain the modelled reduction in burnt area. Climate changes, for example in Central America and the Amazon basin (primarily increasing number of dry days and decreasing dry-day seasonality), and vegetation changes (decreasing gross primary production, GPP, and grass cover), also drove a reduction in wetter tropical regions of South America (see Fig 3). When holding socioeconomic variables constant, reductions in burnt area were still observed in tropical grasslands, xerophytic shrublands, evergreen and semi-deciduous forests because of increasing dryness (see Supplementary Section 4, 5 and 6). The modelled decrease in burnt area does not occur homogeneously across all tropical regions, and an increase in burnt area is simulated in the tropical regions of South America (see Table 2 and Fig 2). In drier tropical biomes, increased dryness and GPP also led to increases in modelled burnt area, explaining some of the variability between tropical regions.

Whilst the global signal is dominated by the reduction of burnt area in tropical regions, the number of grid cells overall in which the probability of burning exceeded our ignition threshold was 27-30% greater by the end of the 21st century than under modern condition. Most (87-88%) of this increase occurred in the northern extra-tropics. In these regions, the probability of burning increases above a threshold under which no wildfire activity is expected to occur. Thus, these increases represent an encroachment of wildfires into areas where previously there were none. This is the case for example in Scandinavia and some regions of Siberia.

Increases in burnt area in temperate and boreal forests, and in tundra (Fig 3), were driven by decreased precipitation (as reflected by an increase in the number of dry days) and increased atmospheric dryness (VPD). This was observed in both climate-only and CO₂-only sensitivity experiments (see Supplementary Section 5 and 6). However, these regional increases in burnt

area were not sufficient to counteract the impact of reductions in the tropics on the global decline in burnt area (Fig 2, Table 2).

Burnt-area change under low climate-change mitigation

In stark contrast to the high climate-change mitigation scenario, under low climate-change mitigation there is an increase (18-60%) in modelled global burnt area. Increased burning is seen across all vegetation types, even in the tropics (Fig 2). Human-driven reductions in burnt area in tropical regions are offset by the combined effects of changes in climate and CO₂. In temperate regions, vegetation types where burning increased under the high climate change mitigation scenario showed an even larger increase under the low climate- change mitigation scenario (Fig 2, Table 2).

Changes in climate are more important than changes in vegetation and human activity in driving burnt area increases in the low climate-change mitigation scenario, although vegetation changes still contribute to increasing burnt area in sub-Saharan Africa (Angola, Zambia, Malawi, Mozambique) (see Fig 3 and Supplementary Section 5). Human-driven reductions in burnt area in tropical regions are offset by the combined effects of changes in climate and CO₂ (see Supplementary Section 6 and 7). The role of CO₂ is important, as the increases in burnt area do not occur in response to changes in climate alone (see Supplementary 4). Increases in dryness and grass cover drove an increase in modelled burnt area in North America and increases in GPP drove an increase over much of Eurasia (see Fig S12-15). Increased road density in previously fire-free regions also caused increases in burnt area in Europe and the east coast of the United States across all four models (see Fig S12-15). This effect does not occur under the high climate-change mitigation scenario. Increased fire size in these northern regions was driven by increasing dryness and diurnal temperature range (DTR).

When only socio-economic changes are considered, we model large decreases in burnt area in tropical regions of Africa and India, but much smaller reductions in South America and South-East Asia (see Supplementary 7). This probably reflects deforestation fires in the Amazon and Indonesia which, though reduced under the future socio-economic scenario, still occur (deforestation fires are captured implicitly through cropland cover and population density in the models).

Fire size and intensity changes under high climate-change mitigation conditions

Under the high climate-change mitigation scenario, global fire size and mean global fire intensity are increased compared to modern conditions (by 23-36% and 2-5% respectively, see Table 2 and Table S4). Global increase in fire size and fire intensity primarily reflects increases in the northern extra-tropics (Fig 2). The largest increases in fire intensity and fire size occurred in boreal forests and shrub tundra north of 60°N (see Fig 2 and Fig 3). Increases in fire size and intensity were driven by changes in climate factors (increasing wind strength and dryness) and by human activities. Modelled increases in fire intensity in these regions were driven by increased GPP and dryness. Larger fires in South America were driven by increased dryness in the tropical rainforests and increased cropland and wind in drier, tropical savannas.

A decrease in fire intensity is modelled in the tropics (Fig 3). Fire size also decreases in tropical biomes, though large increases are observed in South America, dominating the signal when looking at trends within these biomes (see Fig 2). Modelled decreases in fire size and fire intensity in tropical Africa were primarily driven by changes in human activity (see Fig S12-15). Increased dryness also caused an increase in vegetation fragmentation, reducing both fire size and fire intensity in these regions (see Fig 2 and 3).

Fire size and intensity change under low climate-change mitigation conditions

Under the low climate-change mitigation scenario, both global fire size and fire intensity are reduced compared to a high climate-change mitigation scenario. Under the low climate-change mitigation scenario, global mean fire size increased 19-32% compared to today while the trend in global mean fire intensity was opposite, with a global decrease of 1-12%. Although the increase in fire intensity was somewhat smaller than in the high-mitigation scenario overall, this was primarily driven by decreases in temperate grasslands and forests, including conifer forests, compared to the high-mitigation scenario (80% decrease) as well as changes in tropical regions. The overall modelled decrease in fire intensity in tropical regions doubled (49%) compared to the high climate-change mitigation scenario, except for tropical grasslands, where fire intensity increased (see Fig 2). Whereas increased GPP was the major cause of increased fire intensity in the high-mitigation scenario, changes in precipitation (as reflected in the number and seasonality of dry days) and increased atmospheric dryness were the most important causes of increased fire intensity in the low-mitigation scenario (see Fig 3 and Fig S12-15) and this shift is likely the cause of the reduction in fire intensity compared to the high-mitigation scenario.. The largest increases in fire intensity and fire size compared to today were concentrated in taiga forests and tundra above 60°N. The increase in fire size was substantially higher than in the high climate change mitigation scenario across all northern biomes including temperate forests (50% increase).”

For example, present day global population is roughly 8 billion whereas SSP2 simulates roughly 9.5 billion I believe This complexity makes it difficult to intuitively understand or visualize the differences between present-day and future scenarios. For example, the quite unintuitive prediction that a 2°C warming scenario leads to less fire than present-day (lines 100–102) might reflect changes in population or land use rather than the climate mitigation itself. This means comparisons between scenarios (e.g., present-day vs. mitigation) are not purely driven by changes in climate but also by shifts in human and land-use factors. It would be good to explain that when introducing that finding.

The reviewer is correct, the decrease in burnt area is driven by population and land use change. This is now clearly stated:

Line 106 - *“The key driver in the reduction of tropical burnt area, particularly in Africa, India, and South-East Asia was landscape fragmentation, primarily driven by increased human activity, as measured by road density and cropland cover. There was no modelled global decline*

in burnt area when socioeconomic variables were held constant highlighting the importance of human activities in reducing burnt area (see Supplementary Section 7).”

We have referenced the Supplementary section which contains the relevant the sensitivity maps supporting these findings.

Another example: in response to my earlier comment about the surprising prediction that high mitigation leads to more intense fires than low mitigation, you explained that this is due to fuel loading and shifts in productivity-dryness limitations. I think this challenge would benefit from clearer visualization and explanation of how all of the drivers of fire differ under the scenarios (incl present day). For example, supplementary tables and maps showing how productivity, dryness, population size, land cover and other variables differ across scenarios would make it easier for readers to grasp these effects. I think this additional context would make it quite a lot easier to interpret the model results.

We have added maps in the Supplementary (Section 8) to show how the different input drivers change in each model and for each predictor between the present day and the high and low climate mitigation scenario and have added references to these supplementary figures when discussing these issues.

Line 1417 - **“Supplementary Section 8: Change by the end of century in the climate, vegetation and human activity under the high and low climate change mitigation scenarios**

Supplementary Fig S26. Percentage change in all predictors between the modern-day and the end of century high climate change mitigation scenarios for GFDL-ESM2M climate model outputs.

Supplementary Fig S27. Percentage change in all predictors between the present-day and the end-of-century low climate-change mitigation scenarios for GFDL-ESM2M climate model outputs.

Supplementary Fig S28. Percentage change in all predictors between the present-day and the end-of-century high climate-change mitigation scenarios for IPSL-CM5A-LR climate model outputs.

Supplementary Fig S29. Percentage change in all predictors between the present-day and the end-of-century low climate-change mitigation scenarios for IPSL-CM5A-LR climate model outputs.

Supplementary Fig S30. Percentage change in all predictors between the present-day and the end-of-century high climate-change mitigation scenarios for HadGEM2-ES climate model outputs.

Supplementary Fig S31. Percentage change in all predictors between the present-day and the end-of-century low climate-change mitigation scenarios for HadGEM2-ES climate model outputs.

Supplementary Fig S32. Percentage change in all predictors between the present-day and the end-of-century high climate-change mitigation scenarios for MIROC5 climate model outputs.

Supplementary Fig S33. Percentage change in all predictors between the present-day and the end-of-century low climate-change mitigation scenarios for MIROC5 climate model outputs.”

Clarifying “mitigation”: Since many readers will approach this paper as fire ecologists or practitioners, it’s crucial to explicitly clarify what “mitigation” refers to. In this paper, you generally and implicitly use it to mean climate mitigation, but for fire ecologists, the term typically implies vegetation management to mitigate fire risk. I recommend adding “climate” before “mitigation” throughout the manuscript to avoid confusion, especially since you’ve occasionally used “mitigation” to mean fire mitigation (e.g., L56).

We now refer to these consistently as either “*high climate -change mitigation scenario*” or “*low climate-change mitigation scenario*”.

In your response to my request for clear definition of the units of the response variables, you mentioned adding a table for response and input variables. However, the response variables still do not seem to be there and remain somewhat vaguely defined. It is essential to clearly define them, including their units, the time scale of measurement, and spatial resolution. Fire intensity seems to be undefined entirely. I know BA is the proportion burned per month, but at what resolution (i.e., proportion of what)? Each variable needs clear definitions, units, and temporal dimension, so that it’s crystal clear to reader what the models themselves were actually modelling.

We have added the information below to the table in the supplementary:

Response variables	Abbreviation	Value	Transformation and link function
Burnt area	BA	Monthly mean burnt area fraction	None (logit link function)
Fire size	FS	Monthly median fire size (km²)	Min-max normalized (log link function)
Fire intensity	FI	Monthly median fire radiative power (FRP) divided by the square root of median fire size (W.km⁻¹)	Min-max normalized (log link function)

L816–818: I now understand your rationale for selecting the median predicted BA value (0.0011) as a threshold to determine whether modelled fire should occur in grid cells where fire had not been empirically observed. However, selecting the median value as a threshold means that half of the grid cells without observed fire will have fire assigned, which seems problematic. In this case, 26% of grid cells had an observed BA of zero, meaning approximately 13% will have been assigned as having fire even though they didn't, which represents a significant proportion. I suggest revisiting this threshold to ensure it aligns better with empirical observations. Rather than selecting the median value, you could use an approach that balances sensitivity and specificity (e.g., True Skill Statistic).

We conducted True Skill Statistic (TSS) for the different quantiles of the predicted burnt area values when the observed values of burnt area were zero. The median value resulted a good combination of high sensitivity (above 90% of fires that did occur are simulated) with medium specificity (0.49). While some other quantile values had higher sensitivity, they had lower specificity; similarly, values with higher specificity all had lower sensitivity (see Table below). Since the main concern in this analysis is to detect when burning could occur, we think that using a value with a high sensitivity but medium specificity is the most appropriate approach. This suggests the use of the median value of 0.0011 is a good threshold choice (see Table and Figure below). We equally conducted a sensitivity analysis to the threshold value, using both the 25th percentile and the 50th percentile.

We have modified the methods section to reflect this:

Line 837 - “Globally, 26% of grid-cells had an observed BA of zero (a total of 14,816 data points and associated fitted values). We took the median value (0.0011) of the fitted values in these grid cells to set the threshold for ignition. In grid cells in which burnt area is simulated to be lower than a fractional value of 0.0011, fire size and fire intensity were set to zero. We conducted a True Skill Statistic test (95-96), which balances sensitivity (the ability for the threshold to capture true positives) as well as specificity (how many false negatives the

threshold results in) to test the robustness of this threshold value. Since the main concern was to determine under which conditions burning could occur, the selected threshold needed to provide high sensitivity as well as balancing specificity. The median value of 0.0011 appears to be a good threshold choice, since it has sensitivity >0.9 and a moderate level of specificity (see Supplementary Figure S3 and Supplementary Table S2). We examined the sensitivity of fire size and fire intensity to the chosen threshold value through testing the 25th percentile and the 75th percentile. Overall, although the absolute values changed, the trends between modern, RCP26 and RCP60 did not. The uncertainty in the 75th percentile was much larger than in the 25th percentile and 50th percentile threshold values (see Supplementary Table S3).

Supplementary Table S1. Results of the true skill statistic

Percentile	Threshold value	TSS	Sensitivity	Specificity	maxTSS
0.1	0.00013	0.094512	0.994417	0.100095	1.094512
0.2	0.00027	0.184416	0.985105	0.199312	1.184416
0.25	0.00034	0.215538	0.979644	0.235894	1.215538
0.3	0.00047	0.266873	0.969356	0.297516	1.266873
0.4	0.00076	0.343016	0.944393	0.398623	1.343016
0.5	0.0011	0.406785	0.914749	0.492036	1.406785
0.6	0.0018	0.460293	0.857094	0.603199	1.460293
0.7	0.0027	0.482525	0.786251	0.696274	1.482525
0.75	0.00342	0.478716	0.728986	0.74973	1.478716
0.8	0.004	0.471159	0.687006	0.784152	1.471159
0.9	0.008	0.408702	0.514871	0.893831	1.408702

Table S2. Fire Size and Fire Intensity under present-day, RCP26 and RCP60 end-of-century conditions using a 25th percentile threshold, a 50th percentile threshold and a 75th percentile threshold.

Ignition threshold	Fire size	Fire intensity	Unburnt (%)
25 th quantile			
Modern	2.11 [1.98-2.24]	24.91 [24.02-25.77]	18.79 [18.18-19.53]
RCP26	2.81 [2.66-2.97]	24.69 [23.90-25.50]	9.60 [9.35-9.87]
RCP60	2.74 [2.59-2.92]	24.08 [23.31-24.87]	6.27 [6.09-6.43]
50 th quantile			

Modern	1.59 [1.44-1.74]	20.85 [19.97-21.80]	31.65 [30.36-32.98]
RCP26	2.68 [2.51-2.86]	22.68 [21.88-23.53]	13.04 [12.43-13.66]
RCP60	2.62 [2.45-2.78]	22.23 [21.41-23.06]	8.01 [7.67-8.42]
75 th quantile			
Modern	0 [0-0.44]	0 [0-13.61]	50.64 [49.37-51.84]
RCP26	1.89 [1.69-2.13]	18.20 [17.22-19.20]	25.44 [24.01-26.99]
RCP60	2.13 [1.95-2.32]	18.41 [17.55 - 19.29]	15.95 [14.73-17.31]

L43–44: “Physical indices such as measures of fire weather equally do not take account of the potential for human activities to mitigate wildfire risk.” Sure, but this work also does not evaluate fire mitigation.

This was poorly phrased. We were trying to make the point that studies have only looked at the impact of changes in climate indices (such as these physical indices) on wildfire occurrence, without considering the effects of land cover and human activity. These studies are then used to infer future changes and used as a baseline to develop mitigation strategies. The novelty of this study is that it accounts for the effect of changes in climate, landcover and human activity together and as such provides a more robust and consistent analysis of potential change in wildfire activity under end-of-century conditions. We have added the following sentence:

Line 43 - *“Analyses that consider only physical indices such as fire-weather indices are often used to assess future fire risk but do not account for other influences on wildfires such as vegetation and human activity (Jones et al., 2022).”*

L130: “this mitigation scenario” is unclear—specify which scenario you mean.
L137: Similarly, clarify what “this scenario” refers to, as the prior sentence mentions both scenarios.

We have reworked this paragraph as part of the revised results section (see above).

L281: Please provide specific references to the 95% CIs.

We have added the reference to the supplementary tables.

Line 318 - *“We have also characterized the uncertainty in this response by considering a 95% confidence interval on the modelled relationships (see Table S4-8), which shows that globally the uncertainties are relatively small.”*

L286–288: Consider dropping the sentence about overfitting. The argument is weak since machine learning models can be tuned to minimize overfitting, and the cited reference even points out that linear models can suffer from even worse specification bias.

We have dropped this sentence, as suggested, and reformulated this part of the argument as follows:

Line 322 - *“It has been argued that machine-learning models have greater predictive power than linear modelling tools, but they have limited ability to predict outside of the range on*

which they were trained. In addition, a major part of this analysis was disentangling the effects of independent variables in driving the change in BA, FS and FI. This is something that is better done through linear modelling than a machine-learning approach.”

L312: Spell out “PFT” at first use, as it may not be familiar to all readers.

We have now spelt this out, the sentence reads:

Line 342 - *“We were interested in producing a plant functional type (PFT) independent analysis, and the optimality-based productivity model used here was the only model that allows this (48).”*

Table 2 issue: There is no Table 2 in the main text (it goes from Table 1 to Table 3), but it is referenced in the text.

This was a typo, as there are only two tables. We have corrected the labelling and the references in the text.

As I noted last time, at several points, the text says "see Supplementary Material". This is too vague because the Supplement is about 50 pages long. Please be specific as it took me considerable time to locate the relevant part each time.

We recognise that the Supplementary Material is extensive and have now referenced the appropriate sections and figures we are referring to in the main text to make it easier for to locate the relevant part.

Point-to-point response to reviewers: Wildfires on a changing planet

Reviewer 1

Apologies for the slow review. The request came in the middle of summer, and a combination of annual leave, fieldwork, and conferences delayed my response. Also, I've spent the past couple of weeks working through the bias correction methods. While I suspect there may be some issues with the mathematics, I wanted to be thorough because I don't want the authors to undertake another major round of revisions or discard the hard work they've already put in unnecessarily. Especially because, aside from the model evaluation for burned area, I'm very impressed with the revised manuscript, and the evaluation appears to be in - and possibly beyond - the right direction.

Main concern: Bias correction

1. Purpose of the test: The main objective seems to be to show that the non-human driven component of the model is plausible (or the converse, as expressed by the authors, "we tested the hypothesis that the model underestimates the influence of human activity"). I don't think there is an absolute need to carry out a full bias correction, particularly for future projections where the human contribution has already been heavily simplified and the difference between scenarios is essentially due to non-human components. However, if an appropriate correction is applied, it certainly wouldn't harm the findings. But...

2. Problems with the current correction: I think (though might be wrong) the correction as implemented essentially doubles the human trend and removes the none-human trend. If I'm right, this is not a reliable test if the none-human component is already correct, and it is not a defensible approach for future projections. I might have misunderstood how this works, so I've attached my derivation (for historic evaluation only) to check whether my interpretation matches what the authors implemented.

Recommendation: If I'm wrong, feel free to clarify the bias correction and how it evaluates the model under recent changes in burned area. Otherwise, I would remove it and take a simpler evaluation approach using your new sensitivity experiments. I think that simply showing that your non-human sensitivity run yields plausible results comparable to independent analysis, while the human component is very flat, is enough in itself (both globally and for your sub regions).

Given that the reviewer is happy with the sensitivity experiments, we have removed any attempt at the bias-correction and have instead added this discussion of the relative contribution of the human-driven trend and the climate-driven trend from the sensitivity experiments in relation to the literature in the Supplementary:

Line 1148: "For burnt area, when only changes in human activity are considered, a negative trend of $-248.72 \text{ km}^2 \text{ yr}^{-1}$ is observed whilst when only climate changes are considered, a positive trend of $+83.74 \text{ km}^2 \text{ yr}^{-1}$ is observed. Over the 18-year period this corresponds to a total modelled decline of $\sim 4.5 \text{ Mha}$ from human activity and an increase of $\sim 1.5 \text{ Mha}$ from climate. It is worth noting that the absolute magnitudes of these trends are about an order of magnitude smaller than observed, likely reflecting simplifications in how the model represents fire-human-climate interactions. Nevertheless, the relative balance between human- and climate-driven trends is consistent with independent analyses. The sensitivity

analysis gives a relative balance of roughly 3:1, consistent with the literature, which suggests a relative balance of ~2.5:1. This balance can be derived from the ~-25% decrease in burned area from human drivers over 18 years cited in Andela et al., 2017 and the ~+16% increase from climate over 2003-2019 cited in Burton et al., 2024. Despite underestimating their strength, the relative trends in the human and non-human predictors of this model are plausible and aligns with previous work (Andela et al., 2017; Jones et al., 2022; Burton et al., 2024).”

But if you want to do something a bit clever, similar to the technique you’ve already started, I one suggestion is that you could obtain another point of evaluation by: assuming the non-human (climate/vegetation) component is correct, infer the implied human contribution to the trend directly from the full model, the non-human model, and the observations. The attachment has the math for this. The derivation is a bit long, and probably longer than necessary. But the actual comparison is very simple: essentially the difference between observed, modelled, and modelled-without-none-humans trends. This approach provides two clear evaluation points:

- Does the non-human component of the model behave plausibly? From the sensitivity runs.
- If the climate component is correct, what does the “corrected” human component look like, and is it consistent with independent evidence?

Both non-human and inferred human-driven trends could be compared to the standard literature already cited in the manuscript (e.g., Jones 2022; Burton & Lampe 2024; Andela 2017).

Either of these alternatives seem simple to implement and avoids the problematic assumptions of the current correction, while still addressing the underlying question: the plausibility of the model’s none-human-driven trends.

We thank the reviewer for this suggestion. However, since the reviewer has indicated they are satisfied with the sensitivity analysis as a validation test and given that this already provides a clear assessment of the relative contributions of human- and climate-driven trends, we have not included this additional analysis here. We agree it could be a valuable approach in future work but consider it beyond the scope of the present study.

After this, it’s mostly a matter of tightening the discussion language to make it clear this is a modelled (albeit heavily trained) system and that some unknowns remain in the response to human and bioclimatic variables.

We have added this sentence to the discussion to further emphasize this point:

Line 352: *“It is important to note that these results remain the outputs of a modelled system. Despite extensive model training and evaluation, uncertainties remain in the response of fire regimes to human, climate and vegetation changes. However, the relative balance of drivers identified here is consistent with independent analyses, suggesting that the model captures the key processes underlying observed trends, despite underestimating the sensitivity of the system to them.”*

This feels like minor revisions, but I’d suggest to the editor a final quick check with a reviewer (doesn’t have to be me) to ensure the literature comparisons are fully consistent.

Minor comments

- I may have missed it, but what trend evaluation (regression technique?) Did you perform to get your trend coefficients?

We have performed a linear regression. We have added this to the supplementary discussion:

Line 1125: *“A simple linear regression was performed to assess these trends.”*

- The trend table S1 shows zero trend in other fire metrics for the model result (which I *think* correspond to the time series in Figure S1?). While the trends appear very small, I’m surprised it’s exactly zero. Could you check? This might be an artefact of the regression technique (e.g., if outliers are downweighed), which is fine, but it would be good to note.

The values are not exactly zero, however, they become zero when rounded up. This is because many grid-cells have very small fire sizes and fire intensities, and that we are representing median values in fire size and intensity here. We have added the following clarification:

Line 1137: *“The trends in fire size and fire intensity are close to zero due to the distribution of these variables, with many grid-cells with very small values of median fire size and fire intensity. As such, the trends in the mean global values are very small but remain significant and are rounded to zero in the table. The trend in fire size was best when all changes in all predictors were applied and the trend in fire intensity was best when only changes in climate predictors were applied.”*

- Line 1167: these might be a small formatting problem here

This has now been resolved.

Reviewer 3

I am pleased to see a bit more explicit acknowledgement in the Results of the role of changes in human / socioeconomic variables in driving the predicted outcomes. However, I still believe there needs to be more effort on this point in other areas so that the results are not misinterpreted. For example:

- Abstract: says there will be generally reduced burning in the tropics. The way the abstract is worded, I think readers will interpret that as a result of changes in the climate scenarios, because there has been no mention yet that population and roads etc were also allowed to change in the predictions. The abstract would essentially lead a reader to think all of the changes are due to climate change and varying degrees of mitigation. Thus, in the Abstract, I think it should be explicitly stated that you compared two climate scenarios with X and X degrees warming, in which rising population and other socioeconomic variables were also projected, and that this particular effect of reduced burning in the tropics was strongly driven by those socioeconomic variables. At present, I think a reader would interpret the abstract as these effects all stem from climate mitigation differences.

We have reworded the abstract to read:

Line 14: *“We use global empirical models of burnt area, fire size and fire intensity to explore future wildfire trajectories under ~1.5 and 3-4 °C warming with middle of the road future socio-economic conditions. Even under ~1.5 °C warming we find a change in wildfire patterns by the end of the 21st century with reduced burning in tropical regions driven by changes in human activity but larger and more intense wildfires in extra-tropical regions driven by changes in climate and CO₂. With low climate change mitigation, burnt areas increase greatly across all vegetation types, overwhelming the current global decline.”*

- Introduction (last para): You explain that you compare two climate scenarios, involving 2 and 3-4 C warming. But the socioeconomic variables are just as important (even more so for some results), and they are only given a cursory mention. So, given their importance, I think it's absolutely imperative that you summarise those changes here too, e.g., under the middle of the road socioeconomic pathway in which human population and road density increase by X. Please explicitly state in the main text how much the population, or road density etc, is projected to increase under those scenarios.

We have added the following sentence:

Line 77: *“Under this scenario, population increases 9 billion by the end of the century. Most of this population increase occurs in Africa, with increases also observed in Australia, the Eastern United States and India. Decreases are observed in Brazil, Southeast Asia, China and Russia. Europe shows both increases and decreases in population (see S30-S33). Road density increases, with the largest increases in Africa and India. Cropland increases are concentrated in Europe, some regions of Eastern Africa, the Middle East and south-east Asia (see S30-33).”*

- Opening paragraph of the Discussion: “under both high- and low-climate change mitigation scenarios. Both scenarios show a reduction in burning in the tropics but increased burning elsewhere.” But it's not the climate mitigation that led to the predicted reduction, it's the simultaneous change in socioeconomics. I feel these sentences would lead a reader to believe the reduction in burning in the tropics is driven by the climate mitigation scenarios; please be explicit at this point that this effect was strongly driven by humans/roads.

We have changed this sentence, and it now reads:

Line 249: *“Model simulations show major changes in the geographic patterns of wildfire occurrence by the end of the 21st century, under both high- and low-climate change mitigation scenarios and middle of the road socio-economic changes. Both experiments show a reduction in burning in the tropics but increased burning elsewhere.”*

- I think it may help to add 2 panels to Fig 2, in which human activity variables are held at present values, so a reader can easily discern the separate effects of climate change and climate mitigation. That way, it will be more explicable why we see such large reductions in fire in much of Africa, for example. I believe that is already shown in Supplementary Figure S20. I think this is quite important because it helps explain some of the less intuitive findings in the Results.

We have redrawn Figure 2:

Fig 2. Regions of change (>5%) in burnt area (BA), fire size (FS) and fire intensity (FI) when all predictors are varied on the left with the high climate change mitigation scenario (top) and the low climate change mitigation scenario (bottom) and when all predictors are varied except for human activity on the right with the high climate change mitigation scenario (top) and the low climate change mitigation scenario (bottom). The tropics and extra tropics are delimited by dotted horizontal lines. Regions where all three properties increase are shown in red, regions where all three properties decrease are shown in dark blue (only increases of more than 5% in all three properties are considered).

Significant (but easily rectified) concern:

Fig 4 and L896-911: I have concerns about the method used to identify the dominant driver of projected change in each grid cell. As I understand it, the authors re-fit the GLM multiple times, each time excluding one predictor, and then compare the resulting projections to those from the full model. This appears to be intended as a sensitivity analysis i.e., identifying which variable drives the largest change in predictions for a given cell. However, in its current form, this approach does not cleanly achieve that aim. Specifically: (1) Apples-to-oranges comparison: Comparing predictions from different models with different predictor sets does not constitute a controlled sensitivity analysis of those predictors. It introduces confounding due to changes in model structure. (2) Lack of marginal effect isolation: This approach does not isolate the marginal contribution of the excluded variable. Instead, the observed differences in predictions may arise from a combination of the variable's absence (globally from the model, not cell-wise), shifts in the coefficients of other predictors, or even differences in overall model performance.

This makes interpretation difficult and potentially misleading, as the observed changes could reflect changes in the model or instability rather than the true influence of the omitted

variable. This is particularly problematic in the presence of variables with some degree of correlation (as is the case here), where excluding a predictor can redistribute explained variance among correlated variables. As a result, the spatial attribution of projected changes may reflect changes in the model rather than the independent contribution of each predictor.

My suggestion: I think that a more interpretable and statistically coherent alternative would be to retain the full GLM model and conduct a post hoc sensitivity analysis by perturbing one predictor at a time e.g., by permuting its values randomly, breaking its association with the response. This would allow the influence of each variable on future predictions to be assessed directly, without altering the fitted model structure, and would provide a clearer basis for spatial attribution of change in projected fire risk.

We thank the reviewer for raising this point. We agree that our initial approach risked mixing model structure changes with predictor sensitivity. We have now implemented the reviewer's suggested method of retaining the full GLM but permuting one predictor at a time, thereby breaking its association with the response while preserving model structure. However, we believe that this approach provides a way to quantify the sensitivity of each grid cell to each predictor, rather than providing the "dominant driver" of change in the projection for that grid-cell. This approach would therefore represent a shift in the question and interpretation. Instead, we have kept the GLM models fixed and multiplied the model coefficient for each predictor with the difference in the model inputs (transformed future experiment – transformed modern-day experiment value) for that predictor. This produced spatial maps of predictor individual contributions. We were then able to identify which predictor had the largest positive and negative contributors in each grid cell. This approach is analogous to the first-order term of the Stein–Alpert decomposition (Stein and Alpert, 1993), representing a simplified linear formulation that assumes additive relationships between predictors. This provided maps of the predictors most responsible for increases and decreases in burnt area, fire size and fire intensity between the modern-day and the future, allowing for a more diagnostic approach.

We have updated the methods section to read:

Line 924: *“We explored which predictors each grid cell was responsible for driving the largest increase and decrease in the fire property. For each predictor we calculated the difference in the model inputs (future experiment – modern-day experiment value) and multiplied this by the corresponding coefficient from the GLM. This approach is analogous to the first-order term of the Stein–Alpert decomposition (Stein and Alpert, 1993), representing a simplified linear formulation that assumes additive relationships between predictors. This produced spatial maps of predictor contributions, which allowed us to identify the largest positive and negative contributors in each grid cell. This provided maps of the predictors most responsible for increases and decreases in burnt area, fire size and fire intensity between the modern-day and the future. Although the GLM models provide information on the most influential variables globally through the individual t-values of each fitted relationship as through the partial residuals, this grid-cell sensitivity analysis provides a way of quantifying which variable was responsible for the most change within each grid cell*

between each experiment and the modern-day. This allowed the spatial patterns of the variables responsible for driving the most change to be mapped. Examining the sign of the difference allows us to determine whether the predictor was responsible for an increase or decrease in the given fire property.”

A change in this method also did not lead to major changes in the discussion, but it has been rewritten as follows:

Line 140: *“However, human activity does not explain the modelled reduction in the tropics entirely. When holding socioeconomic variables constant, reductions in burnt area were still observed in tropical grasslands, xerophytic shrublands, evergreen and semi-deciduous forests because of increasing dryness (see Supplementary Section 4, 5 and 6). Climate changes, for example in Central America and the Amazon basin (primarily decreasing dry-day seasonality), and vegetation changes (decreasing gross primary production), also drove a reduction in wetter tropical regions of South America (see Fig 3). Furthermore, even when considering changes in human activity, the decrease in burnt area does not occur homogenously across all tropical regions. For example, an increase in burnt area is simulated in the tropical regions of South America (see Table 2 and Fig 2). In drier tropical biomes, increased dryness and GPP also led to increases in burnt area, explaining some of the variability between tropical regions.*

Whilst the global signal is dominated by the reduction of burnt area in tropical regions, the number of grid cells overall in which the probability of burning exceeded our ignition threshold was 27-30% greater by the end of the 21st century than under modern conditions, and most (87-88%) of this increase occurred in the northern extra-tropics. These increases represent an encroachment of wildfires into areas where previously there was none. This is the case for example in Scandinavia and regions of Siberia. Increases in burnt area in temperate and boreal forests, and in tundra (Fig 3) were driven by decreased precipitation (as reflected by an increase in the number of dry days) and increased atmospheric dryness (VPD). This was observed in both climate-only and CO2-only sensitivity experiments (see Supplementary Section 5 and 6). However, these regional increases in burnt area were not sufficient to offset the declining global signal, which is driven by reductions in the tropics (Fig 2, Table 2).”

Line 206: *“Increased fire size in these northern regions was driven by increasing dryness and vapor pressure deficit (VPD). Modelling increases in fire intensity in these regions were driven by increased GPP, tree cover but also road density. Larger fires in South America are driven by increased dryness in the tropical rainforests.”*

Line 214: *“Modelled decreases in fire size and fire intensity in tropical Africa, the Middle East and India were primarily driven by changes in human activity (see Fig S12-15). Outside these regions, decreases in fire intensity were primarily driven by increasing VPD (see Fig 2 and 3).”*

Line 230: *“Increased GPP was the major cause of increased fire intensity in northern latitudes, though changes in tree cover and grass cover also played a role. For fire size,*

increases in the number of dry days and dry day seasonality and changes were the main drivers in this region (see Supplementary 3)."

Updated figure 4:

Fig 4. Predictors responsible for driving changes in burnt area (BA), fire size (FS) and fire intensity (FI) respectively of (a) the high climate change mitigation scenario and (b) the low climate change mitigation scenario. The top row shows top predictors driving increases and the bottom row shows predictors driving decreases in the fire properties.

And the following figures have been replaced in the supplementary:

Supplementary Fig S12. Map showing for each grid-cell which variable in the GLM model caused the positive contribution in the future projections for the high climate change mitigation (left) and low climate change mitigation (right) experiment for the GFDL-ESM2M climate model outputs.

Supplementary Fig S13. Map showing for each grid-cell which variable in the GLM model caused the positive contribution in the future projections for the high climate change mitigation (left) and low climate change mitigation (right) experiment for the IPSL-CM5A-LR climate model outputs.

Supplementary Fig S14. Map showing for each grid-cell which variable in the GLM model caused the positive contribution in the future projections for the high climate change mitigation (left) and low climate change mitigation (right) experiment for the Had-GEM2-ES climate model outputs.

Supplementary Fig S15. Map showing for each grid-cell which variable in the GLM model caused the positive contribution in the future projections for the high climate change mitigation (left) and low climate change mitigation (right) experiment for the MIROC5 climate model outputs.”

“Supplementary Fig S16. Map showing for each grid-cell which variable in the GLM model caused the negative contribution in the future projections for the high climate change mitigation (left) and low climate change mitigation (right) experiment for the GFDL-ESM2M climate model outputs.”

Supplementary Fig S17. Map showing for each grid-cell which variable in the GLM model caused the negative contribution in the future projections for the high climate change mitigation (left) and low climate change mitigation (right) experiment for the IPSL-CM5A-LR climate model outputs.

Supplementary Fig S18. Map showing for each grid-cell which variable in the GLM model caused the negative contribution in the future projections for the high climate change mitigation (left) and low climate change mitigation (right) experiment for the Had-GEM2-ES climate model outputs.

Supplementary Fig S19. Map showing for each grid-cell which variable in the GLM model caused the negative contribution in the future projections for the high climate change mitigation (left) and low climate change mitigation (right) experiment for the MIROC5 climate model outputs.

Minor points:

L105: sentence is ambiguous because it could be interpreted that the global decrease in burnt area is primarily driven by decreases in tropical evergreen and deciduous forests themselves, or in the area burned in those veg types. Minor tweak to wording would help.

This section has been reworded and now reads:

Line 123: *“The global decrease in burnt area is primarily driven by a decrease in burnt area in tropical evergreen and deciduous forests, tropical grasslands, tropical shrublands and tropical savannas. These biomes account for 83% of global burnt area under modern conditions but only 77-81% at the end of the 21st century (see Table2). This decline was primarily driven by increased human activity, as measured by road density and cropland cover (see Supplementary 6) and was particularly concentrated in Africa, India, and South-East Asia. In these regions, increased landscape fragmentation was the key driver.”*

L106: “Increased road density in previously fire-free regions also caused increases in burnt area in Europe and the east coast of the United States across all four models”. I find it **very** hard to believe that an increase in roads in eastern USA and northern Europe, where road density is already high, would be the dominant driver of such an effect. It implies that a lack of roads in these heavily populated regions is currently limiting burned area more than the climate. These regions currently have relatively little fire (compared to other parts of those continents) because of more benign climate conditions, not because of a lack of roads. I think this is a likely a spurious finding. Moreover, the fact that the finding held across the four climate models is no evidence of reliability, because (1) it could easily be an issue with the GLMs, (2) road density would have been held constant in each climate scenario, or (3) it could also be an issue with the iterative approach to assigning the most important variable.

We have removed this sentence since this result does not emerge from the new predictor contribution method. Road density is not the dominant driver in these regions; however, the system is still sensitive to changes in this driver (as can be seen from the revised Figure 4). Even when re-working the contribution method as suggested by the reviewer, whilst results do change for burnt area, for fire intensity, the impact of increasing road density remains. Although this may be a counter-intuitive finding, it is in line with previous work that has shown the positive impact of roads in vegetation types that are not adapted to fire (Browning et al., 2024). We see that, in regions such as Europe where population density declines but road density increases under future scenarios, both burnt area and fire intensity increase. Whilst we acknowledge that road density may not be the best variable to represent this process, it is our understanding that this is an indirect response to less people on the landscape managing and fragmenting it, and more potential ignition points introduced through the additional access routes and changes in the local climate. In the northern high latitude, despite very small increases in road density (simulated), we see that it still emerges as the dominant driver.

Whilst we do not believe this finding itself is spurious, the addition of roads in this region may be. We have added the following point to the discussion:

Line 332: “. Furthermore, given that we use a linear model to project future road density, the expansion of roads into the northern high latitude, which is responsible in part to increases in fire intensity may not be fully realistic. Nevertheless, we see that the impact of fragmentation on landscapes has a very different effect depending on the fire property and on if the ecosystem is adapted to fire or not.”

L788: I raised in my first review that the Figshare URL for the code is not accessible and this has not been changes. The URL links to the private project (which is not accessible to others), whereas it is possible to provide a private link for reviewers. As a result, I have not looked at the code.

The code is available at the following DOIs: Haas, Olivia (2023). results scripts. figshare. Software. <https://doi.org/10.6084/m9.figshare.24764571.v4>; Haas, Olivia (2023). run models. figshare. Dataset. <https://doi.org/10.6084/m9.figshare.24764577.v2>; Haas, Olivia (2023). set up the models. figshare. Dataset. <https://doi.org/10.6084/m9.figshare.24764583.v2>; Haas, Olivia (2023). final data. figshare. Dataset. <https://doi.org/10.6084/m9.figshare.24764589.v1>

Point-to-point response to reviewers

Reviewer #1 (Remarks to the Author):

The authors have now addressed all my comments, apart from a couple of very small points. I therefore recommend the paper be accepted, barring some tweaks outlined below that do not need to be re-reviewed:

1. On bias correction: even though you've chosen not to refine this here (which I'm fully happy with - you've now done plenty of model evaluation!), a correct implementation would make for an excellently nerdy follow-up.

We fully agree with the reviewer. We are hoping to do this as a follow-up analysis.

2. There are a few instances, mainly in the new text, where “observed” is used for model results. You should instead use terms like “simulated”, “modelled”, or “reconstructed”. (I always get picked up on that when my papers go to review, so I'll pass on the pain ;)).

We have replaced any phrasing of “observed” with “simulated” when referring to modelled results.

3. In Supplementary Table 6, your response says: “The values are not exactly zero, however, they become zero when rounded up.” Given that some of these zeros look significant, it would be better to use a different unit that avoids rounding to zero. e.g. ha or m² for fire size. I'm not sure what the small-number equivalent would be for W km⁻¹, but there should be a way to avoid zeros in significant results just due to rounding.

We have changed this to m² for fire size and kept it at W km⁻¹ but included the actual number (not rounded) for fire intensity.

	Burnt Area			Fire Size		Fire Intensity	
Observational Product	Predicted	MODIS	GFED5	Predicted	GFA	Predicted	MODIS
	Trend (km ²)	NME	NME	Trend (m ²)	NME	Trend (W.km-1)	NME
All predictors	176.86***	1.51	1.45	0.003*	0.87	- 0.0002***	1.10
Fragmentation and human activity only	- 248.72***	0.87	0.91	0	1.11	0	1.00
Climate predictors only	83.74***	1.17	1.15	0.003*	1.81	0	0.90
GPP and natural vegetation only	87.86***	1.17	1.25	0	1.00	- 0.00004**	1.00

I suspect this paper will encourage a lot of future studies moving away from burned area and towards more meaningful fire regime measures. Thanks for persevering with this and for going all out in your responses. It'll be an important paper for the rest of the community to test and build on, and I look forward to seeing what comes from it.

We appreciate this, thank you very much for your thoughtful and rigorous review.

Defining the “bias-corrected” model

Note, this is mostly (though not entirely) the non-linear version. However, I think assuming linearity from the start just makes the math simpler; functionally, it should still yield the same result.

We start with the model, M:

$$M(t) = f(\beta_H \times H(t) + \beta_c \times C(t))$$

$$f(x) = 1/(1 + e^{-x})$$

Where $M(t)$ is the model, $H(t)$ are human predictors, $C(t)$ are none-human predictors, and the β 's are the relative contributions of each.

Using the chain rule, we can get to this:

$$\frac{dM(t)}{dt} = M(t) \times (1 - M(t)) \times \left[\beta_c \times \frac{dC}{dt} + \beta_h \times \frac{dH}{dt} \right]$$

I *think* authors set the corrected M, M^* as:

$$M^*(t) = M(t) + MH(t) - MC(t)$$

(based on the future correction: $F^*(t) = F(t) + F(t) * (MH(t) - MC(t))/M(t)$. Assuming in the historic version, F->M)

Where we have a human-only run (I'm assuming holding C's to climatological mean, \bar{C} - which should be specified)

$$MH(t) = f(\beta_H \times H(t) + \beta_c \times \bar{C})$$

Other variables-only run (humans fixed at mean \bar{H}):

$$MC(t) = f(\beta_H \times \bar{H} + \beta_c \times C(t))$$

Combining and taking the derivative here:

$$\begin{aligned} \frac{M^*(t)}{dt} = & [M(t) \times (1 - M(t)) - MC(t) \times (1 - MC(t))] \times \beta_c \times \frac{dC(t)}{dt} \\ & + [M(t) \times (1 - M(t)) + MH(t) \times (1 - MH(t))] \times \beta_H \times \frac{dH(t)}{dt} \end{aligned}$$

A bit of a mess, but notice the change in sign combining the M and MC vs M and MH terms for climate and human derivatives. We can get rid of the mess by using the approximation $\overline{M(t)} \approx \overline{MH(t)} \approx \overline{MC(t)}$

Then

$$M(t) \times (1 - M(t)) \approx M_i(t) \times (1 - M_i(t)) \approx \overline{M(t)} \times (1 - \overline{M(t)})$$

And

$$\begin{aligned} \frac{M^*(t)}{dt} \approx & [\overline{M(t)} \times (1 - \overline{M(t)}) - \overline{M(t)} \times (1 - \overline{M(t)})] \times \beta_c \times \frac{dC(t)}{dt} \\ & + [\overline{M(t)} \times (1 - \overline{M(t)}) + \overline{M(t)} \times (1 - \overline{M(t)})] \times \beta_H \times \frac{dH(t)}{dt} \end{aligned}$$

Here, the C contribution cancels and the H contribution doubles.

$$\frac{M^*(t)}{dt} \approx 2 \times [\overline{M(t)} \times (1 - \overline{M(t)})] \times \beta_H \times \frac{dH(t)}{dt}$$

My interpretation here is that the trend in the corrected model is proportional to twice that of the linear version of your GLM for a human-only run. Therefore, M* doesn't clearly represent the burned area under corrected human drivers. It is essentially a counterfactual time series of what BA would look like if human effects were emphasized and climate effects downplayed in this particular manner.

Alternative test

You could, instead, assume that the model is getting the correct none-human response, and see if the expected human response in the observations under this assumption is plausible:

If this is the case, your observed BA, S, would be something like:

$$S(t) = f(\alpha_H \times H(t) + \beta_c \times C(t))$$

And the equivalent human-only signal (which isn't observable in itself, but calculable using the above assumption):

$$SH(t) = f(\alpha_H \times H(t) + \beta_c \times \bar{C})$$

Therefore:

$$\text{logit}(SH(t)) = \alpha_H \times H + \beta_c \times \bar{C}$$

S, M and MC can be written in an equivalent form to get:

$$\text{logit}(SH(t)) - \text{logit}(S(t)) = \beta_c \times (\bar{C}(t) - C(t))$$

As we are assuming S -> SH is the same as M-MC, then:

$$\begin{aligned} \text{logit}(SH(t)) - \text{logit}(S(t)) &\approx \\ \text{logit}(M(t)) - \text{logit}(MC(t)) &= \beta_h \times (H(t) - \bar{H}(t)) \end{aligned}$$

And

$$\begin{aligned} \frac{dSH(t)}{dt} &\approx \frac{d}{dt} [\text{logit}(S(t)) + \beta_h \times (H(t) - \bar{H}(t))] \frac{dSH(t)}{dt} \\ &\approx SH(1 - SH) \\ &\times \left[(S \times (1 - S))^{-1} \times \frac{dS}{dt} + (M \times (1 - M))^{-1} \times \frac{dM}{dt} \right. \\ &\quad \left. - (MC \times (1 - MC))^{-1} \times \frac{dMC}{dt} \right] \end{aligned}$$

Using the same approximation

$M(t) \times (1 - M(t)) \approx M_i(t) \times (1 - M_i(t)) \approx \bar{M}(t) \times (1 - \bar{M}(t))$ as before:

$$\frac{dSH(t)}{dt} \approx \frac{dS}{dt} + \frac{dM}{dt} - \frac{dMC}{dt}$$

That's a very convoluted way of saying that, if both your non-human model runs represent a plausible climate contribution AND the differences in the observed plus modelled, and non-human model trends reflect a plausible trend for human impacts (considering standard papers, Andelas et al., Burton & Lampe, Jones et al. 2022, etc.), then your model is probably acceptable.